# Towards Understanding Why Lookahead Generalizes Better Than SGD and Beyond

**Pan Zhou**[*]    **Hanshu Yan**[*]    **Xiao-Tong Yuan**[†]    **Jiashi Feng**[*]    **Shuicheng Yan**[*]

[*] Sea AI Lab, Singapore

[†] Nanjing University of Information Science & Technology, Nanjing, China

{zhoupan, yanhanshu, fengjs, yansc}@sea.com   xtyuan@nuist.edu.cn

## Abstract

To train networks, lookahead algorithm [1] updates its fast weights $k$ times via an inner-loop optimizer before updating its slow weights once by using the latest fast weights. Any optimizer, e.g. SGD, can serve as the inner-loop optimizer, and the derived lookahead generally enjoys remarkable test performance improvement over the vanilla optimizer. But theoretical understandings on the test performance improvement of lookahead remain absent yet. To solve this issue, we theoretically justify the advantages of lookahead in terms of the excess risk error which measures the test performance. Specifically, we prove that lookahead using SGD as its inner-loop optimizer can better balance the optimization error and generalization error to achieve smaller excess risk error than vanilla SGD on (strongly) convex problems and nonconvex problems with Polyak-Łojasiewicz condition which has been observed/proved in neural networks. Moreover, we show the stagewise optimization strategy [2] which decays learning rate several times during training can also benefit lookahead in improving its optimization and generalization errors on strongly convex problems. Finally, we propose a stagewise locally-regularized lookahead (SLRLA) algorithm which sums up the vanilla objective and a local regularizer to minimize at each stage and provably enjoys optimization and generalization improvement over the conventional (stagewise) lookahead. Experimental results on CIFAR10/100 and ImageNet testify its advantages. Codes is available at https://github.com/sail-sg/SLRLA-optimizer.

## 1 Introduction

Deep neural networks (DNNs) have been successfully applied to various applications, such as image classification [3–8], speech recognition [9–11], and classic games [12, 13]. Typically, their training models can be formally formulated as a finite-sum optimization problem:

$$\min_{\boldsymbol{\theta}} F_{\mathcal{S}}(\boldsymbol{\theta}) \triangleq \frac{1}{n} \sum\nolimits_{i=1}^{n} \ell(f(\boldsymbol{x}_i; \boldsymbol{\theta}), \boldsymbol{y}_i), \tag{1}$$

where $(\boldsymbol{x}, \boldsymbol{y})$ is a sample pair from an unknown distribution $\mathcal{D}$, the loss $\ell(f(\boldsymbol{x}; \boldsymbol{\theta}), \boldsymbol{y})$ measures the discrepancy between the prediction $f(\boldsymbol{x}; \boldsymbol{\theta})$ parameterized by $\boldsymbol{\theta}$ and the target $\boldsymbol{y}$, and $n$ is the training sample number. Besides DNNs, the formulation (1) also encapsulates a large body of other problems, e.g., least square regression and logistic regression. Though many algorithms, e.g., variance-reduced algorithms [14–17] and adaptive gradient algorithms [18, 19], can solve problem (1), SGD [20, 21] is one of the most preferable algorithms because of its efficiency and good generalization [22].

In this work, we are particularly interested in the lookahead algorithm [1] for solving (1). Its core idea is to maintain two kinds of network parameters, i.e. "fast weights" $\boldsymbol{v}$ and "slow weights" $\boldsymbol{\theta}$, and update them in turn. Specifically, in the inner loop, it takes the slow weights $\boldsymbol{\theta}$ as warm-start

initialization and updates the fast weights $k$ times to obtain $\boldsymbol{v}_k$ using any vanilla optimizer; for the outer loop, it updates the slow weights as $\boldsymbol{\theta}_+ = (1 - \alpha)\boldsymbol{\theta} + \alpha\boldsymbol{v}_k$, where $\alpha \in (0, 1]$. Any standard optimizer, e.g. SGD, Adam [18] and RAdam [23], can serve as the inner-loop optimizer, and the derived lookahead algorithm generally enjoys remarkable test performance improvement than the standard optimizer [1]. Because of its simplicity and strong compatibility, lookahead has been widely used [24–29]. However, the theoretical reasons for the superiority in test performance of lookahead are rarely investigated, though heavily desired. Moreover, in practice, to train faster and generalize better, one often uses the stagewise optimization strategy [2], namely running a large learning rate at the beginning and geometrically decaying it several times during the following training. But for lookahead, the theoretical benefits of the stagewise optimization strategy still remain unclear.

**Our Contributions.** In this work, we provide a theoretical viewpoint to understand the test performance improvement of lookahead, and also show the benefits of stagewise optimization strategy in lookahead. Moreover, we further propose a new stagewise locally-regularized lookahead algorithm which provably enjoys faster convergence speed, smaller generalization error, and better test performance than conventional (stagewise) lookahead. Our contributions are highlighted below.

Firstly, we prove that on convex problems, lookahead using SGD as its inner-loop optimizer has optimization error $\mathcal{O}\left(\frac{1}{\alpha\eta kT} + \eta\right)$, where $T$ and $k$ respectively denote the outer- and inner-loop iteration numbers, and $\eta$ is the learning rate for the inner-loop optimizer, i.e. SGD. Then we prove that lookahead has generalization error bound $\mathcal{O}\left(\frac{\alpha\eta kT}{n}\right)$ on convex problems with $n$ training samples. Since the excess risk error can well measure the test performance of an algorithm and is upper bounded by the sum of optimization and generalization errors, we can bound the excess risk error of lookahead with $\mathcal{O}\left(\frac{1}{\alpha\eta kT} + \eta + \frac{\alpha\eta kT}{n}\right)$. When $\alpha = 1$, lookahead degenerates to vanilla SGD and has excess risk error $\mathcal{O}\left(\frac{1}{\eta kT} + \eta + \frac{\eta kT}{n}\right)$. Since the optimum of $\alpha$ in lookahead to optimally balance the optimization and generalization errors is often not one, lookahead can enjoy a smaller excess risk error than vanilla SGD. Similarly, on strongly convex problems and a class of nonconvex problems that obey Polyak-Łojasiewicz (PŁ) condition, our results also show that lookahead can better trade-off optimization and generalization errors and achieve smaller excess risk error than vanilla SGD. For PŁ condition, it is observed /proved for deep learning models in [30–34] and our empirical results. These results well explain the better test performance of lookahead than SGD.

Secondly, we propose a Stagewise Locally-Regularized LookAhead (SLRLA) algorithm, and prove its advantages over lookahead and stagewise lookahead in terms of excess risk error. SLRLA first divides the optimization into several stages. Then at each stage, it minimizes a locally-regularized function that contains the vanilla loss $F_{\mathcal{S}}(\boldsymbol{\theta})$ in (1) and a local regularizer $\frac{\beta}{2}\|\boldsymbol{\theta} - \boldsymbol{\theta}_q\|^2$ with the output $\boldsymbol{\theta}_q$ of the previous stage. A similar stagewise strategy is commonly used in practice, but does not has the local regularizer. Our results show two advantages of SLRLA over lookahead and stagewise lookahead. (i) On strongly convex problems and weakly quasi convex problems (a class of nonconvex problems including convex problems as a special case), SLRLA achieves faster convergence rates (i.e. smaller optimization error), and enjoys smaller generalization error than lookahead and its stagewise variant. (ii) On nonconvex problems under PŁ condition, when the problems are not heavily nonconvex (the smallest eigenvalue of $\nabla^2 F_{\mathcal{S}}(\boldsymbol{\theta})$ is not very small), SLRLA achieves linear convergence rate, and greatly improves optimization and generalization errors in lookahead, while stagewise lookahead cannot enjoy the linear convergence rate and the improvement.

Finally, when $\beta = 0$, SLRLA degenerates to conventional stagewise lookahead, and our results show the advantages of stagewise lookahead over lookahead in terms of excess risk error on strongly convex problems, explaining the benefits of the stagewise strategy in lookahead.

## 2 Related Work

**Optimization Algorithms.** When problem (1) involves deep networks, SGD and adaptive gradient algorithms, e.g. Adam [18] and AdaGrad [19], are more preferable than other algorithms, such as variance-reduced SGD [14, 15, 35, 36], because of their high efficiency and good generalization. Moreover, SGD generally enjoys better generalization performance than adaptive gradient algorithms, since SGD tends to converge to flat minima while adaptive gradient algorithms approach sharp minima [22, 37–41]. To further boost the generalization performance of SGD and adaptive gradient algorithms, Zhang *et al.* [1] proposed lookahead which can employ any standard optimizer as its

inner-loop optimizer and brings remarkable generalization improvement. Recently, a few works analyze lookahed but focus on its optimization performance, e.g. small gradient noise variance on least square problems [1] and sublinear convergence rate on nonconvex problems [42]. In contrast, we analyze the intrinsic theoretical reasons for the superiority of lookahead in terms of test performance.

**Generalization Analysis of Algorithms.** Uniform stability [43] is a classical tool to analyze generalization error of an algorithm. For instance, Hardt *et al.* [44] and Zhang *et al.* [45] analyzed the generalization of SGD via uniform stability. We also utilize stability to analyze generalization but target at analyzing the test performance, including optimization error and generalization error, of an algorithm and its lookahead variant, which is of more practical interest especially in deep learning. Yuan *et al.* [46] analyzed the test performance of SGD and stagewise SGD and showed advantages of the statewised strategy. Differently, we show that lookahead can well balance the optimization and generalization errors, and thus enjoys better test performance than its vanilla inner-loop optimizer.

# 3 Notations and Preliminaries

**Convexity, Lipschitz Continuity, and Smoothness.** For analysis, we first introduce necessary definitions, i.e. convexity, Lipschitz continuity and smoothness. These definitions are commonly used in the convergence and generalization analysis of optimization algorithms, e.g. [47–51].

**Definition 1** (Convexity, Lipschitz Continuity and Smoothness). *We say a function $f(\boldsymbol{\theta})$ is $\lambda$-strongly convex if $\forall \boldsymbol{\theta}_1, \boldsymbol{\theta}_2$, $f(\boldsymbol{\theta}_1) \geq f(\boldsymbol{\theta}_2) + \langle \nabla f(\boldsymbol{\theta}_2), \boldsymbol{\theta}_1 - \boldsymbol{\theta}_2 \rangle + \frac{\lambda}{2} \|\boldsymbol{\theta}_1 - \boldsymbol{\theta}_2\|^2$. If $\lambda = 0$, then we say $f(\boldsymbol{\theta})$ is convex. Moreover, we say $f(\boldsymbol{\theta})$ is $G$-Lipschitz continuous if $\|f(\boldsymbol{\theta}_1) - f(\boldsymbol{\theta}_2)\|_2 \leq G\|\boldsymbol{\theta}_1 - \boldsymbol{\theta}_2\|_2$. $f(\boldsymbol{\theta})$ is said to be $L$-smooth if its gradient obeys $\|\nabla f(\boldsymbol{\theta}_1) - \nabla f(\boldsymbol{\theta}_2)\|_2 \leq L\|\boldsymbol{\theta}_1 - \boldsymbol{\theta}_2\|_2$ ($\forall \boldsymbol{\theta}_1, \boldsymbol{\theta}_2$).*

**Polyak-Łojasiewicz (PŁ) Condition and Weakly Quasi-Convexity.** For nonconvex problems, such two conditions establish the relation between the gradient norm and the loss distance at two points.

**Definition 2** (PŁ Condition & Weakly Quasi-Convexity). *Let $\boldsymbol{\theta}^* \in \operatorname{argmin}_{\boldsymbol{\theta}} f(\boldsymbol{\theta})$. We say a function $f(\boldsymbol{\theta})$ satisfies $\mu$-PŁ condition if it satisfies $2\mu(f(\boldsymbol{\theta}) - f(\boldsymbol{\theta}^*)) \leq \|\nabla f(\boldsymbol{\theta})\|^2$ ($\forall \boldsymbol{\theta}$) with a universal constant $\mu$. $f(\boldsymbol{\theta})$ is said to be $\rho$-weakly-quasi-convex if it obeys $\langle \nabla f(\boldsymbol{\theta}), \boldsymbol{\theta} - \boldsymbol{\theta}^* \rangle \geq \rho(f(\boldsymbol{\theta}) - f(\boldsymbol{\theta}^*))$.*

**Excess Risk Error Decomposition.** Given a dataset $\mathcal{S} = \{(\boldsymbol{x}_i, \boldsymbol{y}_i)\}_{i=1}^n$ where $(\boldsymbol{x}_i, \boldsymbol{y}_i)$ is drawn from an unknown distribution $\mathcal{D}$, one often minimizes the empirical risk $F_{\mathcal{S}}(\boldsymbol{\theta}) \triangleq \frac{1}{n} \sum_{i=1}^n \ell(f(\boldsymbol{x}_i; \boldsymbol{\theta}), \boldsymbol{y}_i)$ in (1) via a randomized algorithm $\mathcal{A}$, e.g. SGD, to find an estimated optimum $\boldsymbol{\theta}_{\mathcal{A},\mathcal{S}} \approx \operatorname{argmin}_{\boldsymbol{\theta}} F_{\mathcal{S}}(\boldsymbol{\theta})$. However, this empirical solution $\boldsymbol{\theta}_{\mathcal{A},\mathcal{S}}$ differs from the desired optimum $\boldsymbol{\theta}_{\mathcal{D}}^*$ of the population risk

$$\boldsymbol{\theta}_{\mathcal{D}}^* \in \operatorname{argmin}_{\boldsymbol{\theta}} F(\boldsymbol{\theta}) \triangleq \mathbb{E}_{(\boldsymbol{x},\boldsymbol{y}) \sim \mathcal{D}}[\ell(f(\boldsymbol{x}; \boldsymbol{\theta}), \boldsymbol{y})].$$

This raises a particularly important question: what performance of the estimated optimum $\boldsymbol{\theta}_{\mathcal{A},\mathcal{S}}$ can achieve on the test data $(\boldsymbol{x}, \boldsymbol{y}) \sim \mathcal{D}$? To answer this question, we analyze the test error $\mathbb{E}_{\mathcal{A},\mathcal{S}}[F(\boldsymbol{\theta}_{\mathcal{A},\mathcal{S}})]$ of $\boldsymbol{\theta}_{\mathcal{A},\mathcal{S}}$ via investigating the well-known ***excess risk error*** $\varepsilon_{\text{exc}}$ defined as

$$\varepsilon_{\text{exc}} = \mathbb{E}_{\mathcal{A},\mathcal{S}}[F(\boldsymbol{\theta}_{\mathcal{A},\mathcal{S}})] - \mathbb{E}_{\mathcal{A},\mathcal{S}}[F_{\mathcal{S}}(\boldsymbol{\theta}_{\mathcal{S}}^*)] = \mathbb{E}_{\mathcal{A},\mathcal{S}}[F(\boldsymbol{\theta}_{\mathcal{A},\mathcal{S}}) - F_{\mathcal{S}}(\boldsymbol{\theta}_{\mathcal{A},\mathcal{S}})] + \mathbb{E}_{\mathcal{A},\mathcal{S}}[F_{\mathcal{S}}(\boldsymbol{\theta}_{\mathcal{A},\mathcal{S}}) - F_{\mathcal{S}}(\boldsymbol{\theta}_{\mathcal{S}}^*)], \tag{2}$$

where $\boldsymbol{\theta}_{\mathcal{S}}^* \in \operatorname{argmin}_{\boldsymbol{\theta}} F_{\mathcal{S}}(\boldsymbol{\theta})$ is the optimum of empirical risk $F_{\mathcal{S}}$. Generally, one can expect very small $\mathbb{E}_{\mathcal{A},\mathcal{S}}[F_{\mathcal{S}}(\boldsymbol{\theta}_{\mathcal{S}}^*)]$, since by selecting a powerful model, e.g., a deep network, one can well fit the data. So to bound the test loss $\mathbb{E}_{\mathcal{A},\mathcal{S}}[F(\boldsymbol{\theta}_{\mathcal{A},\mathcal{S}})]$, one only needs to upper bound the right side of Eqn. (2). The ***optimization error*** $\varepsilon_{\text{opt}} \triangleq \mathbb{E}_{\mathcal{A},\mathcal{S}}[F_{\mathcal{S}}(\boldsymbol{\theta}_{\mathcal{A},\mathcal{S}}) - F_{\mathcal{S}}(\boldsymbol{\theta}_{\mathcal{S}}^*)]$ denotes the difference between the exact optimum $\boldsymbol{\theta}_{\mathcal{S}}^*$ and the estimated solution $\boldsymbol{\theta}_{\mathcal{A},\mathcal{S}}$; the ***generalization error*** $\varepsilon_{\text{gen}} \triangleq \mathbb{E}_{\mathcal{A},\mathcal{S}}[F(\boldsymbol{\theta}_{\mathcal{A},\mathcal{S}}) - F_{\mathcal{S}}(\boldsymbol{\theta}_{\mathcal{A},\mathcal{S}})]$ measures the effects of minimizing empirical risk instead of population risk. So one often analyzes $\varepsilon_{\text{opt}}$ and $\varepsilon_{\text{gen}}$ to compare test performance of different algorithms.

**Uniform Stability and Generalization.** One popular approach to analyze generalization error $\varepsilon_{\text{gen}}$ of a randomized algorithm $\mathcal{A}$ is uniform stability [44, 45]. This is because as shown in following lemma, for an algorithm $\mathcal{A}$, if it is $\epsilon$-uniformly stable, its generalization error is upper bounded by $\epsilon$. In the following, we also analyze the uniform stability of lookahead to bound its generalization error.

**Lemma 1** (Uniform Stability and Generalization Error). *[44] We say a randomized algorithm $\mathcal{A}$ is $\epsilon$-uniformly stable if for all datasets $\mathcal{S} \sim \mathcal{D}$ and $\mathcal{S}' \sim \mathcal{D}$ where $\mathcal{S}$ and $\mathcal{S}'$ differ in at most one sample,*

$$\sup_{(\boldsymbol{x},\boldsymbol{y}) \sim \mathcal{D}} \mathbb{E}_{\mathcal{A}} \left[ \ell(f(\boldsymbol{x}; \boldsymbol{\theta}_{\mathcal{A},\mathcal{S}}); \boldsymbol{y}) - \ell(f(\boldsymbol{x}; \boldsymbol{\theta}_{\mathcal{A},\mathcal{S}'}); \boldsymbol{y}) \right] \leq \epsilon.$$

*Moreover, if $\mathcal{A}$ is $\epsilon$-uniformly stable, then its generalization error $\varepsilon_{gen}$ which is defined as $\varepsilon_{gen} = |\mathbb{E}_{\mathcal{A},\mathcal{S}}[F(\boldsymbol{\theta}_{\mathcal{A},\mathcal{S}}) - F_{\mathcal{S}}(\boldsymbol{\theta}_{\mathcal{A},\mathcal{S}})]|$ satisfies $\varepsilon_{gen} \leq \epsilon$.*

**Algorithm 1:** Lookahead Optimization Procedure $(F_{\mathcal{S}}(\boldsymbol{\theta}), \eta, T, \alpha, k, \boldsymbol{\theta}_0, \mathcal{A}, \mathcal{S})$

---

**Input** : Objective $F_{\mathcal{S}}(\boldsymbol{\theta})$, dataset $\mathcal{S}$, inner-loop optimizer $\mathcal{A}$, inner-loop step number $k$ and
learning rate $\{\{\eta_\tau^{(t)}\}_{\tau=0}^{k-1}\}_{t=1}^{T}$, outer-loop learning rate $\alpha \in (0,1)$, initialization $\boldsymbol{\theta}_0$.

**for** $t = 1, 2, ..., T$ **do**
  $\boldsymbol{v}_0^{(t)} = \boldsymbol{\theta}_{t-1}$;
  **for** $\tau = 1, 2, ..., k$ **do**
    $\boldsymbol{v}_\tau^{(t)} = \mathcal{A}(F_{\mathcal{S}}(\boldsymbol{\theta}), \boldsymbol{v}_{\tau-1}^{(t)}, \eta_{\tau-1}^{(t)}, \mathcal{S})$;
  **end**
  $\boldsymbol{\theta}_t = (1-\alpha)\boldsymbol{\theta}_{t-1} + \alpha \boldsymbol{v}_k^{(t)}$.
**end**

**Output** : $\boldsymbol{\theta}_{\mathcal{A},\mathcal{S}} = \boldsymbol{\theta}_T$ for strongly convex problem; $\boldsymbol{\theta}_{\mathcal{A},\mathcal{S}} = \frac{1}{Tk}\sum_{t=1}^{T}\sum_{\tau=0}^{k-1}\boldsymbol{v}_\tau^{(t)}$ for convex and nonconvex problems.

---

# 4 Excess Risk Analysis of Lookahead Algorithm

The lookahead algorithm [1] to solve problem (1) is described in Algorithm 1. It maintains (i) slow weights $\boldsymbol{\theta}$ updated in the outer loop, and (ii) fast weights $\boldsymbol{v}$ updated in the inner loop. For the inner loop, one can run any standard optimizer $\mathcal{A}$, e.g. SGD or Adam used in deep learning, $k$ steps via an operator $\mathcal{A}(F_{\mathcal{S}}(\boldsymbol{\theta}), \boldsymbol{v}_{\tau-1}^{(t)}, \eta_{\tau-1}^{(t)}, \mathcal{S})$ to update the fast weights $\boldsymbol{v}$, where $k$ is often small, e.g. $k = 5$ in [1]. Here the operator $\mathcal{A}(F_{\mathcal{S}}(\boldsymbol{\theta}), \boldsymbol{v}_{\tau-1}^{(t)}, \eta_{\tau-1}^{(t)}, \mathcal{S})$ denotes a minibatch gradient descent step in SGD or Adam, given the loss $F_{\mathcal{S}}(\boldsymbol{\theta})$, current solution $\boldsymbol{v}_{\tau-1}^{(t)}$, learning rate $\eta_{\tau-1}^{(t)}$, and dataset $\mathcal{S}$. Next, lookahead uses the fast weights $\boldsymbol{v}$ to update the slow weights $\boldsymbol{\theta}$ as $\boldsymbol{\theta} = \boldsymbol{\theta} + \alpha(\boldsymbol{v} - \boldsymbol{\theta})$ with an outer-loop learning rate $\alpha \in [0, 1]$.

Because of its effectiveness and compatibility, lookahead has been widely used to boost the performance of SGD [1], Adam [1, 52], RAdam [23, 24] and natural gradient algorithm [25], and sets new state-of-the-arts in image classification and generation, and machine translation [1, 26–29]. However, there is no theoretical analysis that explicitly justifies the performance improvement of lookahead over the vanilla inner optimizer $\mathcal{A}$, hindering the development of new and more advanced optimizers in a principle way. The following sections aim to solve this problem by comparing the optimization and generalization errors of lookahead with its vanilla inner optimizer $\mathcal{A}$. For analysis, we choose SGD as $\mathcal{A}$, as SGD is widely used in deep learning. In this way, the inner-loop updating becomes

$$\boldsymbol{v}_\tau^{(t)} = \mathcal{A}(F_{\mathcal{S}}(\boldsymbol{\theta}), \boldsymbol{v}_{\tau-1}^{(t)}, \eta_{\tau-1}^{(t)}, \mathcal{S}) = \boldsymbol{v}_{\tau-1}^{(t)} - \eta_{\tau-1}^{(t)}\boldsymbol{g}_{\tau-1}^{(t)},$$

where $\boldsymbol{g}_{\tau-1}^{(t)} = \frac{1}{|\mathcal{B}|}\sum_{(\boldsymbol{x},\boldsymbol{y})\in\mathcal{B}}\nabla\ell(f(\boldsymbol{x};\boldsymbol{v}_{\tau-1}^{(t)});\boldsymbol{y})$. Here $\mathcal{B}$ is the sampled minibatch at the $(t, \tau)$-th iteration. In the following, we investigate the optimization and generalization errors of lookahead, and combine these two errors to upper bound its excess risk error which measures the test performance.

## 4.1 Results on Strongly Convex Problems

To begin with, we first investigate the convergence performance of lookahead when its inner optimizer $\mathcal{A}$ is SGD. Our main results are summarized in Theorem 1. See its proof in Appendix D.1.

**Theorem 1.** *Suppose that $F_{\mathcal{S}}(\boldsymbol{\theta})$ is $\lambda$-strongly convex, and each individual loss $\ell(f(\boldsymbol{x};\boldsymbol{\theta}),\boldsymbol{y})$ is $G$-Lipschitz and $L$-smooth w.r.t. $\boldsymbol{\theta}$. Let $\boldsymbol{\theta}_{\mathcal{S}}^* = \arg\min_{\boldsymbol{\theta}} F_{\mathcal{S}}(\boldsymbol{\theta})$. By setting the inner learning rate $\eta_\tau^{(t)} = \frac{\lambda+L}{\lambda L((t-1)k+\tau+2)}$, the optimization error of the output $\boldsymbol{\theta}_{\mathcal{A},\mathcal{S}}$ of lookahead satisfies*

$$\varepsilon_{opt} = \mathbb{E}_{\mathcal{A},\mathcal{S}}[F_{\mathcal{S}}(\boldsymbol{\theta}_{\mathcal{A},\mathcal{S}}) - F_{\mathcal{S}}(\boldsymbol{\theta}_{\mathcal{S}}^*)] \le \begin{cases} \frac{3L(k+2)^{2\alpha}}{2((T+1)k+2)^{2\alpha}}\mathbb{E}[\|\boldsymbol{\theta}_0-\boldsymbol{\theta}_{\mathcal{S}}^*\|^2] + \frac{16LG^2}{\lambda^2((T+1)k+2)^{2\alpha}(1-2\alpha)}, & 0<\alpha<\frac{1}{2}, \\ \frac{3L(k+2)}{2(T+1)k+2}\mathbb{E}[\|\boldsymbol{\theta}_0-\boldsymbol{\theta}_{\mathcal{S}}^*\|^2] + \frac{16LG^2\log(Tk+2)}{\lambda^2((T+1)k+2)}, & \alpha=\frac{1}{2}, \\ \frac{4L(k+2)^{2\alpha}}{((T+1)k+2)^{2\alpha}}\mathbb{E}[\|\boldsymbol{\theta}_0-\boldsymbol{\theta}_{\mathcal{S}}^*\|^2] + \frac{90LG^2}{\lambda^2(2\alpha-1)(Tk+2)}, & \frac{1}{2}<\alpha\le1. \end{cases}$$

Theorem 1 shows that lookahead using SGD as its inner optimizer can converge on the strongly convex problems. The optimization error $\varepsilon_{\text{opt}}$ has two terms: the first bias term characterizes the effect of initialization $\boldsymbol{\theta}_0$; the second term reveals the impact of the stochastic gradient noise. For

the outer-loop learning rate $\alpha$, it can be observed that with the increase of $\alpha$, $\varepsilon_{\mathrm{opt}}$ becomes smaller. So when $\alpha=1$, lookahead degenerates to vanilla SGD and achieves the smallest optimization error $\mathcal{O}\left(\frac{1}{T^2}+\frac{1}{\lambda^2 Tk}\right)$. This rate matches the one $\mathcal{O}\left(\frac{1}{\lambda^2 Tk}\right)$ of SGD in [53] under the same assumptions, as $k$ is often much smaller than iteration number $T$, e.g. $k=5$ in [1]. So lookahead indeed does not benefit the convergence of SGD. It can be intuitively understood: for every $k$ steps, lookahead updates $\boldsymbol{\theta}$ as $\boldsymbol{\theta}_t = \boldsymbol{\theta}_{t-1} + \alpha(\boldsymbol{v}_k^{(t)} - \boldsymbol{\theta}_{t-1}) = \boldsymbol{\theta}_{t-1} - \alpha \sum_{\tau=0}^{k-1} \eta_\tau^{(t)} \boldsymbol{g}_\tau^{(t)}$ and goes forward slowly due to $\alpha < 1$, while SGD updates $\boldsymbol{\theta}$ as $\boldsymbol{\theta}_t = \boldsymbol{\theta}_{t-1} - \sum_{\tau=0}^{k-1} \eta_\tau^{(t)} \boldsymbol{g}_\tau^{(t)}$ and runs faster, where $\boldsymbol{g}_\tau^{(t)}$ denotes the stochastic gradient.

Next, to bound the excess risk error, we investigate the generalization error of lookahead by analyzing its stability, since as shown in Lemma 1, the uniform stability can upper bound generalization error.

**Theorem 2.** *Suppose the assumptions and parameter setting in Theorem 1 hold. The generalization error of the output $\boldsymbol{\theta}_{\mathcal{A},\mathcal{S}}$ of lookahead satisfies $\varepsilon_{gen} = \mathbb{E}_{\mathcal{A},\mathcal{S}}[F(\boldsymbol{\theta}_{\mathcal{A},\mathcal{S}}) - F_{\mathcal{S}}(\boldsymbol{\theta}_{\mathcal{A},\mathcal{S}})] \leq \frac{16G^2}{n\lambda} \frac{(Tk+1)^\alpha - 1}{((T+1)k+2)^\alpha}$.*

See its proof in Appendix D.2. Theorem 2 shows that when $\alpha \neq 0$, the generalization error $\varepsilon_{\mathrm{gen}}$ of lookahead using SGD as its inner optimizer can be upper bounded by $\mathcal{O}\left(\frac{G^2}{n\lambda}\right)$. Particularly, for $\alpha=0$, $\varepsilon_{\mathrm{gen}}$ becomes zero. This is because $\alpha=0$ means no updating, namely $\boldsymbol{\theta}_t = \boldsymbol{\theta}_0$ $(\forall t)$, and thus $\mathbb{E}_{\mathcal{A},\mathcal{S}}[F(\boldsymbol{\theta}_{\mathcal{A},\mathcal{S}}) - F_{\mathcal{S}}(\boldsymbol{\theta}_{\mathcal{A},\mathcal{S}})] = \mathbb{E}_{\mathcal{S}}[F(\boldsymbol{\theta}_0) - F_{\mathcal{S}}(\boldsymbol{\theta}_0)] = 0$. For the effects of $\alpha$ on $\varepsilon_{\mathrm{gen}}$, one can find that when $\alpha$ increases, $\varepsilon_{\mathrm{gen}}$ also becomes larger. This can be intuitively understood: larger $\alpha$ means quick updating of $\boldsymbol{\theta}_t$. Accordingly, the empirical risk $F_{\mathcal{S}}(\boldsymbol{\theta}_t)$ quickly decreases, while the population risk $F(\boldsymbol{\theta}_t)$ may not due to possible overfitting. Interestingly, when $\alpha=1$ indicating lookahead becomes vanilla SGD, our generalization error bound matches the previous bound $\mathcal{O}\left(\frac{G^2}{\lambda n}\right)$ of vanilla SGD in [44], even though Hardt *et al.* [44] used constant learning rate while we use decaying learning rate.

Based on Theorems 1 and 2, we can derive the excess risk error bound in Corollary 1.

**Corollary 1.** *With the same assumptions and parameter setting in Theorem 1, the excess risk error $\varepsilon_{exc}$ in (2) of the output $\boldsymbol{\theta}_{\mathcal{A},\mathcal{S}}$ of lookahead obeys $\varepsilon_{exc} \leq \varepsilon_{opt} + \varepsilon_{gen}$, where $\varepsilon_{opt}$ and $\varepsilon_{gen}$ are given in Theorems 1 and 2, respectively.*

See its proof in Appendix D.3. Corollary 1 shows that the excess risk error $\varepsilon_{\mathrm{exc}}$ of lookahead using SGD as its inner optimizer satisfies $\varepsilon_{\mathrm{exc}} \leq \varepsilon_{\mathrm{opt}} + \varepsilon_{\mathrm{gen}}$, guaranteeing good test performance of the estimated solution $\boldsymbol{\theta}_{\mathcal{A},\mathcal{S}}$ by lookahead. In this work, we are particularly interested in the effects of $\alpha$ on $\varepsilon_{\mathrm{exc}}$. When $\alpha \in (0, \frac{1}{2})$, $\varepsilon_{\mathrm{exc}}$ is of the order $\mathcal{O}\left(\frac{L}{T^{2\alpha}} + \frac{LG^2}{\lambda^2 T^{2\alpha} k^{2\alpha}} + \frac{G^2}{n\lambda}\left(1 - \frac{1}{T^\alpha k^\alpha}\right)\right)$. As in most cases, the factor $\frac{LG^2}{\lambda^2}$ is much larger than $\frac{G^2}{n\lambda}$, especially for ill-conditioned problems where the strongly convex parameter $\lambda$ is very small, increasing $\alpha$ or the total training iteration number $Tk$ will decrease $\varepsilon_{\mathrm{exc}}$. When $\alpha \in (\frac{1}{2}, 1)$, $\varepsilon_{\mathrm{exc}}$ is of the order $\mathcal{O}\left(e(\alpha)\right)$ where $e(\alpha) = \frac{L}{T^{2\alpha}} + \frac{LG^2}{\lambda^2(2\alpha-1)Tk} + \frac{G^2}{n\lambda}\left(1 - \frac{1}{T^\alpha k^\alpha}\right)$. Then we have $e'(\alpha) = -\frac{2L\ln T}{T^{2\alpha}} - \frac{2LG^2}{\lambda^2(2\alpha-1)^2 Tk} + \frac{G^2}{n\lambda} \frac{\ln(Tk)}{T^\alpha k^\alpha}$. There are two common cases that lead to $e'(\alpha) < 0$: (i) the problem is large-scale but not ill-conditioned, and thus the iteration number $T$ is not large since the problem is easy, leading to $\frac{2L\ln T}{T^{2\alpha}} > \frac{G^2}{n\lambda} \frac{\ln(Tk)}{T^\alpha k^\alpha}$; (ii) the problem is heavily ill-conditioned, but iteration number $T$ is moderate due to the moderate precision requirement, giving $\frac{2LG^2}{\lambda^2(2\alpha-1)^2 Tk} > \frac{G^2}{n\lambda} \frac{\ln(Tk)}{T^\alpha k^\alpha}$. For both cases, increasing $\alpha$ can decrease $\varepsilon_{\mathrm{exc}}$. For other cases, in theory, by choosing a proper $\alpha \in (0, 1)$, one can expect better balance between the optimization error and generalization error, and could achieve a smaller excess risk error than vanilla SGD which corresponds to $\alpha = 1$. As Sec. 3 shows that *the excess risk error can well measure the test performance of an algorithm, our results explain the better test performance of lookahead than SGD.* Moreover, though in practice, it is hard to precisely decide whether $e'(\alpha)$ is positive or not, from the above discussion, at least we know that $\alpha$ should be selected from the range $[\frac{1}{2}, 1]$, providing some guidance to set $\alpha$. Finally, our result is the first one that uses the same learning rate strategy to analyze both optimization and generalization errors, and matches the lower bounds of optimization error [53] and generalization error [45] of SGD.

## 4.2 Results on Convex Problems

Now we analyze lookahead using SGD as its inner optimizer $\mathcal{A}$ on the convex problems. Our main results are summarized in Theorem 3 with proof in Appendix D.4.

**Theorem 3.** *Suppose that $F_{\mathcal{S}}(\boldsymbol{\theta})$ is convex, and each loss $\ell(f(\boldsymbol{x};\boldsymbol{\theta}),\boldsymbol{y})$ is $G$-Lipschitz w.r.t. $\boldsymbol{\theta}$. By setting the inner learning rate $\eta_{\tau}^{(t)} = \eta$, we have following properties.*
*(1) The optimization error $\varepsilon_{opt}$ of the output $\boldsymbol{\theta}_{\mathcal{A},\mathcal{S}}$ of lookahead satisfies $\varepsilon_{opt} = \mathbb{E}_{\mathcal{A},\mathcal{S}}[F_{\mathcal{S}}(\boldsymbol{\theta}_{\mathcal{A},\mathcal{S}}) - F_{\mathcal{S}}(\boldsymbol{\theta}_{\mathcal{S}}^{*})] \leq \frac{\Delta}{2\alpha\eta kT} + \frac{\eta G^2}{2}$, where $\boldsymbol{\theta}_{\mathcal{S}}^{*} \in \arg\min_{\boldsymbol{\theta}} F_{\mathcal{S}}(\boldsymbol{\theta})$ and $\Delta = \mathbb{E}\left[\|\boldsymbol{\theta}_0 - \boldsymbol{\theta}_{\mathcal{S}}^{*}\|^2\right]$.*
*(2) The generalization error $\varepsilon_{gen}$ of the output $\boldsymbol{\theta}_{\mathcal{A},\mathcal{S}}$ obeys $\varepsilon_{gen} = \mathbb{E}_{\mathcal{A},\mathcal{S}}[F(\boldsymbol{\theta}_{\mathcal{A},\mathcal{S}}) - F_{\mathcal{S}}(\boldsymbol{\theta}_{\mathcal{A},\mathcal{S}})] \leq \frac{\alpha\eta G^2 kT}{n}$.*
*(3) The excess risk error $\varepsilon_{exc}$ of the output $\boldsymbol{\theta}_{\mathcal{A},\mathcal{S}}$ of lookahead satisfies $\varepsilon_{exc} \leq \varepsilon_{opt} + \varepsilon_{gen}$.*

To begin with, Theorem 3 guarantees the convergence of lookahead on the convex problems. By setting $\eta = \frac{\Delta^{0.5}}{\alpha^{0.5}k^{0.5}T^{0.5}G}$, lookahead achieves the optimization error $\mathcal{O}\left(\frac{G\Delta^{0.5}}{\alpha^{0.5}k^{0.5}T^{0.5}}\right)$. Moreover, when $\alpha$ increases, the optimization error $\varepsilon_{\mathrm{opt}}$ decreases. This accords with the analysis results on the strongly convex problems that large $\alpha$ can reduce $\varepsilon_{\mathrm{opt}}$. When $\alpha=1$, lookahead degenerates to vanilla SGD and its optimization error matches the one of vanilla SGD in [54] under the same assumptions.

The second part of Theorem 3 shows that the generalization error $\varepsilon_{\mathrm{gen}}$ of lookahead is bounded by $\mathcal{O}\left(\frac{\alpha\eta G^2 kT}{n}\right)$. When $\alpha = 1$, this error bound is consistent with the lower bound $\mathcal{O}\left(\frac{kT}{n}\right)$ of SGD with a constant learning rate $\eta$ in [45]. Moreover, one can find that smaller $\alpha$ can lead to a smaller generalization error, which also accords with the results on the strongly convex problems.

Finally, by combining the optimization error $\varepsilon_{\mathrm{opt}}$ and generalization error $\varepsilon_{\mathrm{gen}}$, we can bound the excess risk error $\varepsilon_{\mathrm{exc}}$ of lookahead by $\mathcal{O}\left(\frac{\Delta}{2\alpha\eta kT} + \frac{\eta G^2}{2} + \frac{\alpha\eta G^2 kT}{n}\right)$. So when fixing the learning rate, tuning $\alpha \in (0, 1]$ can yield smaller excess risk error. It means that *a proper $\alpha$ can benefit lookahead in terms of excess risk error, explaining the better test performance of lookahead than SGD.*

### 4.3 Results on Nonconvex Problems

For general nonconvex problems, one often uses the gradient norm $\mathbb{E}[\|\nabla F_{\mathcal{S}}(\boldsymbol{\theta})\|^2]$ instead of the loss distance $\mathbb{E}[F_{\mathcal{S}}(\boldsymbol{\theta}) - F_{\mathcal{S}}(\boldsymbol{\theta}_{\mathcal{S}}^{*})]$ to measure whether $\boldsymbol{\theta}$ is a stationary point. This is because many stationary points may exist in a nonconvex problem. But as shown in Sec. 3, to bound the excess risk error, one needs to bound the loss distance. To solve this issue, we follow [46] and are particularly interested in nonconvex problems under PŁ condition which allows us to bound the loss distance, since PŁ condition in Definition 2 establishes the relation between gradient norm and loss distance. Moreover, as observed/proved in [30–34] and our empirical results in Sec. 6.2, deep learning models often satisfy PŁ condition. We summarize our main results in Theorem 4 with proof in Appendix D.5.

**Theorem 4.** *Assume each loss $\ell(f(\boldsymbol{x};\boldsymbol{\theta}),\boldsymbol{y})$ is $G$-Lipschitz and $L$-smooth w.r.t. $\boldsymbol{\theta}$. Suppose that $F_{\mathcal{S}}(\boldsymbol{\theta})$ obeys the $\mu$-PŁ condition. By setting $\eta_{\tau}^{(t)} = \frac{1}{tk+\tau+1}$ and $\alpha > \frac{1}{2}$, we have following properties.*
*(1) The optimization error of the output $\boldsymbol{\theta}_{\mathcal{A},\mathcal{S}}$ produced by lookahead satisfies*

$$\varepsilon_{opt} = \mathbb{E}_{\mathcal{A},\mathcal{S}}[F_{\mathcal{S}}(\boldsymbol{\theta}_{\mathcal{A},\mathcal{S}}) - F_{\mathcal{S}}(\boldsymbol{\theta}_{\mathcal{S}}^{*})] \leq \frac{4\Delta'}{(Tk+1)^{2\alpha}} + \frac{2\alpha LG^2\left(\alpha + 2(1-\alpha)(k-1)\right)}{\mu^2(Tk+1)^{2\alpha-1}},$$

*where $\boldsymbol{\theta}_{\mathcal{S}}^{*} \in \arg\min_{\boldsymbol{\theta}} F_{\mathcal{S}}(\boldsymbol{\theta})$ and $\Delta' = \mathbb{E}\left[F_{\mathcal{S}}(\boldsymbol{\theta}_0) - F_{\mathcal{S}}(\boldsymbol{\theta}_{\mathcal{S}}^{*})\right]$.*
*(2) The generalization error of the output $\boldsymbol{\theta}_{\mathcal{A},\mathcal{S}}$ produced by lookahead satisfies*

$$\varepsilon_{gen} = \mathbb{E}_{\mathcal{A},\mathcal{S}}[F(\boldsymbol{\theta}_{\mathcal{A},\mathcal{S}}) - F_{\mathcal{S}}(\boldsymbol{\theta}_{\mathcal{A},\mathcal{S}})] \leq \frac{\xi}{n-1}\alpha^{\frac{1}{1+\gamma}}(Tk)^{\frac{\gamma}{\gamma+1}},$$

*where $\gamma = (1 - \frac{1}{n})\frac{\alpha L}{\mu}$ and $\xi = \ell_{max}^{\frac{\gamma}{1+\gamma}}\left[\frac{2G^2}{\mu}\right]^{\frac{1}{1+\gamma}}$ in which $\ell_{max} = \max_{\boldsymbol{\theta},(\boldsymbol{x},\boldsymbol{y})} \ell(f(\boldsymbol{x};\boldsymbol{\theta}),\boldsymbol{y})$.*
*(3) The excess risk error $\varepsilon_{exc}$ of the output $\boldsymbol{\theta}_{\mathcal{A},\mathcal{S}}$ of lookahead satisfies $\varepsilon_{exc} \leq \varepsilon_{opt} + \varepsilon_{gen}$.*

For the optimization error $\varepsilon_{\mathrm{opt}}$ of lookahead on nonconvex problems, it is of the order $\mathcal{O}\left(\frac{1}{(Tk)^{2\alpha}} + \frac{1}{\mu^2(Tk)^{2\alpha-1}}\right)$. The same as (strongly) convex problems, larger $\alpha$ benefits the convergence of lookahead, for which we have discussed the reasons in Sec. 4.1. When $\alpha=1$, lookahead degenerates to SGD and achieves the smallest optimization error $\mathcal{O}\left(\frac{1}{(Tk)^2} + \frac{1}{\mu^2 Tk}\right)$ which matches the one of SGD in [55].

For the generalization error $\varepsilon_{\mathrm{gen}}$, it is of the order $\mathcal{O}\left(\frac{1}{n}\alpha^{\frac{1}{1+\gamma}}(Tk)^{\frac{\gamma}{\gamma+1}}\right)$ which accords with the one of SGD in [44]. The sublinear dependence on $kT$ and the inverse linear dependence on $n$ also match the

**Algorithm 2:** Stagewise Locally-Regularized LookAhead (SLRLA)
___
**Input**   : Loss $F_{\mathcal{S}}(\boldsymbol{\theta})$, constant $\{\beta_q\}_{q=1}^Q$, inner-loop iteration number $\{k_q\}_{q=1}^Q$ and learning rate
$\{\eta_q\}_{q=1}^Q$, outer-loop learning rate $\{\alpha_q\}_{q=1}^Q$, inner optimizer $\mathcal{A}$, dataset $\mathcal{S}$, initialization
$\boldsymbol{\theta}_0$.
**for** $q = 1, 2, ..., Q$ **do**
$\quad$ $F_q(\boldsymbol{\theta}) = F_{\mathcal{S}}(\boldsymbol{\theta}) + \frac{\beta_q}{2}\|\boldsymbol{\theta} - \boldsymbol{\theta}_{q-1}\|^2$;
$\quad$ $\boldsymbol{\theta}_q = \text{Look-ahead}(F_q(\boldsymbol{\theta}), \eta_q, T_q, \alpha_q, k_q, \boldsymbol{\theta}_{q-1}, \mathcal{A}, \mathcal{S})$.
**end**
**Output** : $\boldsymbol{\theta}_{\mathcal{A}, \mathcal{S}} = \boldsymbol{\theta}_Q$.
___

lower bound in [45]. Similarly, to achieve smaller generalization error, one should use small $\alpha$. For the excess risk error $\varepsilon_{\text{exc}}$, it is bounded by $\mathcal{O}\left(\varepsilon_{\text{opt}} + \varepsilon_{\text{gen}}\right)$. So similar to (strongly) convex problems, *when $\alpha$ is well chosen, the optimization error $\varepsilon_{opt}$ and generalization error $\varepsilon_{gen}$ can be balanced well, giving smaller excess risk error and better test performance than SGD which corresponds to $\alpha = 1$.*

Regarding the lookahead method with inner optimizers other than SGD, we believe that one could still expect similar performance trade-off between optimization and generation with respect to the choice of $\alpha$. Intuitively, for any inner optimizer, let $g_\tau^t$ denote the "gradient" (or any descent direction) at the $(t, \tau)$-th iteration. After the $k$ inner-steps, lookahead updates parameter $\theta$ as $\boldsymbol{\theta}_t = \boldsymbol{\theta}_{t-1} + \alpha(\boldsymbol{v}_k^{(t)} - \boldsymbol{\theta}_{t-1}) = \boldsymbol{\theta}_{t-1} - \alpha\sum_{\tau=0}^{k-1}\eta_\tau^{(t)}\boldsymbol{g}_\tau^{(t)}$. Obviously, $\alpha \approx 1$ is preferable for preserving the optimization speed of the inner optimizer. When it comes to the generalization error, obviously the best possible performance occurs at the initialization point as it is not dependent on the training data. Along with more training iterations, the network will gradually fit the training data and thus could give larger and larger prediction discrepancy between training data and test data. Therefore, it is desirable to have $\alpha \ll 1$ as opposed to $\alpha \approx 1$ for generalization. Overall, for generic inner-loop optimizers, lookahead is still expected to be able to balance the optimization and generalization performances with proper choices of $\alpha$.

## 5    Stagewise Locally-Regularized Lookahead

We introduce the Stagewise Locally-Regularized LookAhead (SLRLA) algorithm in Algorithm 2. SLRLA divides the optimization into $Q$ stages. For the $q$-th stage, it first combines the vanilla loss $F_{\mathcal{S}}(\boldsymbol{\theta})$ and a local regularizer $\frac{\beta_q}{2}\|\boldsymbol{\theta} - \boldsymbol{\theta}_{q-1}\|_2^2$ to construct a locally regularized loss $F_q(\boldsymbol{\theta}) = F_{\mathcal{S}}(\boldsymbol{\theta}) + \frac{\beta_q}{2}\|\boldsymbol{\theta} - \boldsymbol{\theta}_{q-1}\|_2^2$. Here $\boldsymbol{\theta}_{q-1}$ is the output of the $(q-1)$-th stage, and $\beta_q$ is a constant. The intuition behind this local regularizer is that (i) it improves the convexity of the loss, e.g. converting an ill-conditioned loss to a well-conditioned one, accelerating convergence; (ii) it may avoid overfitting by preventing current solution $\boldsymbol{\theta}$ having extreme values and far from $\boldsymbol{\theta}_{q-1}$. Theoretically, for $\lambda$-strongly-convex problems, as shown in Theorems 1 & 2 that both optimization and generalization errors depend on $\mathcal{O}(1/\lambda)$. For nonconvex problems under $\mu$-PŁ condition, Theorem 4 also shows that both optimization and generalization error scale with $\mathcal{O}(1/\mu)$. So on strongly convex problems, adding a regularization on the vanilla loss can enhance the convexity and thus reduces both optimization and generalization errors; on nonconvex problems, regularization can also increase the PŁ condition parameter $\mu$ and thus helps optimization and generalization. We use $\|\boldsymbol{\theta} - \boldsymbol{\theta}_{q-1}\|_2^2$ instead of $\|\boldsymbol{\theta} - \mathbf{0}\|^2$ or $\|\boldsymbol{\theta} - \boldsymbol{\theta}_0\|_2^2$. This is because compared with $\mathbf{0}$ and $\boldsymbol{\theta}_0$, $\boldsymbol{\theta}_{q-1}$ is closer to the optimum $\boldsymbol{\theta}_{\mathcal{S}}^*$ of $F_{\mathcal{S}}(\boldsymbol{\theta})$ and thus allows to use larger $\beta_q$, improving the convexity of a loss more and benefiting convergence and generalization more (see discussion below). Next, SLRLA uses lookahead, i.e. Algorithm 1, to minimize $F_q(\boldsymbol{\theta})$, where constant inner- and outer-loop learning rates $\eta_q$ and $\alpha_q$ are used. A similar conventional stagewise optimization strategy which however does not regularize $F_{\mathcal{S}}(\boldsymbol{\theta})$ is widely used in SGD. But it is theoretically unclear whether conventional stagewise strategy benefits lookahead.

In the following sections, we investigate two questions: (i) whether the conventional stagewise strategy improves lookahead; (ii) what advantages SLRLA has over the conventional (stagewise) lookahead. For (i), we show the advantages of stagewise lookahead over vanilla lookahead in terms of the optimization error. For (ii), we prove that SLRLA can improve the optimization and generalization of conventional (stagewise) lookahead because of the local regularizer in SLRLA.

## 5.1 Results on Strongly Convex Problems

We analyze the optimization and generalization errors of SLRLA in Theorem 5 with proof in Appendix E.1, and then analyze the aforementioned two questions.

**Theorem 5.** *Suppose that $F_{\mathcal{S}}(\boldsymbol{\theta})$ is $\lambda$-strongly convex, and each loss $\ell(f(\boldsymbol{x};\boldsymbol{\theta}),\boldsymbol{y})$ is $G$-Lipschitz and $L$-smooth w.r.t. $\boldsymbol{\theta}$. By setting $\varepsilon_q = \frac{\Delta}{2^q}$, $\beta_q = \beta \leq \frac{1}{6}\lambda$, $\alpha_q = \alpha \in [0,1]$, $\eta_q \leq \frac{\varepsilon_q}{3G^2}$, $\eta_q k_q T_q \geq \frac{6}{\lambda \alpha_q}$, $\Delta' = \mathbb{E}[F_{\mathcal{S}}(\boldsymbol{\theta}_0) - F_{\mathcal{S}}(\boldsymbol{\theta}_{\mathcal{S}}^*)]$ and $\boldsymbol{\theta}_{\mathcal{S}}^* \in \operatorname{argmin}_{\boldsymbol{\theta}} F_{\mathcal{S}}(\boldsymbol{\theta})$, the following properties hold.*
*(1) The optimization error $\varepsilon_{opt}$ of the output $\boldsymbol{\theta}_{\mathcal{A},\mathcal{S}}$ of SLRLA satisfies $\varepsilon_{opt} \leq \frac{\Delta'}{2^Q}$. Moreover, to achieve $\varepsilon_{opt} \leq \epsilon$, $Q$ satisfies $Q \geq \log \frac{\Delta'}{\epsilon}$ and the stochastic gradient complexity is $\sum_{q=1}^{Q} T_q k_q = \frac{36G^2}{\lambda \alpha \epsilon}$.*
*(2) The generalization error $\varepsilon_{gen}$ of the output $\boldsymbol{\theta}_{\mathcal{A},\mathcal{S}}$ of SLRLA satisfies $\varepsilon_{gen} \leq \frac{\gamma_1}{n}\left(\frac{\beta}{\alpha} + \frac{\lambda L}{\lambda+L}\right)^{-1}\left(1 - \exp\left(-\frac{6QL}{\lambda+L}\right)\right)$, where $\gamma_1 = 2G^2/\left(1 - \exp\left(-\frac{6L}{\lambda+L}\right)\right)$.*
*(3) The excess risk error $\varepsilon_{exc}$ of the output $\boldsymbol{\theta}_{\mathcal{A},\mathcal{S}}$ of lookahead satisfies $\varepsilon_{exc} \leq \varepsilon_{opt} + \varepsilon_{gen}$.*

By Theorem 5, we here show the advantages of the Stagewise LookAhead (SLA for short) [1] over lookahead, and also discuss the superiority of SLRLA over SLA. When $\beta_q = 0$ and the learning rate is geometrically decayed as SLRLA after each stage, SLRLA degenerates to SLA, and Theorem 5 still holds. Theorem 5 shows the linear convergence of both SLRLA and SLA w.r.t. the stage number $Q$. For both SLRLA and SLA, to achieve optimization error $\epsilon$, i.e. $\varepsilon_{\mathrm{opt}} = \mathbb{E}[F_{\mathcal{S}}(\boldsymbol{\theta}_{\mathcal{A},\mathcal{S}}) - F_{\mathcal{S}}(\boldsymbol{\theta}_{\mathcal{S}}^*)] \leq \epsilon$, $Q$ is of the order $\mathcal{O}\left(\log \frac{1}{\epsilon}\right)$ and the stochastic gradient complexity (evaluation number) is $\sum_{q=1}^{Q} T_q k_q = \mathcal{O}\left(\frac{G^2}{\lambda \alpha \epsilon}\right)$. With the optimization error $\epsilon$, Theorem 1 shows that vanilla lookahead needs stochastic gradient complexity $\mathcal{O}\left(\left(\frac{L}{\epsilon}\right)^{\frac{1}{2\alpha}} + \left(\frac{LG^2}{(1-2\alpha)\lambda^2\epsilon}\right)^{\frac{1}{2\alpha}}\right)$ for $\alpha \in (0,\frac{1}{2})$, $\mathcal{O}\left(\frac{LG^2 \log \frac{1}{\epsilon}}{\lambda^2 \epsilon}\right)$ for $\alpha = \frac{1}{2}$, and $\mathcal{O}\left(\frac{LG^2}{(2\alpha-1)\lambda^2\epsilon}\right)$ for $\alpha \in (\frac{1}{2},1]$. By comparison, both SLA and SLRLA improve the dependences on two important factors, i.e. $\lambda$ and $\alpha$, in vanilla lookahead. Specifically, for factor $\lambda$, SLA and SLRLA rely on $\mathcal{O}\left(\frac{1}{\lambda}\right)$, while lookahead depends on at least $\mathcal{O}\left(\frac{1}{\lambda^2}\right)$. For factor $\alpha$, SLA and SLRLA only linearly depend on $\mathcal{O}\left(\frac{1}{\alpha}\right)$. In contrast, when $\alpha \in (0,\frac{1}{2})$, lookahead exponentially depends on $\frac{1}{\epsilon}$ due to the term $\mathcal{O}\left(\left(\frac{1}{\epsilon}\right)^{\frac{1}{2\alpha}}\right)$. For $\alpha = \frac{1}{2}$, SLA and SLRLA improve lookahead by removing the logarithm factor $\log \frac{1}{\epsilon}$. When $\alpha \in (\frac{1}{2},1]$, SLA and SLRLA also enjoy smaller dependence on $\alpha$ than lookahead, as the factor $\frac{1}{\alpha}(\leq 2)$ in SLA and SLRLA is often smaller than the factor $\frac{1}{2\alpha-1}$ in lookahead.

For generalization error, by comparing Theorems 5 and 2, SLA and lookahead enjoy the same generalization error bound $\mathcal{O}\left(\frac{G^2}{n\lambda}\right)$, while SLRLA has superior one $\mathcal{O}\left(\frac{G^2}{n(\beta/\alpha+\lambda)}\right)$ especially for small $\alpha$. This is because $\beta$ can be at the same order of $\lambda$ in Theorem 5. By combining the optimization and generalization errors together, SLRLA enjoys smaller excess risk error than SLA which however outperforms lookahead, when the computational budget (stochastic gradient complexity) is the same.

## 5.2 Results on Nonconvex Problems

Now we analyze SLRLA on two classes of nonconvex problems. The first one requires the smallest eigenvalue $(-\sigma)$ of the Hessian $\nabla^2 F_{\mathcal{S}}(\boldsymbol{\theta})$ to satisfy $\sigma \leq \frac{1}{6}\mu$, where $\mu$ is the parameter in $\mu$-PŁ condition in Definition 2. It actually means the function is not heavily nonconvex. The second one satisfies the weakly-quasi-convex assumption which guarantees that any local minimizer of the loss is also a global minimizer. Indeed, any convex function is 1-weakly-quasi-convex, but the converse is generally not true. We use these two classes of nonconvex problems as examples to investigate the aforementioned two problems. Theorem 6 with proof in Appendix E.2 summarizes the main results.

**Theorem 6.** *Assume each $\ell(f(\boldsymbol{x};\boldsymbol{\theta}),\boldsymbol{y})$ is $G$-Lipschitz and $L$-smooth w.r.t. $\boldsymbol{\theta}$, and $F_{\mathcal{S}}(\boldsymbol{\theta})$ satisfies $\mu$-PŁ condition. Let $\Delta' = \mathbb{E}[F(\boldsymbol{\theta}_0) - F(\boldsymbol{\theta}_{\mathcal{S}}^*)]$ and $\varepsilon_q = \frac{\Delta'}{2^q}$, where $\boldsymbol{\theta}_{\mathcal{S}}^* \in \operatorname{argmin}_{\boldsymbol{\theta}} F_{\mathcal{S}}(\boldsymbol{\theta})$.*
*(1) When $\sigma \leq \frac{1}{6}\mu$ where $\nabla^2 F_{\mathcal{S}}(\boldsymbol{\theta}) \succeq -\sigma \boldsymbol{I} (\forall \boldsymbol{\theta})$, by setting $\sigma \leq \beta_q = \beta \leq \frac{1}{6}\mu$, $\eta_q \leq \frac{\varepsilon_q}{3G^2}$, $\eta_q k_q T_q \geq \frac{6}{\mu \alpha_q}$, the output $\boldsymbol{\theta}_{\mathcal{A},\mathcal{S}}$ of SLRLA satisfies*

$$\varepsilon_{opt} \leq \frac{\Delta'}{2^Q}, \qquad \varepsilon_{gen} \leq \frac{\gamma_2}{n}\left(\frac{\beta-\sigma}{\alpha} + \frac{\mu L}{\mu+L}\right)^{-1}\left(1 - \exp\left(-\frac{6QL}{\lambda+L}\right)\right), \qquad \varepsilon_{exc} \leq \varepsilon_{opt} + \varepsilon_{gen},$$

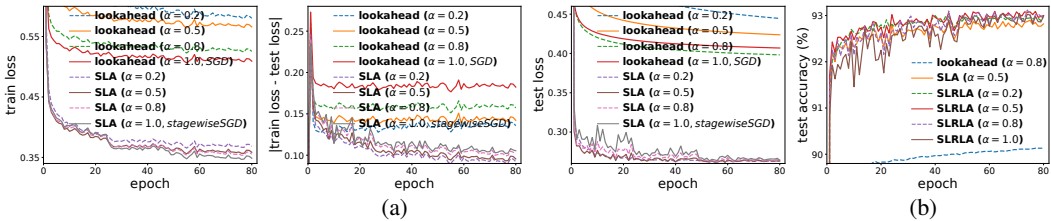

Figure 1: Investigation of lookahead, SLA and SLRLA on a softmax problem with MNIST. (a) reveals the effects of $\alpha$ on convergence, generalization and test performance of lookahead and SLA. (b) shows the impact of $\alpha$ on test performance of SLRLA, and further compares the test performance.

where $\gamma_2 = 2G^2/\left(1 - \exp\left(-\frac{6L}{\mu+L}\right)\right)$. *Moreover, to achieve $\varepsilon_{opt} \leq \epsilon$, the total stage number $Q$ satisfies $Q \geq \log \frac{\Delta}{\epsilon}$ and the total stochastic gradient complexity is $\sum_{q=1}^{Q} T_q k_q = \frac{36G^2}{\mu\alpha\epsilon}$.*
*(2) When $F_q(\boldsymbol{\theta})$ is $\rho$-weakly-quasi-convex, by setting $\beta_q = \beta \leq \frac{1}{6}\mu, \eta_q \leq \frac{\rho\varepsilon_q}{3G^2}, \eta_q k_q T_q \geq \frac{6}{\mu\rho\alpha_q}$, $\alpha \leq \frac{\beta}{L}$, the output $\boldsymbol{\theta}_{\mathcal{A},\mathcal{S}}$ of SLRLA satisfies*

$$\varepsilon_{opt} \leq \frac{\Delta}{2^Q}, \qquad \varepsilon_{gen} \leq \frac{\gamma_2}{n}\left(\frac{\beta}{\alpha} - L\right)^{-1}\left(1 - \exp\left(-\frac{6S(\beta - \alpha L)}{\mu\rho\alpha}\right)\right), \qquad \varepsilon_{exc} \leq \varepsilon_{opt} + \varepsilon_{gen},$$

where $\gamma_2 = 2G^2/\left(1 - \exp\left(-\frac{6(\beta-\alpha L)}{\mu\rho\alpha}\right)\right)$. *Moreover, to achieve $\varepsilon_{opt} \leq \epsilon$, the total stage number $Q$ satisfies $Q \geq \log \frac{\Delta}{\epsilon}$ and the total stochastic gradient complexity is $\sum_{q=1}^{Q} T_q k_q = \frac{36G^2}{\mu\rho^2\alpha\epsilon}$.*

Here we discuss the advantages of SLRLA over SLA and lookahead. Theorem 6 shows that for the nonconvex loss $F_{\mathcal{S}}(\boldsymbol{\theta})$ which obeys $\nabla^2 F_{\mathcal{S}}(\boldsymbol{\theta}) \succeq -\frac{1}{6}\mu\boldsymbol{I}$, SLRLA enjoys linear convergence rate and has stochastic gradient complexity $\mathcal{O}\left(\frac{G^2}{\mu\alpha\epsilon}\right)$ to achieve $\varepsilon_{\text{opt}} \leq \epsilon$. Compared with lookahead which has complexity $\mathcal{O}\left(\left(\frac{LG^2}{\mu^2\epsilon}\right)^{1/\alpha}\right)$, SLRLA improves the factor $\frac{1}{\mu^{2/\alpha}}$ in lookahead to $\frac{1}{\mu}$, and also reduces its exponential dependence on $\frac{1}{\alpha}$ to linear dependence. These improvements accord with the ones on the strongly convex problems. For generalization error $\varepsilon_{\text{gen}}$, SLRLA has an upper bound $\mathcal{O}\left(\frac{1}{n}/\left(\frac{\beta-\sigma}{\alpha}+\mu\right)\right)$ that does not reply on the total iteration number $Tk$. In contrast, Theorem 4 shows that lookahead has generalization error bound $\mathcal{O}\left(\frac{1}{n}(Tk)^{\frac{\gamma}{\gamma+1}}\right)$ sublinearly relying on $Tk$. So SLRLA enjoys smaller excess risk error than lookahead. For SLA, Theorem 6 cannot guarantee its linear convergence and small generalization error, as Theorem 6 requires $\beta_q \geq \sigma > 0$ but SLA needs $\beta_s = 0$. It is because the regularizer $\frac{\beta_q}{2}\|\boldsymbol{\theta} - \boldsymbol{\theta}_{q-1}\|^2$ in SLRLA transforms the nonconvex loss $F_{\mathcal{S}}(\boldsymbol{\theta})$ into a strongly convex one and greatly accelerates the convergence. This shows the advantage of SLRLA over SLA.

On weakly-quasi-convex problems, SLRLA has stochastic gradient complexity $\mathcal{O}\left(\frac{G^2}{\mu\rho^2\alpha\epsilon}\right)$ to achieve $\varepsilon_{\text{opt}} \leq \epsilon$, and has generalization error $\mathcal{O}\left(\frac{\gamma_2}{n}/\left(\frac{\beta}{\alpha} - L\right)\right)$. When viewing $\rho$ as a constant, then same as the above case, SLRLA makes improvements on vanilla lookahead in terms of both stochastic complexity and generalization error. Moreover, the results on SLRLA also do not hold for SLA, since Theorem 6 needs $\beta_q \geq \alpha L > 0$ while SLA sets $\beta_q = 0$. Thus, all these results show the advantages of SLRLA over SLA and lookahead on achieving smaller excess risk error and thus enjoying better test performance.

**Limitation Discussion.** As explained in Sec. 4.3, one cannot bound the loss distance $\mathbb{E}[F_{\mathcal{S}}(\boldsymbol{\theta}) - F_{\mathcal{S}}(\boldsymbol{\theta}_{\mathcal{S}}^*)]$ for general nonconvex probems (GNP). So our main limitation is that our analysis is not applicable to GNP. But we analyze lookahead on GNP under PŁ condition which is observed/proved for deep networks [30–34] and our empirical results. See more discussions in Appendix A.

**Societal Impact.** This work analyzes the intrinsic theoretical reasons for the superiority of lookahead in terms of test performance, and further proposes a general and more advanced deep learning optimizer with provable improvement over lookahead which could advance deep network training. For the negative social impact of this work, it is mainly determined by which applications the developed optimizer is applied to.

# 6 Experiments

Table 1: Classification accuracy (%). $\diamond$, $*$, $\dagger$, $\ddagger$ are respectively reported in [1], [23], [61], [62].

| optimizer | CIFAR10 | | | CIFAR100 | | | ImageNet |
|---|---|---|---|---|---|---|---|
| | ResNet18 | VGG16 | WRN-16-10 | ResNet18 | VGG16 | WRN-16-10 | ResNet18 |
| Adam [18] | 94.84$^\diamond$ | 91.08 | 93.54 | 76.88$^\diamond$ | 64.07 | 74.81 | 66.54$^*$ |
| Adabound [63] | 92.56 | 91.35 | 91.68 | 71.43 | 64.74 | 71.64 | 68.13$^\dagger$ |
| RAdam [23] | 93.85 | 90.84 | 94.16 | 74.30 | 63.99 | 75.92 | 67.62$^*$ |
| AdamW [64] | 94.95 | 90.75 | 95.95 | 77.30 | 63.40 | 79.63 | 67.93$^\dagger$ |
| AdaBelief [62] | 95.20$^\ddagger$ | 92.25 | 95.71 | 77.02$^\ddagger$ | 68.63 | 77.93 | 70.08$^\ddagger$ |
| Stagewise SGD [20] | 95.23±0.19$^\diamond$ | 92.13±0.02 | 95.51±0.02 | 78.24±0.18$^\diamond$ | 69.97±0.02 | 78.95±0.03 | 70.23$^\dagger$ |
| SLA [1] | 95.27±0.06$^\diamond$ | 92.38±0.02 | 95.73±0.02 | 78.34±0.05$^\diamond$ | 70.20±0.04 | 79.54±0.02 | 70.30±0.09 |
| SLRLA | **95.47**±0.20 | **92.63**±0.03 | **96.08**±0.07 | **78.58**±0.15 | **70.63**±0.02 | **79.85**±0.05 | **70.47**±0.12 |

## 6.1 Results on Strongly Convex Problems

Here we investigate the effects of $\alpha$ on the performance of lookahead, stagewise lookahead [1] (SLA) and SLRLA on a regularized softmax problem with MNIST [56]. The regularized softmax problem is strongly convex, as its regularization constant is set to $5 \times 10^{-6}$. Following our theory, we use a linearly decayed learning rate (LR) for lookahead, and multi-step decayed LRs for SLA/SLRLA. See more details in Appendix B. Fig. 1 (a) shows that when $\alpha$ increases, for lookahead & SLA, (i) their training loss reflecting the optimization error $\varepsilon_{\mathrm{opt}}$ decreases faster; (ii) the distance between their training and test losses reflecting the generalization error $\varepsilon_{\mathrm{gen}}$ becomes larger; (iii) their test loss first decreases and then increases which indicates the balance of $\alpha$. Fig. 1 (b) shows the balance impact of $\alpha$ on test accuracy of SLRLA, and also testifies the superiority of SLRLA over lookahead and SLA.

## 6.2 Results on Nonconvex Problems

**Assumption Investigation.** We investigate the key assumption, PŁ assumption on the nonconvex problems, for networks. We train ResNet18 [4] and wide-ReseNet-16-10 (WRN) [57] via SLA/SLARA, and report $\mu \triangleq \|\nabla F_{\mathcal{S}}(\boldsymbol{\theta}_t)\|^2/[2(F_{\mathcal{S}}(\boldsymbol{\theta}_t) - F_{\mathcal{S}}(\boldsymbol{\theta}^*))]$. We estimate the optimum $\boldsymbol{\theta}^*$ of $F_{\mathcal{S}}(\boldsymbol{\theta})$ as the solution found by SLA/SLARA after 200 epochs which gives a small objective value ($\approx 10^{-3}$) and is almost an optimum. Fig. 2 shows that $\mu$ in SLA & SLARA is larger than $5 \times 10^{-4}$. So on networks, PŁ condition holds at least along the optimization trajectory. This accords with the observations/theories in [30–34].

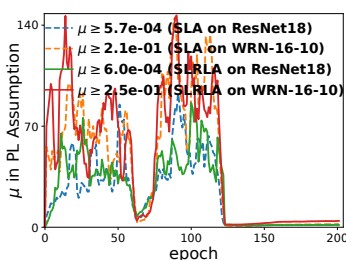

Figure 2: Investigation of PŁ Assp.

**Classification Results.** We evaluate SLA and SLRLA on CIFAR10/100 [58] and ImageNet [59] using different network architectures, i.e. ResNet18 [4], VGG16 [60] and WRN-16-10 [57]. For all experiments, SLRLA and SLA set $k = 5$, a momentum of $0.9$, and a multi-stage learning rate (LR) decay at the $\{0.3S, 0.6S, 0.8S\}$-th epoch with total epoch number $S$. On CIFAR10/100, we train 200 epochs with $\alpha = 0.8$, a weight decay of $10^{-3}$, and set LR decay rate as $0.2$. On Imagenet, we run 100 epochs using $\alpha = 0.5$, a weight decay of $10^{-4}$ and an LR decay rate of $0.1$. These settings follow [1, 61, 62]. For regularization constant $\beta_q$, SLRLA selects it from $\{0.02, 0.2, 2.0, 20\}$ via cross validation, and finally sets it as $0.2$ on CIFAR10/100 and $20$ on ImageNet.

Table 1 reports the average accuracy and variance of 5 random seeds. SLRLA achieves the highest accuracy on CIFAR10/100 and ImageNet. This because (i) our theories in Sec. 5 show the advantages of SLRLA over SGD/SLA to achieve small excess risk error, indicating better test performance; (ii) as observed/proved in [22, 37–41], compared with adaptive algorithms, e.g. Adam and its variants, SGD-alike algorithms, e.g. SLA/SLARA, often converge to flatter minima and thus generalize better. The results in Appendix B show the stable performance of SLARA on ImageNet when tuning the regularization constant $\beta_q$ in a large range, testifying the robustness of SLARA.

## 7 Conclusion

In this work, for the first time we theoretically show the advantages of lookahead in terms of the excess risk error, explaining its better test performance than its vanilla inner optimizer. Moreover, we prove that the stagewise optimization strategy can benefit lookahead in improving its excess risk error. Finally, we propose SLRLA which locally regularizes the vanilla objective to further improve the excess risk error of stagewise lookahead. Experimental results validated the advantages of SLRLA.

## Acknowledgements

The authors sincerely thank the anonymous reviewers for their constructive comments on this work.

**1. Funding.** Pan Zhou, Hanshu Yan, Jiashi Feng and Shuicheng Yan are supported by Sea AI Lab, mainly for their GPU resource support. Xiao-Tong Yuan is supported in part by the National Key Research and Development Program of China under Grant No. 2018AAA0100400 and in part by the Natural Science Foundation of China (NSFC) under Grant No.61876090 and No.61936005.

**2. Competing Interests.** Pan Zhou, Jiashi Feng and Shuicheng Yan are staffs in Sea AI Lab. Hanshu Yan is an intern in Sea AI Lab, and is also a Ph.D. student in National University of Singapore. Xiao-Tong Yuan works as a professor in Nanjing University of Information Science & Technology, Nanjing, China.

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
