# Towards Understanding Why Lookahead Generalizes Better Than SGD and Beyond
# (Supplementary File)

**Pan Zhou**[*]    **Hanshu Yan**[*]    **Xiao-Tong Yuan**[†]    **Jiashi Feng**[*]    **Shuicheng Yan**[*]
[*] Sea AI Lab, Singapore
[†] Nanjing University of Information Science & Technology, Nanjing, China
{zhoupan, yanhanshu, fengjs, yansc}@sea.com   xtyuan@nuist.edu.cn

This supplementary document contains the technical proofs of convergence results and some additional experimental results of the NeurIPS'21 submission entitled "Towards Understanding Why Lookahead Generalizes Better Than SGD and Beyond". It is structured as follows. Appendix A first discusses the limitations of this work. In Appendix B, we provides more experimental results and details, including the robustness investigation of SLRLA to regularization parameter. Next, Appendix C presents several auxiliary lemmas which will be used for subsequent analysis. Then Appendix D gives the proofs of the main results in Sec. 4, including Theorem $1 \sim 4$ and Corollary 1 which analyze optimization error, generalization error and excess risk error of vanilla lookahead algorithm. Finally, Appendix E provides the proofs of the results in Sec. 5, including Theorems 5 and 6 which analyze the optimization error, generalization error and excess risk error of the proposed SLRLA.

## A   Limitation Discussion

The main limitation of this work is that the analysis in this work cannot be applicable to general nonconvex problems. This is because as explained in Sec. 4.3, for general nonconvex problems, one often uses the gradient norm $\mathbb{E}[\|\nabla F_{\mathcal{S}}(\boldsymbol{\theta})\|^2]$ instead of the loss distance $\mathbb{E}[F_{\mathcal{S}}(\boldsymbol{\theta}) - F_{\mathcal{S}}(\boldsymbol{\theta}_{\mathcal{S}}^*)]$ to measure whether $\boldsymbol{\theta}$ is a stationary point. The reasons is that many stationary points may exist in a nonconvex problem. But as shown in Sec. 3, to bound the excess risk error, one needs to first bound the loss distance. In this way, our analysis cannot be applicable to general nonconvex problems. However, in this work, we are particularly interested in nonconvex problems under PL condition which allows us to bound the loss distance, since PL condition establishes the relation between gradient norm and loss distance. Moreover, deep learning models often satisfy PL condition, which is well observed/proved in [1, 2, 3, 4, 5] and our empirical results in Sec. 6.2. So in the future, finding and developing new framework which avoids the requirement on the bounded loss distance to analyze excess risk error for nonconvex problems is an interesting avenue of future research.

## B   More Experimental Results and Details

Due to space limitation, we defer more experimental results and details to this appendix. Here we first investigate robustness of SLRLA to the regularization parameter $\beta_q$. Then we present more experimental setting details on strongly convex problems in Sec. 6.1. Note, we use two A100 GPUs to train ImageNet, and use single A100 GPU for all remaining experiments. Our codes are implemented based on lookahead.

**Robustness to Regularization Parameter.** Here we investigate the impact of regularization parameter $\beta_q$ to the performance of SLRLA. For simplicity, we set $\beta_q = \beta$ as constant for all optimization stages. Then we evaluate SLRLA on ImagNet using ResNet 18. We respectively set

r

Figure 4: Effects of the regularization parameter $\beta$ to SLRLA.

$\beta = 0.2, 0.8, 2, 5, 20, 50, 100$ and train ResNet 18 for 200 epochs with the same training strategy in the manuscript for fairness. Fig. 4 reports the effects of regularization parameter $\beta$ to the performance of SLRLA. From Fig. 4, one can observe that when tuning $\beta$ in a relatively large range, SLRLA has relatively stable performance on ImageNet. This testifies the robustness of ImageNet to regularization parameter.

**Details of Experimental Setting on Strongly Convex Problems.** Here we introduce the details of the experiments in Sec. 6.1. We test lookahead, SLA and SLRLA on a regularized softmax problem with MNIST [6]. The regularized softmax problem for $k$-classification task can be formulated as

$$\min_{\boldsymbol{\theta}} \frac{1}{n} \sum_{i=1}^{n} \sum_{j=1}^{k} \left[ \frac{\gamma}{2} \|\boldsymbol{\theta}_j\|^2 - \mathbf{1}\{\boldsymbol{y}_i = j\} \log \frac{\exp(\boldsymbol{\theta}_j^\top \boldsymbol{x}_i)}{\sum_{l=1}^{k} \exp(\boldsymbol{\theta}_l^\top \boldsymbol{x}_i)} \right]$$

where $\boldsymbol{y}_i$ is the target output of the $i$-th sample $\boldsymbol{x}_i$. We set the regularization constant $\gamma = 10^{-5}$. Following our theory, we use a linearly decayed learning rate (LR) for lookahead, namely $\eta_t = \frac{c}{t}$ with a constant $c$. For SLA/SLRLA, we use multi-step decayed LRs with decaying rate 0.5. That is, we decay the LR at the $\{0.3S, 0.6S, 0.8S\}$-th epoch where $S$ denotes the total epoch number. We tune the initial LR of lookahead as 0.1 and set the initial LR of SLA/SLRLA as 0.05. For lookahead, SLA and SLRLA, we set $k = 5$, a momentum of 0.9, a weight decay of $10^{-4}$, and training epoch number $S = 80$.

# C  Auxiliary Lemmas

**Lemma 1.** [7] *Assume that* $\ell(f(\boldsymbol{x}; \boldsymbol{\theta}), \boldsymbol{y})$ *is* $L$-*smooth w.r.t.* $\boldsymbol{\theta}$. *Suppose* $\boldsymbol{v}_{\tau+1}^{(t)} = \boldsymbol{v}_{\tau}^{(t)} - \eta_{\tau}^{(t)} \nabla \ell(f(\boldsymbol{x}; \boldsymbol{v}_{\tau}^{(t)}); \boldsymbol{y})$ *and* $\tilde{\boldsymbol{v}}_{\tau+1}^{(t)} = \tilde{\boldsymbol{v}}_{\tau}^{(t)} - \eta_{\tau}^{(t)} \nabla \ell(f(\boldsymbol{x}; \tilde{\boldsymbol{v}}_{\tau}^{(t)}); \boldsymbol{y})$. *Assume* $\max(\|\tilde{\boldsymbol{v}}_{\tau+1}^{(t)} - \tilde{\boldsymbol{v}}_{\tau}^{(t)}\|, \|\boldsymbol{v}_{\tau+1}^{(t)} - \boldsymbol{v}_{\tau}^{(t)}\|) \leq \eta_{\tau}^{(t)} G$.
*(1) Suppose* $\ell(f(\boldsymbol{x}; \boldsymbol{\theta}), \boldsymbol{y})$ *is nonconvex w.r.t.* $\boldsymbol{\theta}$.
*(1.1) For the* $(t, \tau)$ *iteration, when* $\boldsymbol{v}_{\tau+1}^{(t)}$ *and* $\tilde{\boldsymbol{v}}_{\tau+1}^{(t)}$ *sample the same sample* $(\boldsymbol{x}, \boldsymbol{y})$, *we have*

$$\|\boldsymbol{v}_{\tau+1}^{(t)} - \tilde{\boldsymbol{v}}_{\tau+1}^{(t)}\|_2 \leq (1 + \eta_{\tau}^{(t)} L) \|\boldsymbol{v}_{\tau}^{(t)} - \tilde{\boldsymbol{v}}_{\tau}^{(t)}\|_2.$$

*(1.2) Assume that* $\ell(f(\boldsymbol{x}; \boldsymbol{\theta}), \boldsymbol{y})$ *is further* $G$-*Lipschitz w.r.t.* $\boldsymbol{\theta}$. *For the* $(t, \tau)$ *iteration, when* $\boldsymbol{v}_{\tau+1}^{(t)}$ *and* $\tilde{\boldsymbol{v}}_{\tau+1}^{(t)}$ *sample different samples* $(\boldsymbol{x}, \boldsymbol{y})$ *and* $(\boldsymbol{x}', \boldsymbol{y}')$, *we have*

$$\|\boldsymbol{v}_{\tau+1}^{(t)} - \tilde{\boldsymbol{v}}_{\tau+1}^{(t)}\|_2 \leq \|\boldsymbol{v}_{\tau}^{(t)} - \tilde{\boldsymbol{v}}_{\tau}^{(t)}\|_2 + 2\eta_{\tau}^{(t)} G.$$

*(2) Suppose* $\ell(f(\boldsymbol{x}; \boldsymbol{\theta}), \boldsymbol{y})$ *is convex w.r.t.* $\boldsymbol{\theta}$. *Set the learning rate* $\eta_{\tau}^{(t)} \leq \frac{2}{L}$.
*(2.1) For the* $(t, \tau)$ *iteration, when* $\boldsymbol{v}_{\tau+1}^{(t)}$ *and* $\tilde{\boldsymbol{v}}_{\tau+1}^{(t)}$ *sample the same sample* $(\boldsymbol{x}, \boldsymbol{y})$, *we have*

$$\|\boldsymbol{v}_{\tau+1}^{(t)} - \tilde{\boldsymbol{v}}_{\tau+1}^{(t)}\|_2 \leq \|\boldsymbol{v}_{\tau}^{(t)} - \tilde{\boldsymbol{v}}_{\tau}^{(t)}\|_2.$$

*(2.2) Assume that* $\ell(f(\boldsymbol{x}; \boldsymbol{\theta}), \boldsymbol{y})$ *is further* $G$-*Lipschitz w.r.t.* $\boldsymbol{\theta}$. *For the* $(t, \tau)$ *iteration, when* $\boldsymbol{v}_{\tau+1}^{(t)}$ *and* $\tilde{\boldsymbol{v}}_{\tau+1}^{(t)}$ *sample different samples* $(\boldsymbol{x}, \boldsymbol{y})$ *and* $(\boldsymbol{x}', \boldsymbol{y}')$, *we have*

$$\|\boldsymbol{v}_{\tau+1}^{(t)} - \tilde{\boldsymbol{v}}_{\tau+1}^{(t)}\|_2 \leq \|\boldsymbol{v}_{\tau}^{(t)} - \tilde{\boldsymbol{v}}_{\tau}^{(t)}\|_2 + 2\eta_{\tau}^{(t)} G.$$

*(3) Suppose $\ell(f(\boldsymbol{x}; \boldsymbol{\theta}), \boldsymbol{y})$ is $\lambda$-strongly convex w.r.t. $\boldsymbol{\theta}$. Set the learning rate $\eta_\tau^{(t)} \le \frac{2}{\lambda + L}$.*
*(3.1)For the $(t, \tau)$ iteration, when $\boldsymbol{v}_{\tau+1}^{(t)}$ and $\tilde{\boldsymbol{v}}_{\tau+1}^{(t)}$ sample the same sample $(\boldsymbol{x}, \boldsymbol{y})$, we have*

$$\|\boldsymbol{v}_{\tau+1}^{(t)} - \tilde{\boldsymbol{v}}_{\tau+1}^{(t)}\|_2 \le \left(1 - \frac{\eta_\tau^{(t)}\lambda L}{\lambda + L}\right)\|\boldsymbol{v}_\tau^{(t)} - \tilde{\boldsymbol{v}}_\tau^{(t)}\|_2.$$

*(3.2) Assume that $\ell(f(\boldsymbol{x}; \boldsymbol{\theta}), \boldsymbol{y})$ is further G-Lipschitz w.r.t. $\boldsymbol{\theta}$. For the $(t, \tau)$ iteration, when $\boldsymbol{v}_{\tau+1}^{(t)}$ and $\tilde{\boldsymbol{v}}_{\tau+1}^{(t)}$ sample different samples $(\boldsymbol{x}, \boldsymbol{y})$ and $(\boldsymbol{x}', \boldsymbol{y}')$, we have*

$$\|\boldsymbol{v}_{\tau+1}^{(t)} - \tilde{\boldsymbol{v}}_{\tau+1}^{(t)}\|_2 \le \left(1 - \frac{\eta_\tau^{(t)}\lambda L}{\lambda + L}\right)\|\boldsymbol{v}_\tau^{(t)} - \tilde{\boldsymbol{v}}_\tau^{(t)}\|_2 + 2\eta_\tau^{(t)}G.$$

**Lemma 2.** *[8] If $F_{\mathcal{S}}(\boldsymbol{\theta})$ satisfies the PL condition, then for any $\boldsymbol{\theta}$ we have*

$$\|\boldsymbol{\theta} - \boldsymbol{\theta}_{\mathcal{S}}^*\|^2 \le \frac{1}{2\mu}(F_{\mathcal{S}}(\boldsymbol{\theta}) - F_{\mathcal{S}}(\boldsymbol{\theta}_{\mathcal{S}}^*)),$$

*where $\boldsymbol{\theta}_{\mathcal{S}}^* = \arg\min_{\boldsymbol{\theta}} F_{\mathcal{S}}(\boldsymbol{\theta})$.*

**Lemma 3.** *Assume that $\ell(f(\boldsymbol{x}; \boldsymbol{\theta}); \boldsymbol{y})$ satisfies $|\ell(f(\boldsymbol{x}; \boldsymbol{\theta}); \boldsymbol{y})| \le \ell_{max}$ and is G-Lipschitz for all $(\boldsymbol{x}, \boldsymbol{y})$. Let $\mathcal{S}$ and $\mathcal{S}'$ be two datasets of size $n$ differing in only one single sample. Denote by $\boldsymbol{v}_\tau^{(t)}$ and $\tilde{\boldsymbol{v}}_\tau^{(t)}$ the output of the $(t, \tau)$-iteration of lookahead on the datasets $\mathcal{S}$ and $\mathcal{S}'$, respectively. Then for every $(\boldsymbol{x}, \boldsymbol{y}) \in \mathcal{S}$ and every $t_0$ under the random update rule and the random permutation rule, we have*

$$\mathbb{E}|\ell(f(\boldsymbol{x}; \boldsymbol{u}_\tau^{(t)}); \boldsymbol{y}) - \ell(f(\boldsymbol{x}; \widetilde{\boldsymbol{u}}_\tau^{(t)}); \boldsymbol{y})| \le \frac{t_0\ell_{max}}{n} + L\mathbb{E}[\boldsymbol{\delta}_\tau^{(t)} \mid \boldsymbol{\delta}_{\tau_0}^{(t_0)} = 0],$$

*where $\boldsymbol{\delta}_\tau^{(t)} = \|\boldsymbol{u}_\tau^{(t)} - \widetilde{\boldsymbol{u}}_\tau^{(t)}\|_2\|$ with $\boldsymbol{u}_\tau^{(t)} = \alpha\boldsymbol{v}_\tau^{(t)} + (1-\alpha)\boldsymbol{\theta}_{t-1}$ and $\widetilde{\boldsymbol{u}}_\tau^{(t)} = \alpha\tilde{\boldsymbol{v}}_\tau^{(t)} + (1-\alpha)\widetilde{\boldsymbol{\theta}}_{t-1}$.*

*Proof.* This proof follows [7]. For completeness, we provide its proof here. To begin with, we first define the variable $\boldsymbol{u}_\tau^{(t)} = \alpha\boldsymbol{v}_\tau^{(t)} + (1-\alpha)\boldsymbol{\theta}_{t-1} = \alpha\boldsymbol{v}_\tau^{(t)} + (1-\alpha)\boldsymbol{v}_0^{(t)}$. It can be observed that when $\tau = k$, then $\boldsymbol{\theta}_t = \boldsymbol{u}_k^{(t)}$. In this way, we also can obtain the updating rule of $\boldsymbol{u}_{\tau+1}^{(t)}$ as follows:

$$\boldsymbol{u}_{\tau+1}^{(t)} = \alpha\boldsymbol{v}_\tau^{(t)} + (1-\alpha)\boldsymbol{v}_0^{(t)} = \boldsymbol{u}_\tau^{(t)} - \alpha\eta_\tau^{(t)}\boldsymbol{g}_\tau^{(t)},$$

where $\boldsymbol{g}_\tau^{(t)}$ denotes the stochastic gradient at the point $\boldsymbol{v}_\tau^{(t)}$.

Suppose given $n$ samples $\mathcal{S} = \{\boldsymbol{z}_1, \boldsymbol{z}_2, \cdots, \boldsymbol{z}_n\}$ where $\boldsymbol{z}_i = (\boldsymbol{x}_i, \boldsymbol{y}_i)$ is sampled from an unknown distribution $\mathcal{D}$, one usually analyze the stability of an algorithm by replacing one sample in $\mathcal{S}$ by another sample from $\mathcal{D}$. Suppose the generated sample set $\mathcal{S}^{(i)} = \{\boldsymbol{z}_1', \boldsymbol{z}_2', \cdots, \boldsymbol{z}_n'\} = \{\boldsymbol{z}_1, \boldsymbol{z}_2, \cdots, \boldsymbol{z}_{i-1}, \boldsymbol{z}_i', \boldsymbol{z}_{i+1}\cdots, \boldsymbol{z}_n\}$ which only differs from the set $\mathcal{S}$ with the $i$-th sample. Then based on these two set, one can train the algorithm to obtain different solution $\boldsymbol{\theta}$ of the function $F_{\mathcal{S}}(\boldsymbol{\theta})$. When using $\mathcal{S}^{(i)}$, we use $\widetilde{\boldsymbol{\theta}}_t$ and $\tilde{\boldsymbol{v}}_\tau^{(t)}$ to denote their corresponding versions $\boldsymbol{\theta}_t$ and $\tilde{\boldsymbol{v}}_\tau^{(t)}$ in Algorithm 1 trained on $\mathcal{S}$. In this way, we can define their corresponding $\boldsymbol{u}_\tau^{(t)}$ and $\widetilde{\boldsymbol{u}}_\tau^{(t)}$. Next, we can define

$$\boldsymbol{\delta}_\tau^{(t)} = \|\boldsymbol{u}_\tau^{(t)} - \widetilde{\boldsymbol{u}}_\tau^{(t)}\|_2.$$

In this way, we can follow [7] and prove our results. Let $\mathcal{E} = \mathbf{1}[\boldsymbol{\delta}_{\tau_0}^{(t_0)} = 0]$ denote the event that $\boldsymbol{\delta}_{\tau_0}^{(t_0)}$. In this way, we can upper bound

$$\begin{aligned}
\mathbb{E}|\ell(f(\boldsymbol{x}; \boldsymbol{u}_\tau^{(t)}); \boldsymbol{y}) - \ell(f(\boldsymbol{x}; \widetilde{\boldsymbol{u}}_\tau^{(t)}); \boldsymbol{y})| =& \mathbb{P}(\mathcal{E})\mathbb{E}[|\ell(f(\boldsymbol{x}; \boldsymbol{u}_\tau^{(t)}); \boldsymbol{y}) - \ell(f(\boldsymbol{x}; \widetilde{\boldsymbol{u}}_\tau^{(t)}); \boldsymbol{y})| \mid \mathcal{E}] \\
&+ \mathbb{P}(\mathcal{E}^c)\mathbb{E}[|\ell(f(\boldsymbol{x}; \boldsymbol{u}_\tau^{(t)}); \boldsymbol{y}) - \ell(f(\boldsymbol{x}; \widetilde{\boldsymbol{u}}_\tau^{(t)}); \boldsymbol{y})| \mid \mathcal{E}^c] \\
\le& \mathbb{E}[|\ell(f(\boldsymbol{x}; \boldsymbol{u}_\tau^{(t)}); \boldsymbol{y}) - \ell(f(\boldsymbol{x}; \widetilde{\boldsymbol{u}}_\tau^{(t)}); \boldsymbol{y})| \mid \mathcal{E}] + 2\ell_{max}\mathbb{P}(\mathcal{E}^c) \\
\le& G\mathbb{E}[\|\boldsymbol{u}_\tau^{(t)} - \widetilde{\boldsymbol{u}}_\tau^{(t)}\|_2 \mid \mathcal{E}] + 2\ell_{max}\mathbb{P}(\mathcal{E}^c)
\end{aligned}$$

Then we only need to bound $\mathbb{P}(\mathcal{E}^c)$. Assume $t^*k + \tau^*$ denote the position in which $\mathcal{S}$ and $\mathcal{S}'$ differ and consider the random variable $(i, j)$ assuming the index of the first time step in which the lookahead

algorithm uses the sample $(\boldsymbol{x}_{t^*k+\tau^*}, \boldsymbol{y}_{t^*k+\tau^*})$. When $ik+j > t^*k+\tau^*$, then we have $\boldsymbol{\delta}_j^{(i)} = 0$ since the execution on $\mathcal{S}$ and $\mathcal{S}'$ is identical until iteration $(t_0, \tau_0)$. In this way, we have

$$\mathbb{P}(\mathcal{E}^c) \leq \mathbb{P}(\boldsymbol{\delta}_{\tau_0}^{(t_0)} \neq 0) \leq \mathbb{P}(ik+j \leq t_0 k + \tau_0).$$

Under the random permutation rule, $ik+j)$ is a uniformly random number in $\{1, \cdots, n\}$ and therefore,

$$\mathbb{P}(ik+j \leq t_0 k + \tau_0) = \frac{t_0 k + \tau_0}{n}.$$

This completes the proof. □

# D   Proof of The Results in Sec. 4

## D.1   Proof of Theorem 1

*Proof.* Since the function $F_{\mathcal{S}}(\boldsymbol{\theta})$ is $\lambda$-strongly-convex and $\boldsymbol{\theta}_{\mathcal{S}}^*$ is the optimum of $F_{\mathcal{S}}(\boldsymbol{\theta})$, then we have

$$F_{\mathcal{S}}(\boldsymbol{v}_{\tau-1}^{(t)}) \geq F_{\mathcal{S}}(\boldsymbol{\theta}_{\mathcal{S}}^*) + \langle \nabla F_{\mathcal{S}}(\boldsymbol{\theta}_{\mathcal{S}}^*), \boldsymbol{\theta}_{\mathcal{S}}^* - \boldsymbol{v}_{\tau-1}^{(t)} \rangle + \frac{\lambda}{2}\|\boldsymbol{\theta}_{\mathcal{S}}^* - \boldsymbol{v}_{\tau-1}^{(t)}\|_2^2 = F_{\mathcal{S}}(\boldsymbol{\theta}_{\mathcal{S}}^*) + \frac{\lambda}{2}\|\boldsymbol{\theta}_{\mathcal{S}}^* - \boldsymbol{v}_{\tau-1}^{(t)}\|_2^2.$$

Similarly, we have

$$F_{\mathcal{S}}(\boldsymbol{\theta}_{\mathcal{S}}^*) \geq F_{\mathcal{S}}(\boldsymbol{v}_{\tau-1}^{(t)}) + \langle \nabla F_{\mathcal{S}}(\boldsymbol{v}_{\tau-1}^{(t)}), \boldsymbol{\theta}_{\mathcal{S}}^* - \boldsymbol{v}_{\tau-1}^{(t)} \rangle + \frac{\lambda}{2}\|\boldsymbol{\theta}_{\mathcal{S}}^* - \boldsymbol{v}_{\tau-1}^{(t)}\|_2^2.$$

Then we can upper bound

$$\begin{aligned}
\mathbb{E}\left[\|\boldsymbol{v}_{\tau}^{(t)} - \boldsymbol{\theta}_{\mathcal{S}}^*\|^2\right] &\leq \mathbb{E}\left[\|\boldsymbol{v}_{\tau-1}^{(t)} - \eta_{\tau-1}^{(t)}\boldsymbol{g}_{\tau-1}^{(t)} - \boldsymbol{\theta}_{\mathcal{S}}^*\|^2\right]\\
&\leq \mathbb{E}\left[\|\boldsymbol{v}_{\tau-1}^{(t)} - \boldsymbol{\theta}_{\mathcal{S}}^*\|^2 - 2\eta_{\tau-1}^{(t)}\langle \boldsymbol{v}_{\tau-1}^{(t)} - \boldsymbol{\theta}_{\mathcal{S}}^*, \boldsymbol{g}_{\tau-1}^{(t)}\rangle + (\eta_{\tau-1}^{(t)})^2\|\boldsymbol{g}_{\tau-1}^{(t)}\|^2\right]\\
&\overset{①}{=} \mathbb{E}\left[\|\boldsymbol{v}_{\tau-1}^{(t)} - \boldsymbol{\theta}_{\mathcal{S}}^*\|^2 - 2\eta_{\tau-1}^{(t)}\langle \boldsymbol{v}_{\tau-1}^{(t)} - \boldsymbol{\theta}_{\mathcal{S}}^*, \nabla F_{\mathcal{S}}(\boldsymbol{v}_{\tau-1}^{(t)})\rangle + (\eta_{\tau-1}^{(t)})^2\|\boldsymbol{g}_{\tau-1}^{(t)}\|^2\right]\\
&\leq \mathbb{E}\left[\|\boldsymbol{v}_{\tau-1}^{(t)} - \boldsymbol{\theta}_{\mathcal{S}}^*\|^2 + 2\eta_{\tau-1}^{(t)}\left[F_{\mathcal{S}}(\boldsymbol{\theta}_{\mathcal{S}}^*) - F_{\mathcal{S}}(\boldsymbol{v}_{\tau-1}^{(t)}) - \frac{\lambda}{2}\|\boldsymbol{\theta}_{\mathcal{S}}^* - \boldsymbol{v}_{\tau-1}^{(t)}\|_2^2\right] + (\eta_{\tau-1}^{(t)})^2\|\boldsymbol{g}_{\tau-1}^{(t)}\|^2\right]\\
&\leq \mathbb{E}\left[\|\boldsymbol{v}_{\tau-1}^{(t)} - \boldsymbol{\theta}_{\mathcal{S}}^*\|^2 + 2\eta_{\tau-1}^{(t)}\left[-\frac{\lambda}{2}\|\boldsymbol{\theta}_{\mathcal{S}}^* - \boldsymbol{v}_{\tau-1}^{(t)}\|_2^2 - \frac{\lambda}{2}\|\boldsymbol{\theta}_{\mathcal{S}}^* - \boldsymbol{v}_{\tau-1}^{(t)}\|_2^2\right] + (\eta_{\tau-1}^{(t)})^2\|\boldsymbol{g}_{\tau-1}^{(t)}\|^2\right]\\
&\overset{②}{\leq} (1 - 2\lambda\eta_{\tau-1}^{(t)})\mathbb{E}\left[\|\boldsymbol{v}_{\tau-1}^{(t)} - \boldsymbol{\theta}_{\mathcal{S}}^*\|^2\right] + (\eta_{\tau-1}^{(t)})^2 G^2
\end{aligned}$$

where ① holds since $\mathbb{E}[\boldsymbol{g}_{\tau-1}^{(t)}] = \nabla F_{\mathcal{S}}(\boldsymbol{v}_{\tau-1}^{(t)})$, ② holds by assuming each individual loss is $G$-Lipschitz. Since $\eta_{\tau}^{(t)} = \frac{c}{\lambda((t-1)k+\tau+2)}$ where $c = \frac{\lambda+L}{L} \in (1,2]$, we have $\frac{1}{\lambda((t-1)k+\tau+2)} \leq \eta_{\tau}^{(t)} \leq \frac{2}{\lambda((t-1)k+\tau+2)}$. For brevity, let

$$A(j,i) = \prod_{s=j}^{i}\left(1 - \frac{2}{(t-1)k+s+1}\right), \qquad B(j,i) = \sum_{s=j}^{i}\left(1 - \frac{2}{(t-1)k+s+1}\right).$$

We can unwind the above recurrence relation from $\tau = k$ to 1 to obtain

$$
\begin{aligned}
\mathbb{E}\left[\|\boldsymbol{v}_\tau^{(t)} - \boldsymbol{\theta}_\mathcal{S}^*\|^2\right] \leq & A(1,k)\mathbb{E}\left[\|\boldsymbol{v}_0^{(t)} - \boldsymbol{\theta}_\mathcal{S}^*\|^2\right] + \sum_{i=1}^{k} A(i+1,k) \cdot \frac{4G^2}{\lambda^2((t-1)k+i+1)^2} \\
\overset{\text{①}}{\leq} & \exp\left\{-B(1,k)\right\}\mathbb{E}\left[\|\boldsymbol{v}_0^{(t)} - \boldsymbol{\theta}_\mathcal{S}^*\|^2\right] + \sum_{i=1}^{k} \exp\left\{-B(i+1,k)\right\}\frac{4G^2}{\lambda^2((t-1)k+i+1)^2} \\
\overset{\text{②}}{\leq} & \exp\left\{-2\log\left(\frac{tk+2}{(t-1)k+2}\right)\right\}\mathbb{E}\left[\|\boldsymbol{v}_0^{(t)} - \boldsymbol{\theta}_\mathcal{S}^*\|^2\right] \\
& + \sum_{i=1}^{k} \exp\left\{-2\log\left(\frac{tk+2}{(t-1)k+i+2}\right)\right\}\frac{4G^2}{\lambda^2((t-1)k+i+1)^2} \\
= & \left(\frac{(t-1)k+2}{tk+2}\right)^2 \mathbb{E}\left[\|\boldsymbol{v}_0^{(t)} - \boldsymbol{\theta}_\mathcal{S}^*\|^2\right] + \frac{4G^2}{\lambda^2(tk+2)^2}\sum_{i=1}^{k} \frac{((t-1)k+i+2)^2}{((t-1)k+i+1)^2} \\
\overset{\text{③}}{\leq} & \left(\frac{(t-1)k+2}{tk+2}\right)^2 \mathbb{E}\left[\|\boldsymbol{v}_0^{(t)} - \boldsymbol{\theta}_\mathcal{S}^*\|^2\right] + \frac{16kG^2}{\lambda^2(tk+2)^2} \\
\overset{\text{④}}{=} & \left(\frac{(t-1)k+2}{tk+2}\right)^2 \mathbb{E}\left[\|\boldsymbol{\theta}_{t-1} - \boldsymbol{\theta}_\mathcal{S}^*\|^2\right] + \frac{16kG^2}{\lambda^2(tk+2)^2},
\end{aligned}
$$

where in ① we have used $1+x \leq e^x$, in ② we have used $\sum_{j=i}^{k} \frac{1}{a+j+1} \geq \int_i^{k+1} \frac{1}{a+s+1} ds = \log\left(\frac{a+k+2}{a+i+1}\right)$ and ③ is due to $(a+1)^2 \leq 4a^2$ for all $a \geq 1$ and ④ is due to $w^{(t-1)} = v_0^{(t)}$. Then it follows from the updating of slow parameter and the above inequalities that

$$
\begin{aligned}
\mathbb{E}\left[\|\boldsymbol{\theta}_t - \boldsymbol{\theta}_\mathcal{S}^*\|^2\right] \leq & (1-\alpha)\mathbb{E}\left[\|\boldsymbol{\theta}_{t-1} - \boldsymbol{\theta}_\mathcal{S}^*\|^2\right] + \alpha\mathbb{E}\left[\|\boldsymbol{v}_k^{(t)} - \boldsymbol{\theta}_\mathcal{S}^*\|^2\right] \\
\leq & \left(1 - \alpha + \alpha\left(\frac{(t-1)k+2}{tk+2}\right)^2\right)\mathbb{E}\left[\|\boldsymbol{\theta}_{t-1} - \boldsymbol{\theta}_\mathcal{S}^*\|^2\right] + \frac{16\alpha kG^2}{\lambda^2(tk+2)^2} \\
= & \left(1 - \frac{2\alpha k}{tk+2} + \alpha\left(\frac{k}{tk+2}\right)^2\right)\mathbb{E}\left[\|\boldsymbol{\theta}_{t-1} - \boldsymbol{\theta}_\mathcal{S}^*\|^2\right] + \frac{16\alpha kG^2}{\lambda^2(tk+2)^2}.
\end{aligned}
$$

Unwinding this recurrence relation from time instance $t$ to 1 yields

$$
\begin{aligned}
& \mathbb{E}\left[\|\boldsymbol{\theta}_t - \boldsymbol{\theta}_\mathcal{S}^*\|^2\right] \\
\leq & \prod_{i=1}^{t}\left(1 - \frac{2\alpha k}{ik+2} + \alpha\left(\frac{k}{ik+2}\right)^2\right)\|\boldsymbol{\theta}_0 - \boldsymbol{\theta}_\mathcal{S}^*\|^2 + \sum_{i=1}^{t}\prod_{j=i+1}^{t}\left(1 - \frac{2\alpha k}{jk+2} + \alpha\left(\frac{k}{jk+2}\right)^2\right)\frac{16\alpha kG^2}{\lambda^2(ik+2)^2} \\
\leq & \prod_{i=1}^{t}\exp\left\{-\frac{2\alpha k}{ik+2} + \alpha\left(\frac{k}{ik+2}\right)^2\right\}\|\boldsymbol{\theta}_0 - \boldsymbol{\theta}_\mathcal{S}^*\|^2 + \sum_{i=1}^{t}\prod_{j=i+1}^{t}\exp\left\{-\frac{2\alpha k}{jk+2} + \alpha\left(\frac{k}{jk+2}\right)^2\right\}\frac{16\alpha kG^2}{\lambda^2(ik+2)^2} \\
= & \exp\left\{-\sum_{i=1}^{t}\left(\frac{2\alpha k}{ik+2} - \alpha\left(\frac{k}{ik+2}\right)^2\right)\right\}\|\boldsymbol{\theta}_0 - \boldsymbol{\theta}_\mathcal{S}^*\|^2 \\
& + \sum_{i=1}^{t}\exp\left\{-\sum_{j=i+1}^{t}\left(\frac{2\alpha k}{jk+2} - \alpha\left(\frac{k}{jk+2}\right)^2\right)\right\}\frac{16\alpha kG^2}{\lambda^2(ik+2)^2}.
\end{aligned}
$$

On the other hand, by using $\sum_{j=i}^{t} \frac{k}{jk+2} \geq \int_{i}^{t+1} \frac{k}{sk+2} ds = \log\left(\frac{(t+1)k+2}{ik+2}\right)$ and $\sum_{j=i}^{t} \frac{k^2}{(jk+2)^2} \leq \int_{i-1}^{t} \frac{k^2}{(sk+2)^2} ds \leq \frac{k}{(i-1)k+2}$, we have

$$\mathbb{E}\left[\|\boldsymbol{\theta}_t - \boldsymbol{\theta}_{\mathcal{S}}^*\|^2\right] \leq \exp\left\{-2\alpha \log\left(\frac{(t+1)k+2}{k+2}\right) + \alpha\left(\frac{k^2}{(k+2)^2} + \frac{k}{k+2}\right)\right\} \|\boldsymbol{\theta}_0 - \boldsymbol{\theta}_{\mathcal{S}}^*\|^2$$

$$+ \sum_{i=1}^{t} \exp\left\{-2\alpha \log\left(\frac{(t+1)k+2}{(i+1)k+2}\right) + \frac{\alpha k}{ik+2}\right\} \frac{16\alpha k G^2}{\lambda^2 (ik+2)^2}$$

$$\leq \frac{e^{2\alpha}(k+2)^{2\alpha}}{((t+1)k+2)^{2\alpha}} \|\boldsymbol{\theta}_0 - \boldsymbol{\theta}_{\mathcal{S}}^*\|^2 + \frac{16\alpha e^{\alpha} G^2}{\lambda^2 ((t+1)k+2)^{2\alpha}} \sum_{i=1}^{t} \frac{k((i+1)k+2)^{2\alpha}}{(ik+2)^2}$$

$$\overset{①}{\leq} \frac{e^{2\alpha}(k+2)^{2\alpha}}{((t+1)k+2)^{2\alpha}} \|\boldsymbol{\theta}_0 - \boldsymbol{\theta}_{\mathcal{S}}^*\|^2 + \frac{16\alpha(4e)^{\alpha} G^2}{\lambda^2 ((t+1)k+2)^{2\alpha}} \sum_{i=1}^{t} \frac{k}{(ik+2)^{2-2\alpha}}$$

$$\leq \frac{e^{2\alpha}(k+2)^{2\alpha}}{((t+1)k+2)^{2\alpha}} \|\boldsymbol{\theta}_0 - \boldsymbol{\theta}_{\mathcal{S}}^*\|^2 + \frac{16\alpha(4e)^{\alpha} G^2}{\lambda^2 ((t+1)k+2)^{2\alpha}} \int_0^t k(sk+2)^{2\alpha-2} ds,$$

where ① is due to $(i+1)k+2 \leq 2(ik+2)$.

Let us now distinguish the following three complementary cases on the value of $\alpha$.

Case I: $\alpha \in (0, 1/2)$. In this case, we have

$$\mathbb{E}\left[\|\boldsymbol{\theta}_t - \boldsymbol{\theta}_{\mathcal{S}}^*\|^2\right] \leq \frac{e^{2\alpha}(k+2)^{2\alpha}}{((t+1)k+2)^{2\alpha}} \|\boldsymbol{\theta}_0 - \boldsymbol{\theta}_{\mathcal{S}}^*\|^2 + \frac{16\alpha(4e)^{\alpha} G^2}{\lambda^2 ((t+1)k+2)^{2\alpha}} \left(\frac{(tk+2)^{2\alpha-1} - 2^{2\alpha-1}}{2\alpha-1}\right)$$

$$\leq \frac{e^{2\alpha}(k+2)^{2\alpha}}{((t+1)k+2)^{2\alpha}} \|\boldsymbol{\theta}_0 - \boldsymbol{\theta}_{\mathcal{S}}^*\|^2 + \frac{16\alpha(4e)^{\alpha} G^2}{\lambda^2 ((t+1)k+2)^{2\alpha}(1-2\alpha)}$$

$$\leq \frac{3(k+2)^{2\alpha}}{((t+1)k+2)^{2\alpha}} \|\boldsymbol{\theta}_0 - \boldsymbol{\theta}_{\mathcal{S}}^*\|^2 + \frac{32 G^2}{\lambda^2 ((t+1)k+2)^{2\alpha}(1-2\alpha)},$$

where in the last inequality we have used $\alpha \in (0, 1/2)$.

Case II: $\alpha = 1/2$. In this case, we have

$$\mathbb{E}\left[\|\boldsymbol{\theta}_t - \boldsymbol{\theta}_{\mathcal{S}}^*\|^2\right] \leq \frac{3(k+2)}{(t+1)k+2} \|\boldsymbol{\theta}_0 - \boldsymbol{\theta}_{\mathcal{S}}^*\|^2 + \frac{32 G^2 \log(tk+2)}{\lambda^2 ((t+1)k+2)}.$$

Case III: $\alpha \in (1/2, 1]$. In this case, we have

$$\mathbb{E}\left[\|\boldsymbol{\theta}_t - \boldsymbol{\theta}_{\mathcal{S}}^*\|^2\right] \leq \frac{e^{2\alpha}(k+2)^{2\alpha}}{((t+1)k+2)^{2\alpha}} \|\boldsymbol{\theta}_0 - \boldsymbol{\theta}_{\mathcal{S}}^*\|^2 + \frac{16\alpha(4e)^{\alpha} G^2}{\lambda^2 ((t+1)k+2)^{2\alpha}} \left(\frac{(tk+2)^{2\alpha-1} - 2^{2\alpha-1}}{2\alpha-1}\right)$$

$$\leq \frac{e^{2\alpha}(k+2)^{2\alpha}}{((t+1)k+2)^{2\alpha}} \|\boldsymbol{\theta}_0 - \boldsymbol{\theta}_{\mathcal{S}}^*\|^2 + \frac{16\alpha(4e)^{\alpha} G^2 (tk+2)^{2\alpha-1}}{\lambda^2 ((t+1)k+2)^{2\alpha}(2\alpha-1)}$$

$$\leq \frac{8(k+2)^{2\alpha}}{((t+1)k+2)^{2\alpha}} \|\boldsymbol{\theta}_0 - \boldsymbol{\theta}_{\mathcal{S}}^*\|^2 + \frac{180 G^2}{\lambda^2 (2\alpha-1)(tk+2)},$$

where in the last inequality we have used $\alpha \leq 1$.

Since $F_{\mathcal{S}}(\boldsymbol{\theta})$ is $L$-smooth and $\boldsymbol{\theta}_{\mathcal{S}}^*$ is its optimum, then it has

$$\mathbb{E}\left[F_{\mathcal{S}}(\boldsymbol{\theta}_t) - F_{\mathcal{S}}(\boldsymbol{\theta}_{\mathcal{S}}^*)\right] \leq \frac{L}{2} \mathbb{E}\left[\|\boldsymbol{\theta}_t - \boldsymbol{\theta}_{\mathcal{S}}^*\|^2\right]$$

$$\leq \begin{cases} \frac{3L(k+2)^{2\alpha}}{2((t+1)k+2)^{2\alpha}} \|\boldsymbol{\theta}_0 - \boldsymbol{\theta}_{\mathcal{S}}^*\|^2 + \frac{16LG^2}{\lambda^2((t+1)k+2)^{2\alpha}(1-2\alpha)} & 0 < \alpha < \frac{1}{2} \\ \frac{3L(k+2)}{2(t+1)k+2} \|\boldsymbol{\theta}_0 - \boldsymbol{\theta}_{\mathcal{S}}^*\|^2 + \frac{16LG^2 \log(tk+2)}{\lambda^2((t+1)k+2)} & \alpha = \frac{1}{2} \\ \frac{4L(k+2)^{2\alpha}}{((t+1)k+2)^{2\alpha}} \|\boldsymbol{\theta}_0 - \boldsymbol{\theta}_{\mathcal{S}}^*\|^2 + \frac{90LG^2}{\lambda^2(2\alpha-1)(tk+2)} & \frac{1}{2} < \alpha \leq 1 \end{cases}.$$

The proof is completed. $\qquad\square$

## D.2 Proof of Theorem 2

*Proof.* Here we aim to use uniform stability to upper bound the generalization error. Suppose given $n$ samples $\mathcal{S} = \{z_1, z_2, \cdots, z_n\}$ where $z_i = (x_i, y_i)$ is sampled from an unknown distribution $\mathcal{D}$, one usually analyze the stability of an algorithm by replacing one sample in $\mathcal{S}$ by another sample from $\mathcal{D}$. Suppose the generated sample set $\mathcal{S}^{(i)} = \{z'_1, z'_2, \cdots, z'_n\} = \{z_1, z_2, \cdots, z_{i-1}, z'_i, z_{i+1} \cdots, z_n\}$ which only differs from the set $\mathcal{S}$ with the $i$-th sample. Then based on these two set, one can train the algorithm to obtain different solution $\theta$ of the function $F_{\mathcal{S}}(\theta)$. When using $\mathcal{S}^{(i)}$, we use $\widetilde{\theta}_t$ and $\tilde{v}_\tau^{(t)}$ to denote their corresponding versions $\theta_t$ and $v_\tau^{(t)}$ in Algorithm 1 trained on $\mathcal{S}$. Next, we can define

$$
\delta_\tau^{(t)} = \begin{cases} \|v_0^{(t)} - \tilde{v}_0^{(t)}\|_2 = \|\theta_{t-1} - \widetilde{\theta}_{t-1}\|_2, & \text{if } \tau = 0 \\ \|v_\tau^{(t)} - \tilde{v}_\tau^{(t)}\|_2, & \text{if } \tau \neq 0 \end{cases}
$$

Then for each iteration $(t, \tau)$ in Algorithm 1, with probability $1 - \frac{1}{n}$, the current selected samples in $\mathcal{S}$ and $\mathcal{S}^{(i)}$ are the same. In this case, by using the third part results in Lemma 1, we know that $\|v_{\tau+1}^{(t)} - \tilde{v}_{\tau+1}^{(t)}\|_2 \leq \left(1 - \frac{\eta_\tau^{(t)}\lambda L}{\lambda + L}\right)\|v_\tau^{(t)} - v_\tau^{(t)}\|_2$. Meanwhile with probability $\frac{1}{n}$, the selected samples are different in which we can use the third part results in Lemma 1: $\|v_{\tau+1}^{(t)} - \tilde{v}_{\tau+1}^{(t)}\|_2 \leq \min(1, a)\|v_\tau^{(t)} - v_\tau^{(t)}\|_2 + 2\eta_\tau^{(t)}G$, where $a = 1 - \frac{\eta_\tau^{(t)}\lambda L}{\lambda + L}$. So by setting $\eta_\tau^{(t)} = \frac{c}{\lambda((t-1)k+\tau+2)}$ where $c = \frac{\lambda+L}{L} \in (1, 2]$, combining these two cases yields

$$
\mathbb{E}\left[\delta_k^{(t)}\right] = \left(1 - \frac{1}{n}\right)\left(1 - \frac{\eta_{k-1}^{(t)}\lambda L}{\lambda + L}\right)\mathbb{E}\left[\delta_{k-1}^{(t)}\right] + \frac{1}{n}\left(1 - \frac{\eta_{k-1}^{(t)}\lambda L}{\lambda + L}\right)\mathbb{E}\left[\delta_{k-1}^{(t)}\right] + \frac{2\eta_{k-1}^{(t)}G}{n}
$$

$$
= \left(1 - \frac{\eta_{k-1}^{(t)}\lambda L}{\lambda + L}\right)\mathbb{E}\left[\delta_{k-1}^{(t)}\right] + \frac{2\eta_{k-1}^{(t)}G}{n}
$$

$$
\leq \prod_{i=1}^{k}\left(1 - \frac{cL}{\lambda + L}\frac{1}{(t-1)k+i+1}\right)\mathbb{E}\left[\|\delta_0^{(t)}\|^2\right]
$$

$$
+ \sum_{i=1}^{k}\frac{2cG}{\lambda n((t-1)k+i+1)}\prod_{j=i+1}^{k}\left(1 - \frac{cL}{\lambda + L}\frac{1}{(t-1)k+j+1}\right)
$$

$$
\overset{①}{\leq} \exp\left\{\sum_{i=1}^{k} - \frac{cL}{\lambda + L}\frac{1}{(t-1)k+i+1}\right\}\mathbb{E}\left[\|\delta_0^{(t)}\|^2\right]
$$

$$
+ \sum_{i=1}^{k}\frac{2cG}{\lambda n((t-1)k+i+1)}\exp\left\{-\sum_{j=i+1}^{k}\frac{cL}{\lambda + L}\frac{1}{(t-1)k+j+1}\right\}
$$

$$
\overset{②}{\leq} \exp\left\{-\frac{cL}{\lambda + L}\log\frac{tk+2}{(t-1)k+2}\right\}\mathbb{E}\left[\|\delta_0^{(t)}\|^2\right]
$$

$$
+ \sum_{i=1}^{k}\frac{2cG}{\lambda n((t-1)k+i+1)}\exp\left\{-\frac{cL}{\lambda + L}\log\frac{tk+2}{(t-1)k+i+2}\right\}
$$

$$
= \left(\frac{(t-1)k+2}{tk+2}\right)^{\frac{cL}{\lambda+L}}\mathbb{E}\left[\|\delta_0^{(t)}\|^2\right] + \sum_{i=1}^{k}\frac{2cG}{\lambda n((t-1)k+i+1)}\left(\frac{(t-1)k+i+2}{tk+2}\right)^{\frac{cL}{\lambda+L}}
$$

$$
\overset{③}{\leq} \left(\frac{(t-1)k+2}{tk+2}\right)\mathbb{E}\left[\|\delta_0^{(t)}\|^2\right] + \frac{4ckG}{\lambda n(tk+2)}
$$

where in ① we have used $1 + x \leq e^x$, in ② we have used $\sum_{j=i}^{k}\frac{1}{a+j+1} \geq \int_i^{k+1}\frac{1}{a+s+1}ds = \log\left(\frac{a+k+2}{a+i+1}\right)$ and ③ is due to $(a+1)^\beta \leq 2a^\beta$ for all $a \geq 1$ and $c = \frac{\lambda+L}{L}$. Then by setting $\beta = \frac{c\lambda L}{\lambda+L}$

it follows from the update of slow parameter and the above inequalities that

$$\mathbb{E}\left[\|\boldsymbol{\delta}_0^{(t+1)}\|^2\right] = \mathbb{E}\left[\|\boldsymbol{\theta}_t - \boldsymbol{\theta}_{\mathcal{S}}^*\|^2\right] \leq (1-\alpha)\mathbb{E}\left[\|\boldsymbol{\theta}_{t-1} - \boldsymbol{\theta}_{\mathcal{S}}^*\|^2\right] + \alpha\mathbb{E}\left[\|\boldsymbol{v}_k^{(t)} - \boldsymbol{\theta}_{\mathcal{S}}^*\|^2\right]$$

$$\leq \left(1 - \alpha + \alpha\left(\frac{(t-1)k+2}{tk+2}\right)\right)\mathbb{E}\left[\|\boldsymbol{\theta}_{t-1} - \boldsymbol{\theta}_{\mathcal{S}}^*\|^2\right] + \frac{4c\alpha kG}{n(tk+2)}$$

$$\leq \left(1 - \frac{\alpha k}{tk+2}\right)\mathbb{E}\left[\|\boldsymbol{\delta}_0^{(t)}\|^2\right] + \frac{4c\alpha kG}{\lambda n(tk+2)}.$$

Unwinding this recurrence relation from time instance $t$ to 1 yields

$$\mathbb{E}\left[\|\boldsymbol{\delta}_0^{(t+1)}\|^2\right] \leq \left(1 - \frac{\alpha k}{tk+2}\right)\mathbb{E}\left[\|\boldsymbol{\delta}_0^{(t)}\|^2\right] + \frac{4c\alpha kG}{n\lambda(tk+2)}$$

$$\leq \prod_{i=1}^t\left(1 - \frac{\alpha k}{ik+2}\right)\|\boldsymbol{\delta}_0^{(0)}\|^2 + \sum_{i=1}^t\prod_{j=i+1}^t\left(1 - \frac{\alpha k}{jk+2}\right)\frac{4c\alpha kG}{n\lambda(ik+2)}$$

$$\leq \prod_{i=1}^t\left(1 - \frac{\alpha k}{ik+2}\right)\|\boldsymbol{\delta}_0^{(0)}\|^2 + \sum_{i=1}^t\exp\left\{-\sum_{j=i+1}^t\frac{\alpha k}{jk+2}\right\}\frac{4c\alpha kG}{n\lambda(ik+2)}$$

$$\overset{①}{\leq} \sum_{i=1}^t\exp\left\{-\alpha\log\left(\frac{(t+1)k+2}{(i+1)k+2}\right)\right\}\frac{4c\alpha kG}{n\lambda(ik+2)}$$

$$\leq \frac{4c\alpha G}{n\lambda((t+1)k+2)^\alpha}\sum_{i=1}^t\frac{k((i+1)k+2)^\alpha}{(ik+2)}$$

$$\overset{②}{\leq} \frac{4\cdot 2^\alpha c\alpha G}{n\lambda((t+1)k+2)^\alpha}\sum_{i=1}^t\frac{k}{(ik+2)^{1-\alpha}}$$

$$\leq \frac{4\cdot 2^\alpha c\alpha G}{n\lambda((t+1)k+2)^\alpha}\int_0^t k(sk+2)^{\alpha-1}ds$$

$$= \frac{4\cdot 2^\alpha c\alpha G}{n\lambda((t+1)k+2)^\alpha}\frac{(tk+1)^\alpha - 1}{\alpha}$$

$$\leq \frac{8G(\lambda+L)}{n\lambda L}\frac{(tk+1)^\alpha - 1}{((t+1)k+2)^\alpha}$$

$$\leq \frac{16G}{n\lambda}\frac{(tk+1)^\alpha - 1}{((t+1)k+2)^\alpha},$$

where in ① we have used $\sum_{j=i}^t\frac{k}{jk+2} \geq \int_i^{t+1}\frac{k}{sk+2}ds = \log\left(\frac{(t+1)k+2}{ik+2}\right)$ and $\sum_{j=i}^t\frac{k^2}{(jk+2)^2} \leq \int_{i-1}^t\frac{k^2}{(sk+2)^2}ds \leq \frac{k}{(i-1)k+2}$, ② is due to $(i+1)k+2 \leq 2(ik+2)$.

Finally, we have that function $\ell(\boldsymbol{\theta}, \cdot)$ is $G$-Lipschitz, and thus obtain

$$\mathbb{E}\left[|\ell(f(\boldsymbol{x}; \boldsymbol{\theta}_T); \boldsymbol{y}) - \ell(f(\boldsymbol{x}; \widetilde{\boldsymbol{\theta}}_T); \boldsymbol{y})|\right] \leq G\mathbb{E}\left[\|\boldsymbol{\theta}_{t+1} - \widetilde{\boldsymbol{\theta}}_{t+1}\|_2\right] \leq \frac{16G^2}{n\lambda}\frac{(tk+1)^\alpha - 1}{((t+1)k+2)^\alpha}.$$

The proof is completed. □

## D.3 Proof of Corollary 1

*Proof.* Now we combine all results in Theorems 1 and 2 together, including the above optimization error and generalization error, and use Lemma 1 in the manuscript to obtain

$$\varepsilon_{\text{opt}} + \varepsilon_{\text{gen}} \leq \begin{cases} \frac{3L(k+2)^{2\alpha}}{2((t+1)k+2)^{2\alpha}}\|\boldsymbol{\theta}_0 - \boldsymbol{\theta}_{\mathcal{S}}^*\|^2 + \frac{16LG^2}{\lambda^2((t+1)k+2)^{2\alpha}(1-2\alpha)} + \frac{16G}{n\lambda}\frac{(tk+1)^\alpha - 1}{((t+1)k+2)^\alpha}, & 0 < \alpha < \frac{1}{2} \\ \frac{3L(k+2)}{2(t+1)k+2}\|\boldsymbol{\theta}_0 - \boldsymbol{\theta}_{\mathcal{S}}^*\|^2 + \frac{16LG^2\log(tk+2)}{\lambda^2((t+1)k+2)} + \frac{16G}{n\lambda}\frac{(tk+1)^\alpha - 1}{((t+1)k+2)^\alpha}, & \alpha = \frac{1}{2} \\ \frac{4L(k+2)^{2\alpha}}{((t+1)k+2)^{2\alpha}}\|\boldsymbol{\theta}_0 - \boldsymbol{\theta}_{\mathcal{S}}^*\|^2 + \frac{90LG^2}{\lambda^2(2\alpha-1)(tk+2)} + \frac{16G}{n\lambda}\frac{(tk+1)^\alpha - 1}{((t+1)k+2)^\alpha}, & \frac{1}{2} < \alpha \leq 1 \end{cases}.$$

The proof is completed.

□

### D.4 Proof of Theorem 3

*Proof.* Here we prove our results in three steps. In the first step, we first prove the optimization error. Then in the second step, we consider to prove the generalization error bound. Finally, we combine these two error bounds by using the risk decomposition.

**Step 1. Optimization error.** To begin with, since function $F_{\mathcal{S}}(\boldsymbol{\theta})$ is convex, then we have

$$\langle F_{\mathcal{S}}(\boldsymbol{v}_\tau^{(t)}), \boldsymbol{\theta}_{\mathcal{S}}^* - \boldsymbol{v}_\tau^{(t)} \rangle \leq F_{\mathcal{S}}(\boldsymbol{\theta}_{\mathcal{S}}^*) - F_{\mathcal{S}}(\boldsymbol{v}_\tau^{(t)}).$$

Next, we can bound

$$
\begin{aligned}
\mathbb{E}\left[\|\boldsymbol{v}_{\tau+1}^{(t)} - \boldsymbol{\theta}_{\mathcal{S}}^*\|_2^2\right] =& \mathbb{E}\left[\|\boldsymbol{v}_\tau^{(t)} - \eta_\tau^{(t)}\boldsymbol{g}_\tau^{(t)} - \boldsymbol{\theta}_{\mathcal{S}}^*\|_2^2\right] \\
=& \mathbb{E}\left[\|\boldsymbol{v}_\tau^{(t)} - \boldsymbol{\theta}_{\mathcal{S}}^*\|_2^2 - 2\eta_\tau^{(t)}\langle\boldsymbol{v}_\tau^{(t)} - \boldsymbol{\theta}_{\mathcal{S}}^*, \boldsymbol{g}_\tau^{(t)}\rangle + (\eta_\tau^{(t)})^2\|\boldsymbol{g}_\tau^{(t)}\|_2^2\right] \\
=& \mathbb{E}\left[\|\boldsymbol{v}_\tau^{(t)} - \boldsymbol{\theta}_{\mathcal{S}}^*\|_2^2 - 2\eta_\tau^{(t)}\langle\boldsymbol{v}_\tau^{(t)} - \boldsymbol{\theta}_{\mathcal{S}}^*, \nabla F_{\mathcal{S}}(\boldsymbol{v}_\tau^{(t)})\rangle + (\eta_\tau^{(t)})^2\|\boldsymbol{g}_\tau^{(t)}\|_2^2\right] \\
\leq& \mathbb{E}\left[\|\boldsymbol{v}_\tau^{(t)} - \boldsymbol{\theta}_{\mathcal{S}}^*\|_2^2 + 2\eta_\tau^{(t)}(F_{\mathcal{S}}(\boldsymbol{\theta}_{\mathcal{S}}^*) - F_{\mathcal{S}}(\boldsymbol{v}_\tau^{(t)})) + (\eta_\tau^{(t)})^2\|\boldsymbol{g}_\tau^{(t)}\|_2^2\right] \\
\leq& \mathbb{E}\left[\|\boldsymbol{v}_\tau^{(t)} - \boldsymbol{\theta}_{\mathcal{S}}^*\|_2^2 + 2\eta_\tau^{(t)}(F_{\mathcal{S}}(\boldsymbol{\theta}_{\mathcal{S}}^*) - F_{\mathcal{S}}\boldsymbol{v}_\tau^{(t)})) + (\eta_\tau^{(t)})^2 G^2\right]
\end{aligned}
$$

Then by rearranging the above inequality, we can obtain

$$F_{\mathcal{S}}(\boldsymbol{v}_\tau^{(t)}) - F_{\mathcal{S}}(\boldsymbol{\theta}_{\mathcal{S}}^*) \leq \frac{1}{2\eta_\tau^{(t)}}\mathbb{E}\left[\|\boldsymbol{v}_\tau^{(t)} - \boldsymbol{\theta}_{\mathcal{S}}^*\|_2^2 - \|\boldsymbol{v}_{\tau+1}^{(t)} - \boldsymbol{\theta}_{\mathcal{S}}^*\|_2^2\right] + \frac{G^2}{2}\eta_\tau^{(t)}.$$

Next, by setting a constant learning rate $\eta_\tau^{(t)} = \eta$, we sum up the above inequality from $\tau = 0$ to $k - 1$ and obtain

$$
\begin{aligned}
\frac{1}{k}\sum_{\tau=0}^{k-1}\left(F_{\mathcal{S}}(\boldsymbol{v}_\tau^{(t)}) - F_{\mathcal{S}}(\boldsymbol{\theta}_{\mathcal{S}}^*)\right) \leq& \frac{1}{2\eta k}\mathbb{E}\left[\|\boldsymbol{v}_0^{(t)} - \boldsymbol{\theta}_{\mathcal{S}}^*\|_2^2 - \|\boldsymbol{v}_k^{(t)} - \boldsymbol{\theta}_{\mathcal{S}}^*\|_2^2\right] + \frac{\eta G^2}{2} \\
=& \frac{1}{2\eta k}\mathbb{E}\left[\|\boldsymbol{\theta}_{t-1} - \boldsymbol{\theta}_{\mathcal{S}}^*\|_2^2 - \|\boldsymbol{v}_k^{(t)} - \boldsymbol{\theta}_{\mathcal{S}}^*\|_2^2\right] + \frac{\eta G^2}{2}.
\end{aligned}
$$

Now we consider the term $\|\boldsymbol{\theta}_{t-1} - \boldsymbol{\theta}_{\mathcal{S}}^*\|_2^2 - \|\boldsymbol{v}_k^{(t)} - \boldsymbol{\theta}_{\mathcal{S}}^*\|_2^2$ as follows:

$$
\begin{aligned}
\|\boldsymbol{\theta}_{t-1} - \boldsymbol{\theta}_{\mathcal{S}}^*\|_2^2 - \|\boldsymbol{v}_k^{(t)} - \boldsymbol{\theta}_{\mathcal{S}}^*\|_2^2 =& \left\langle\boldsymbol{\theta}_{t-1} - \boldsymbol{v}_k^{(t)}, \boldsymbol{\theta}_{t-1} + \boldsymbol{v}_k^{(t)} - 2\boldsymbol{\theta}_{\mathcal{S}}^*\right\rangle \\
\overset{①}{=}& \left\langle\boldsymbol{\theta}_{t-1} - \frac{\boldsymbol{\theta}_t - (1-\alpha)\boldsymbol{\theta}_{t-1}}{\alpha}, \boldsymbol{\theta}_{t-1} + \frac{\boldsymbol{\theta}_t - (1-\alpha)\boldsymbol{\theta}_{t-1}}{\alpha} - 2\boldsymbol{\theta}_{\mathcal{S}}^*\right\rangle \\
=& -\frac{1}{\alpha^2}\left\langle\boldsymbol{\theta}_{t-1} - \boldsymbol{\theta}_{\mathcal{S}}^* - (\boldsymbol{\theta}_t - \boldsymbol{\theta}_{\mathcal{S}}^*), (1-2\alpha)(\boldsymbol{\theta}_{t-1} - \boldsymbol{\theta}_{\mathcal{S}}^*) - (\boldsymbol{\theta}_t - \boldsymbol{\theta}_{\mathcal{S}}^*)\right\rangle \\
=& -\frac{1}{\alpha^2}\left[(1-\alpha)\|\boldsymbol{\theta}_{t-1} - \boldsymbol{\theta}_t\|^2 + \alpha\|\boldsymbol{\theta}_t - \boldsymbol{\theta}_{\mathcal{S}}^*\|^2 - \alpha\|\boldsymbol{\theta}_{t-1} - \boldsymbol{\theta}_{\mathcal{S}}^*\|^2\right] \\
\leq& \frac{1}{\alpha}\left[\|\boldsymbol{\theta}_{t-1} - \boldsymbol{\theta}_{\mathcal{S}}^*\|^2 - \|\boldsymbol{\theta}_t - \boldsymbol{\theta}_{\mathcal{S}}^*\|^2\right]
\end{aligned}
$$

where ① holds since $\boldsymbol{\theta}_t = (1-\alpha)\boldsymbol{\theta}_{t-1} + \alpha\boldsymbol{v}_k^{(t)}$. In this way, we can upper bound

$$\frac{1}{k}\sum_{\tau=0}^{k-1}\left(F_{\mathcal{S}}(\boldsymbol{v}_\tau^{(t)}) - F_{\mathcal{S}}(\boldsymbol{\theta}_{\mathcal{S}}^*)\right) \leq \frac{1}{2\alpha\eta k}\mathbb{E}\left[\|\boldsymbol{\theta}_{t-1} - \boldsymbol{\theta}_{\mathcal{S}}^*\|^2 - \|\boldsymbol{\theta}_t - \boldsymbol{\theta}_{\mathcal{S}}^*\|^2\right] + \frac{\eta G^2}{2}.$$

Finally, we can sum up from $t = 1$ to $t = T$ and obtain

$$
\begin{aligned}
\frac{1}{kT}\sum_{t=1}^{T}\sum_{\tau=0}^{k-1}\left(F_{\mathcal{S}}(\boldsymbol{v}_\tau^{(t)}) - F_{\mathcal{S}}(\boldsymbol{\theta}_{\mathcal{S}}^*)\right) \leq& \frac{1}{2\alpha\eta kT}\mathbb{E}\left[\|\boldsymbol{\theta}_0 - \boldsymbol{\theta}_{\mathcal{S}}^*\|^2\right] + \frac{\eta G^2}{2} \\
\overset{①}{=}& G\sqrt{\frac{1}{\alpha kT}\mathbb{E}\left[\|\boldsymbol{\theta}_0 - \boldsymbol{\theta}_{\mathcal{S}}^*\|^2\right]}
\end{aligned}
$$

where ① holds by setting $\eta = \sqrt{\frac{1}{\alpha k T G^2} \mathbb{E}\left[\|\boldsymbol{\theta}_0 - \boldsymbol{\theta}_{\mathcal{S}}^*\|^2\right]}$.

**Step 2. Generalization error via uniform stability.** Here we aim to use uniform stability to upper bound the generalization error. Suppose given $n$ samples $\mathcal{S} = \{\boldsymbol{z}_1, \boldsymbol{z}_2, \cdots, \boldsymbol{z}_n\}$ where $\boldsymbol{z}_i = (\boldsymbol{x}_i, \boldsymbol{y}_i)$ is sampled from an unknown distribution $\mathcal{D}$, one usually analyze the stability of an algorithm by replacing one sample in $\mathcal{S}$ by another sample from $\mathcal{D}$. Suppose the generated sample set $\mathcal{S}^{(i)} = \{\boldsymbol{z}_1', \boldsymbol{z}_2', \cdots, \boldsymbol{z}_n'\} = \{\boldsymbol{z}_1, \boldsymbol{z}_2, \cdots, \boldsymbol{z}_{i-1}, \boldsymbol{z}_i', \boldsymbol{z}_{i+1} \cdots, \boldsymbol{z}_n\}$ which only differs from the set $\mathcal{S}$ with the $i$-th sample. Then based on these two set, one can train the algorithm to obtain different solution $\boldsymbol{\theta}$ of the function $F_{\mathcal{S}}(\boldsymbol{\theta})$. When using $\mathcal{S}^{(i)}$, we use $\widetilde{\boldsymbol{\theta}}_t$ and $\tilde{\boldsymbol{v}}_\tau^{(t)}$ to denote their corresponding versions $\boldsymbol{\theta}_t$ and $\boldsymbol{v}_\tau^{(t)}$ in Algorithm 1 trained on $\mathcal{S}$. Next, we can define

$$\boldsymbol{\delta}_\tau^{(t)} = \begin{cases} \|\boldsymbol{v}_0^{(t)} - \tilde{\boldsymbol{v}}_0^{(t)}\|_2 = \|\boldsymbol{\theta}_{t-1} - \widetilde{\boldsymbol{\theta}}_{t-1}\|_2, & \text{if } \tau = 0 \\ \|\boldsymbol{v}_\tau^{(t)} - \tilde{\boldsymbol{v}}_\tau^{(t)}\|_2, & \text{if } \tau \neq 0 \end{cases}$$

Then for each iteration $(t, \tau)$ in Algorithm 1, with probability $1 - \frac{1}{n}$, the current selected samples in $\mathcal{S}$ and $\mathcal{S}^{(i)}$ are the same. In this case, by using the second part results in Lemma 1 for convex problems, we know that $\|\boldsymbol{v}_{\tau+1}^{(t)} - \tilde{\boldsymbol{v}}_{\tau+1}^{(t)}\|_2 \leq \|\boldsymbol{v}_\tau^{(t)} - \tilde{\boldsymbol{v}}_\tau^{(t)}\|_2$. Meanwhile with probability $\frac{1}{n}$, the selected samples are different in which we can use the second part results in Lemma 1: $\|\boldsymbol{v}_{\tau+1}^{(t)} - \tilde{\boldsymbol{v}}_{\tau+1}^{(t)}\|_2 \leq \|\boldsymbol{v}_\tau^{(t)} - \tilde{\boldsymbol{v}}_\tau^{(t)}\|_2 + 2\eta_\tau^{(t)}G$. So combining these two cases yields

$$\mathbb{E}\left[\boldsymbol{\delta}_{\tau+1}^{(t)}\right] = \left(1 - \frac{1}{n}\right)\mathbb{E}\left[\boldsymbol{\delta}_\tau^{(t)}\right] + \frac{1}{n}\mathbb{E}\left[\boldsymbol{\delta}_{\tau+1}^{(t)}\right] + \frac{2\eta_\tau^{(t)}G}{n} = \mathbb{E}\left[\boldsymbol{\delta}_\tau^{(t)}\right] + \frac{2\eta_\tau^{(t)}G}{n}$$
$$= \mathbb{E}\left[\boldsymbol{\delta}_0^{(t)}\right] + \sum_{s=0}^{\tau} \frac{2\eta_s^{(t)}G}{n}. \tag{7}$$

Let $(t_0, \tau_0)$ denote the iteration at which $\boldsymbol{\delta}_{\tau_0}^{(t_0)} = 0$ and $\boldsymbol{\delta}_{\tau_0+1}^{(t_0)} = 0$. For brevity, we use $\mathcal{E} = \mathbf{1}[\boldsymbol{\delta}_{\tau_0}^{(t_0)} = 0]$ to denote the event that $\boldsymbol{\delta}_{\tau_0}^{(t_0)} = 0$. Now we need to use the recurrent formulation in (7) to derive the upper bound of $\|\boldsymbol{\theta}_t - \widetilde{\boldsymbol{\theta}}_t\|_2$ as follows:

$$\mathbb{E}\left[\|\boldsymbol{\theta}_{t+1} - \widetilde{\boldsymbol{\theta}}_{t+1}\|_2\right]$$
$$= \mathbb{E}\left[\|(1-\alpha)\boldsymbol{\theta}_t + \alpha\boldsymbol{v}_k^{(t+1)} - (1-\alpha)\widetilde{\boldsymbol{\theta}}_t - \alpha\tilde{\boldsymbol{v}}_k^{(t+1)}\|_2\right]$$
$$\leq (1-\alpha)\mathbb{E}\left[\|\boldsymbol{\theta}_t - \widetilde{\boldsymbol{\theta}}_t\|_2\right] + \alpha\mathbb{E}\left[\|\boldsymbol{v}_k^{(t+1)} - \tilde{\boldsymbol{v}}_k^{(t+1)}\|_2\right]$$
$$\leq (1-\alpha)\mathbb{E}\left[\|\boldsymbol{\theta}_t - \widetilde{\boldsymbol{\theta}}_t\|_2\right] + \alpha\mathbb{E}\left[\boldsymbol{\delta}_0^{(t)}\right] + \sum_{\tau=0}^{k-1} \frac{2\alpha\eta_\tau^{(t)}G}{n}$$
$$\overset{①}{=} \mathbb{E}\left[\|\boldsymbol{\theta}_t - \widetilde{\boldsymbol{\theta}}_t\|_2\right] + \sum_{\tau=0}^{k-1} \frac{2\alpha\eta_\tau^{(t)}G}{n} \tag{8}$$
$$= \mathbb{E}\left[\|\boldsymbol{\theta}_{t_0} - \widetilde{\boldsymbol{\theta}}_{t_0}\|_2\right] + \sum_{\tau=\tau_0}^{k-1} \frac{2\alpha\eta_\tau^{(t_0)}G}{n} + \sum_{i=t_0+2}^{t}\sum_{\tau=0}^{k-1} \frac{2\alpha\eta_\tau^{(i)}G}{n}$$
$$\leq \mathbb{E}\left[\|\boldsymbol{\theta}_{t_0} - \widetilde{\boldsymbol{\theta}}_{t_0}\|_2\right] + \sum_{i=t_0+1}^{t}\sum_{\tau=0}^{k-1} \frac{2\alpha\eta_\tau^{(i)}G}{n}$$
$$\leq \frac{2\alpha\eta GkT}{n}$$

where ① holds since $\|\boldsymbol{\theta}_t - \widetilde{\boldsymbol{\theta}}_t\|_2 = \|\boldsymbol{v}_0^{(t+1)} - \tilde{\boldsymbol{v}}_0^{(t+1)}\|_2$. On the other hand, we have that function $\ell(f(\cdot; \boldsymbol{\theta}); \cdot)$ is $G$-Lipschitz, and thus obtain

$$\mathbb{E}\left[|\ell(f(\boldsymbol{x}; \boldsymbol{\theta}_T); \boldsymbol{y}) - \ell(f(\boldsymbol{x}; \widetilde{\boldsymbol{\theta}}_T); \boldsymbol{y})|\right] \leq G\mathbb{E}\left[\|\boldsymbol{\theta}_{t+1} - \widetilde{\boldsymbol{\theta}}_{t+1}\|_2\right] \leq \frac{2\alpha\eta G^2 kT}{n}.$$

Now we consider the average case where $\bar{v}_k^{(T)} = \frac{1}{kT} \sum_{i=1}^{T} \sum_{\tau=0}^{k-1} v_\tau^{(t)}$ is the output which is consistent with our optimization analysis which also needs to output the average of all $v_\tau^{(t)}$. To begin with, we have

$$
\begin{aligned}
v_0^{(t+1)} =& \theta_t = (1-\alpha)\theta_{t-1} + \alpha v_k^{(t)} = (1-\alpha)\theta_{t-1} + \alpha \left( v_0^{(t)} - \sum_{\tau=0}^{k-1} \eta_\tau^{(t)} g_\tau^{(t)} \right) \\
=& (1-\alpha)\theta_{t-1} + \alpha \left( \theta_{t-1} - \sum_{\tau=0}^{k-1} \eta_\tau^{(t)} g_\tau^{(t)} \right) = \theta_{t-1} - \alpha \sum_{\tau=0}^{k-1} \eta_\tau^{(t)} g_\tau^{(t)} \\
=& (1-\alpha)\theta_{t-2} + \alpha \left( \theta_{t-2} - \sum_{\tau=0}^{k-1} \eta_\tau^{(t-1)} g_\tau^{(t-1)} \right) - \alpha \sum_{\tau=0}^{k-1} \eta_\tau^{(t)} g_\tau^{(t)} \\
=& \theta_{t-2} - \alpha \sum_{i=t-1}^{t} \sum_{\tau=0}^{k-1} \eta_\tau^{(i)} g_\tau^{(t)} = \theta_0 - \alpha \sum_{i=1}^{t} \sum_{\tau=0}^{k-1} \eta_\tau^{(i)} g_\tau^{(i)}.
\end{aligned}
$$

In this way, we can know the formulation of the average of all $v_\tau^{(t)}$ as follows:

$$
\begin{aligned}
\bar{v}_k^{(T)} =& \frac{1}{kT} \sum_{i=1}^{T} \sum_{\tau=0}^{k-1} v_\tau^{(t)} = \theta_0 - \alpha \sum_{i=1}^{T} \sum_{\tau=0}^{k-1} \frac{\eta_\tau^{(i)}(kT - (i-1)k - \tau + 1)}{kT} g_\tau^{(t)} \\
=& \bar{v}_{k-1}^{(T)} - \alpha \frac{\eta_{k-1}^{(T)}(kT - (T-1)k - k + 2)}{kT} g_\tau^{(t)}.
\end{aligned}
$$

Then we have $\|\bar{v}_k^{(T)} - \bar{v}_{k-1}^{(T)}\|_2 \leq \alpha \frac{\eta_{k-1}^{(T)}(kT-(T-1)k-k+2)}{kT} \|g_\tau^{(t)}\| \leq \alpha G \frac{\eta_{k-1}^{(T)}(kT-(T-1)k-k+2)}{kT}$. Then we can use the second results in Lemma 1. For each iteration $(t,\tau)$ in Algorithm 1, with probability $1 - \frac{1}{n}$, the current selected samples in $\mathcal{S}$ and $\mathcal{S}^{(i)}$ are the same. In this case, by using the second part results in Lemma 1 for convex problems, we know that $\|v_{\tau+1}^{(t)} - \tilde{v}_{\tau+1}^{(t)}\|_2 \leq \|v_\tau^{(t)} - \tilde{v}_\tau^{(t)}\|_2$. Meanwhile with probability $\frac{1}{n}$, the selected samples are different in which we can use the second part results in Lemma 1: $\|v_{\tau+1}^{(t)} - \tilde{v}_{\tau+1}^{(t)}\|_2 \leq \|v_\tau^{(t)} - \tilde{v}_\tau^{(t)}\|_2 + 2\eta_\tau^{(t)} \alpha G \frac{kT-(t-1)k-\tau+1}{kT}$. So combining these two cases yields

$$
\begin{aligned}
& \mathbb{E}\left[\delta_{\tau+1}^{(T)}\right] \\
=& \left(1 - \frac{1}{n}\right) \mathbb{E}\left[\delta_\tau^{(T)}\right] + \frac{1}{n}\mathbb{E}\left[\delta_{\tau+1}^{(T)}\right] + \frac{2\alpha\eta_\tau^{(T)}G}{n} = \mathbb{E}\left[\delta_\tau^{(T)}\right] + \frac{2\alpha\eta_\tau^{(T)}G}{n}\frac{kT - (t-1)k - \tau + 1}{kT} \\
=& \mathbb{E}\left[\delta_0^{(T)}\right] + \sum_{\tau=0}^{k-1} \frac{2\alpha\eta_\tau^{(T)}G}{n}\frac{kT - (t-1)k - \tau + 1}{kT} \\
\leq& \mathbb{E}\left[\delta_0^{(0)}\right] + \sum_{t=1}^{T}\sum_{\tau=0}^{k-1} \frac{2\alpha\eta_\tau^{(t)}G}{n}\frac{kT - (t-1)k - \tau + 1}{kT} \\
=& \sum_{t=1}^{T}\sum_{\tau=0}^{k-1} \frac{2\alpha\eta_\tau^{(t)}G}{n}\frac{kT - (t-1)k - \tau + 1}{kT} \\
=& \sum_{i=1}^{kT} \frac{2\alpha\eta_\tau^{(t)}G}{n}\frac{kT - i + 1}{kT} \\
\leq& \frac{\alpha\eta GkT}{n}
\end{aligned}
\tag{9}
$$

On the other hand, we have that function $\ell(f(\cdot; \theta); \cdot)$ is $G$-Lipschitz, and thus obtain

$$
\mathbb{E}\left[|\ell(f(x; \theta_T); y) - \ell(f(x; \tilde{\theta}_T); y)|\right] \leq G\mathbb{E}\left[\|\theta_{t+1} - \tilde{\theta}_{t+1}\|_2\right] \leq \frac{\alpha\eta G^2 kT}{n}.
$$

Finally, we can use Lemma 1 to prove that the generalization error

$$\varepsilon_{\text{gen}} \leq \frac{\alpha \eta G^2 kT}{n}.$$

By comparison, when output the average of all $\boldsymbol{v}_\tau^{(t)}$, then its generalization error is $\frac{\alpha \eta G^2 kT}{n}$ which is slightly better than the generalization error $\frac{2\alpha \eta G^2 kT}{n}$ of the solution at the last iteration. Such an improvement is consistent with the analysis results on convex problem in [7].

**Step 3. Excess risk error by combining optimization error and generalization error.** Now we combine all results together, including the above optimization error and generalization error, and use Lemma 1 in the manuscript to obtain

$$\varepsilon_{\text{opt}} + \varepsilon_{\text{gen}} \leq \frac{1}{2\alpha\eta kT}\mathbb{E}\left[\|\boldsymbol{\theta}_0 - \boldsymbol{\theta}_{\mathcal{S}}^*\|^2\right] + \frac{\eta G^2}{2} + \frac{\alpha \eta G^2 kT}{n}.$$

The proof is completed. $\qquad\square$

### D.5 Proof of Theorem 4

*Proof.* Here we prove our results in three steps. In the first step, we first prove the optimization error. Then in the second step, we consider to prove the generalization error bound. Finally, we combine these two error bounds by using the risk decomposition.

To begin with, we first define the variable $\boldsymbol{u}_\tau^{(t)} = \alpha \boldsymbol{v}_\tau^{(t)} + (1-\alpha)\boldsymbol{\theta}_{t-1} = \alpha \boldsymbol{v}_\tau^{(t)} + (1-\alpha)\boldsymbol{v}_0^{(t)}$. It can be observed that when $\tau = k$, then $\boldsymbol{\theta}_t = \boldsymbol{u}_k^{(t)}$. In this way, we also can obtain the updating rule of $\boldsymbol{u}_{\tau+1}^{(t)}$ as follows:

$$\boldsymbol{u}_{\tau+1}^{(t)} = \alpha \boldsymbol{v}_\tau^{(t)} + (1-\alpha)\boldsymbol{v}_0^{(t)} = \boldsymbol{u}_\tau^{(t)} - \alpha\eta_\tau^{(t)}\boldsymbol{g}_\tau^{(t)},$$

where $\boldsymbol{g}_\tau^{(t)}$ denotes the stochastic gradient at the point $\boldsymbol{v}_\tau^{(t)}$.

**Step 1. Optimization error.** Firstly, we can bound

$$\mathbb{E}\left[F_{\mathcal{S}}(\boldsymbol{u}_{\tau+1}^{(t)})\right]$$

$$\leq \mathbb{E}\left[F_{\mathcal{S}}(\boldsymbol{u}_\tau^{(t)}) + \langle \nabla F_{\mathcal{S}}(\boldsymbol{u}_\tau^{(t)}), \boldsymbol{u}_{\tau+1}^{(t)} - \boldsymbol{u}_\tau^{(t)} \rangle + \frac{L}{2}\|\boldsymbol{u}_{\tau+1}^{(t)} - \boldsymbol{u}_\tau^{(t)}\|^2\right]$$

$$= \mathbb{E}\left[F_{\mathcal{S}}(\boldsymbol{u}_\tau^{(t)}) - \alpha\eta_\tau^{(t)}\langle \nabla F_{\mathcal{S}}(\boldsymbol{u}_\tau^{(t)}), \boldsymbol{g}_\tau^{(t)} \rangle + \frac{L\alpha^2(\eta_\tau^{(t)})^2}{2}\|\boldsymbol{g}_\tau^{(t)}\|^2\right]$$

$$\overset{①}{=} \mathbb{E}\left[F_{\mathcal{S}}(\boldsymbol{u}_\tau^{(t)}) - \alpha\eta_\tau^{(t)}\langle \nabla F_{\mathcal{S}}(\boldsymbol{u}_\tau^{(t)}), \nabla F_{\mathcal{S}}(\boldsymbol{v}_\tau^{(t)}) \rangle + \frac{L\alpha^2(\eta_\tau^{(t)})^2}{2}\|\boldsymbol{g}_\tau^{(t)}\|^2\right]$$

$$= \mathbb{E}\left[F_{\mathcal{S}}(\boldsymbol{u}_\tau^{(t)}) - \alpha\eta_\tau^{(t)}\|\nabla F_{\mathcal{S}}(\boldsymbol{u}_\tau^{(t)})\|^2 + \alpha\eta_\tau^{(t)}\langle \nabla F_{\mathcal{S}}(\boldsymbol{u}_\tau^{(t)}), \nabla F_{\mathcal{S}}(\boldsymbol{u}_\tau^{(t)}) - \nabla F_{\mathcal{S}}(\boldsymbol{v}_\tau^{(t)}) \rangle + \frac{L\alpha^2(\eta_\tau^{(t)})^2}{2}\|\boldsymbol{g}_\tau^{(t)}\|^2\right]$$

$$\overset{②}{\leq} \mathbb{E}\left[F_{\mathcal{S}}(\boldsymbol{u}_\tau^{(t)}) - 2\mu\alpha\eta_\tau^{(t)}(F_{\mathcal{S}}(\boldsymbol{u}_\tau^{(t)}) - F_{\mathcal{S}}(\boldsymbol{\theta}_{\mathcal{S}}^*)) + \alpha\eta_\tau^{(t)}\langle \nabla F_{\mathcal{S}}(\boldsymbol{u}_\tau^{(t)}), \nabla F_{\mathcal{S}}(\boldsymbol{u}_\tau^{(t)}) - \nabla F_{\mathcal{S}}(\boldsymbol{v}_\tau^{(t)}) \rangle + \frac{L\alpha^2 G^2(\eta_\tau^{(t)})^2}{2}\right].$$

where ① holds since $\mathbb{E}[\boldsymbol{g}_{\tau-1}^{(t)}] = \nabla F_{\mathcal{S}}(\boldsymbol{v}_{\tau-1}^{(t)})$, ② holds by assuming each individual loss is $G$-Lipschitz, and the fact that the function $F_{\mathcal{S}}(\boldsymbol{\theta})$ satisfies PL condition

$$2\mu(F_{\mathcal{S}}(\boldsymbol{\theta}) - F_{\mathcal{S}}(\boldsymbol{\theta}_{\mathcal{S}}^*)) \leq \|\nabla F_{\mathcal{S}}(\boldsymbol{\theta})\|^2.$$

Next, we can upper bound

$$\langle \nabla F_{\mathcal{S}}(\boldsymbol{u}_\tau^{(t)}), \nabla F_{\mathcal{S}}(\boldsymbol{u}_\tau^{(t)}) - \nabla F_{\mathcal{S}}(\boldsymbol{v}_\tau^{(t)}) \rangle \leq \|\nabla F_{\mathcal{S}}(\boldsymbol{u}_\tau^{(t)})\| \cdot \|\nabla F_{\mathcal{S}}(\boldsymbol{u}_\tau^{(t)}) - \nabla F_{\mathcal{S}}(\boldsymbol{v}_\tau^{(t)})\|$$

$$\leq GL\|\boldsymbol{u}_\tau^{(t)} - \boldsymbol{v}_\tau^{(t)}\|_2 \overset{①}{=} (1-\alpha)GL\|\boldsymbol{v}_0^{(t)} - \boldsymbol{v}_\tau^{(t)}\|_2$$

$$\leq (1-\alpha)GL\sum_{i=0}^{\tau}\eta_i^{(t)}\|\boldsymbol{g}_i^{(t)}\|_2 \leq (1-\alpha)LG^2\sum_{i=0}^{\tau}\eta_i^{(t)},$$

where ① uses $\boldsymbol{u}_\tau^{(t)} = \alpha \boldsymbol{v}_\tau^{(t)} + (1-\alpha)\boldsymbol{\theta}_{t-1} = \alpha \boldsymbol{v}_\tau^{(t)} + (1-\alpha)\boldsymbol{v}_0^{(t)}$. In this way, we can upper bound

$$\mathbb{E}\left[F_{\mathcal{S}}(\boldsymbol{u}_{\tau+1}^{(t)}) - F_{\mathcal{S}}(\boldsymbol{\theta}_{\mathcal{S}}^*)\right] \le (1 - 2\mu\alpha\eta_\tau^{(t)})\mathbb{E}\left[F_{\mathcal{S}}(\boldsymbol{u}_\tau^{(t)}) - F_{\mathcal{S}}(\boldsymbol{\theta}_{\mathcal{S}}^*)\right] + \frac{L\alpha^2 G^2(\eta_\tau^{(t)})^2}{2} + \alpha(1-\alpha)LG^2\eta_\tau^{(t)}\sum_{i=0}^{\tau}\eta_i^{(t)}.$$

By setting $\eta_\tau^{(t)} = \frac{1}{\mu((t-1)k+\tau+c)}$ and fixing the time instance $t \ge 1$, we can unwind this recurrence relation from $\tau = k$ to 1 to obtain

$$\mathbb{E}\left[F_{\mathcal{S}}(\boldsymbol{u}_k^{(t)}) - F_{\mathcal{S}}(\boldsymbol{\theta}_{\mathcal{S}}^*)\right]$$

$$\le \prod_{i=0}^{k-1}\left(1 - \frac{2\alpha}{(t-1)k+i+c}\right)\mathbb{E}\left[F_{\mathcal{S}}(\boldsymbol{u}_0^{(t)}) - F_{\mathcal{S}}(\boldsymbol{\theta}_{\mathcal{S}}^*)\right] + \sum_{i=0}^{k-1}\frac{L\alpha^2 G^2}{2\mu^2((t-1)k+i+c)^2}\prod_{j=i}^{k-1}\left(1 - \frac{2\alpha}{(t-1)k+j+c}\right)$$

$$+ \sum_{i=0}^{k-1}\left[\frac{\alpha(1-\alpha)LG^2}{\mu((t-1)k+i+c)}\sum_{j=0}^{i}\frac{1}{\mu((t-1)k+j+c)}\right]\prod_{j=i}^{k-1}\left(1 - \frac{2\alpha}{(t-1)k+j+c}\right)$$

$$\overset{①}{\le} \prod_{i=0}^{k-1}\exp\left\{-\frac{2\alpha}{(t-1)k+i+c}\right\}\mathbb{E}\left[F_{\mathcal{S}}(\boldsymbol{v}_0^{(t)}) - F_{\mathcal{S}}(\boldsymbol{\theta}_{\mathcal{S}}^*)\right] + \sum_{i=0}^{k-1}\prod_{j=i}^{k-1}\exp\left\{-\frac{2\alpha}{(t-1)k+j+c}\right\}\frac{L\alpha^2 G^2}{2\mu^2((t-1)k+i+c)^2}$$

$$+ \sum_{i=0}^{k-1}\left[\frac{\alpha(1-\alpha)LG^2}{\mu((t-1)k+i+c)}\sum_{j=0}^{i}\frac{1}{\mu((t-1)k+j+c)}\right]\exp\left(-\sum_{j=i}^{k-1}\frac{2\alpha}{(t-1)k+j+c}\right)$$

where in ① we have used $1 + x \le e^x$. Now we consider each term in the above inequality as follows. We first bound the first term:

$$\prod_{i=0}^{k-1}\exp\left\{-\frac{2\alpha}{(t-1)k+i+c}\right\}\mathbb{E}\left[F_{\mathcal{S}}(\boldsymbol{u}_0^{(t)}) - F_{\mathcal{S}}(\boldsymbol{\theta}_{\mathcal{S}}^*)\right]$$

$$= \exp\left\{-\sum_{i=0}^{k-1}\frac{2\alpha}{(t-1)k+i+c}\right\}\mathbb{E}\left[F_{\mathcal{S}}(\boldsymbol{u}_0^{(t)}) - F_{\mathcal{S}}(\boldsymbol{\theta}_{\mathcal{S}}^*)\right]$$

$$\overset{①}{\le} \exp\left\{-2\alpha\log\left(\frac{tk+c}{(t-1)k+c}\right)\right\}\mathbb{E}\left[F_{\mathcal{S}}(\boldsymbol{u}_0^{(t)}) - F_{\mathcal{S}}(\boldsymbol{\theta}_{\mathcal{S}}^*)\right]$$

$$= \left(\frac{(t-1)k+c}{tk+c}\right)^{2\alpha}\mathbb{E}\left[F_{\mathcal{S}}(\boldsymbol{u}_0^{(t)}) - F_{\mathcal{S}}(\boldsymbol{\theta}_{\mathcal{S}}^*)\right]$$

where in ① we have used $\sum_{j=i}^{k}\frac{1}{a+j+1} \ge \int_i^{k+1}\frac{1}{a+s+1}ds = \log\left(\frac{a+k+2}{a+i+1}\right)$. Next, we bound the second term:

$$\sum_{i=0}^{k-1}\prod_{j=i}^{k-1}\exp\left\{-\frac{2\alpha}{(t-1)k+j+c}\right\}\frac{L\alpha^2 G^2}{2\mu^2((t-1)k+i+c)^2}$$

$$= \sum_{i=0}^{k-1}\exp\left\{-\sum_{j=i}^{k-1}\frac{2\alpha}{(t-1)k+j+c}\right\}\frac{L\alpha^2 G^2}{2\mu^2((t-1)k+i+c)^2}$$

$$\overset{①}{\le} \sum_{i=0}^{k-1}\exp\left\{-2\alpha\log\left(\frac{tk+c}{(t-1)k+i+c}\right)\right\}\frac{L\alpha^2 G^2}{2\mu^2((t-1)k+i+c)^2}$$

$$\le \frac{L\alpha^2 G^2}{2\mu^2(tk+c)^{2\alpha}}\sum_{i=1}^{k}\frac{((t-1)k+i+c)^{2\alpha}}{((t-1)k+i+c)^2} \le \frac{kL\alpha^2 G^2}{2\mu^2(tk+c)^{2\alpha}}$$

where in ① we have used $\sum_{j=i}^{k} \frac{1}{a+j+1} \geq \int_{i}^{k+1} \frac{1}{a+s+1} ds = \log\left(\frac{a+k+2}{a+i+1}\right)$. Now we only need to bound the third term:

$$\sum_{i=0}^{k-1} \left[ \frac{\alpha(1-\alpha)LG^2}{\mu^2((t-1)k+i+c)} \sum_{j=0}^{i} \frac{1}{((t-1)k+j+c)} \right] \exp\left( \sum_{j=i}^{k-1} -\frac{2\alpha}{(t-1)k+j+c} \right)$$

$$\leq \sum_{i=0}^{k-1} \left[ \frac{\alpha(1-\alpha)LG^2}{\mu^2((t-1)k+i+c)} \frac{i+1}{((t-1)k+c)} \right] \exp\left( \sum_{j=i}^{k-1} -\frac{2\alpha}{(t-1)k+j+c} \right)$$

$$\leq \sum_{i=0}^{k-1} \left[ \frac{\alpha(1-\alpha)LG^2}{\mu^2((t-1)k+i+c)} \frac{i+1}{((t-1)k+c)} \right] \exp\left( \sum_{j=i}^{k-1} -\frac{2\alpha}{(t-1)k+j+c} \right)$$

$$\leq \sum_{i=0}^{k-1} \left[ \frac{\alpha(1-\alpha)LG^2}{\mu^2((t-1)k+i+c)} \frac{i+1}{((t-1)k+c)} \right] \exp\left\{ -2\alpha \log\left( \frac{tk+c}{(t-1)k+i+c} \right) \right\}$$

$$\leq \frac{\alpha(1-\alpha)LG^2}{\mu^2} \sum_{i=0}^{k-1} \left[ \frac{i+1}{((t-1)k+i+c)((t-1)k+c)} \left( \frac{(t-1)k+i+c}{tk+c} \right)^{2\alpha} \right]$$

$$\leq \frac{2\alpha(1-\alpha)LG^2 k(k+1)}{\mu^2(tk+c)^{2\alpha}}.$$

By combining the above results, one can obtain

$$\mathbb{E}\left[ F_{\mathcal{S}}(\boldsymbol{u}_k^{(t)}) - F_{\mathcal{S}}(\boldsymbol{\theta}_{\mathcal{S}}^*) \right]$$

$$\leq \left( \frac{(t-1)k+c}{tk+c} \right)^{2\alpha} \mathbb{E}\left[ F_{\mathcal{S}}(\boldsymbol{u}_0^{(t)}) - F_{\mathcal{S}}(\boldsymbol{\theta}_{\mathcal{S}}^*) \right] + \frac{kL\alpha^2 G^2}{2\mu^2(tk+c)^{2\alpha}} + \frac{2\alpha(1-\alpha)LG^2 k(k+1)}{\mu^2(tk+c)^{2\alpha}}$$

At the same time, we have $\boldsymbol{\theta}_t = \boldsymbol{u}_k^{(t)}$ and $\boldsymbol{\theta}_{t-1} = \boldsymbol{u}_0^{(t)}$, because $\boldsymbol{u}_\tau^{(t)} = \alpha \boldsymbol{v}_\tau^{(t)} + (1-\alpha)\boldsymbol{\theta}_{t-1} = \alpha \boldsymbol{v}_\tau^{(t)} + (1-\alpha)\boldsymbol{v}_0^{(t)}$. In this way, we have

$$\mathbb{E}\left[ F_{\mathcal{S}}(\boldsymbol{\theta}_t) - F_{\mathcal{S}}(\boldsymbol{\theta}_{\mathcal{S}}^*) \right]$$

$$\leq \left( \frac{(t-1)k+c}{tk+c} \right)^{2\alpha} \mathbb{E}\left[ F_{\mathcal{S}}(\boldsymbol{\theta}_{t-1}) - F_{\mathcal{S}}(\boldsymbol{\theta}_{\mathcal{S}}^*) \right] + \frac{kL\alpha^2 G^2}{2\mu^2(tk+c)^{2\alpha}} + \frac{2\alpha(1-\alpha)LG^2 k(k+1)}{\mu^2(tk+c)^{2\alpha}}$$

$$\leq \prod_{i=1}^{t} \left( \frac{(i-1)k+c}{ik+c} \right)^{2\alpha} \mathbb{E}\left[ F_{\mathcal{S}}(\boldsymbol{u}_0^{(t)}) - F_{\mathcal{S}}(\boldsymbol{\theta}_{\mathcal{S}}^*) \right]$$

$$+ \sum_{i=1}^{t} \left[ \frac{kL\alpha^2 G^2}{2\mu^2(ik+c)^{2\alpha}} + \frac{2\alpha(1-\alpha)LG^2 k(k+1)}{\mu^2(ik+c)^{2\alpha}} \right] \prod_{j=i+1}^{t} \left( \frac{(j-1)k+c}{jk+c} \right)^{2\alpha}$$

$$\leq \frac{4}{(tk+c)^{2\alpha}} \mathbb{E}\left[ F_{\mathcal{S}}(\boldsymbol{\theta}_0) - F_{\mathcal{S}}(\boldsymbol{\theta}_{\mathcal{S}}^*) \right]$$

$$+ \frac{2kLG^2}{\mu^2(tk+c)^{2\alpha}} \sum_{i=1}^{t} \left[ \alpha^2 + 2\alpha(1-\alpha)(k-1) \right]$$

$$\leq \frac{4}{(tk+c)^{2\alpha}} \mathbb{E}\left[ F_{\mathcal{S}}(\boldsymbol{\theta}_0) - F_{\mathcal{S}}(\boldsymbol{\theta}_{\mathcal{S}}^*) \right] + \frac{2\alpha LG^2 (\alpha + 2(1-\alpha)(k-1))}{\mu^2(tk+c)^{2\alpha-1}},$$

where $c = 1$.

**Step 2. Generalization error via uniform stability.** Here we aim to use uniform stability to upper bound the generalization error. Suppose given $n$ samples $\mathcal{S} = \{\boldsymbol{z}_1, \boldsymbol{z}_2, \cdots, \boldsymbol{z}_n\}$ where $\boldsymbol{z}_i = (\boldsymbol{x}_i, \boldsymbol{y}_i)$ is sampled from an unknown distribution $\mathcal{D}$, one usually analyze the stability of an algorithm by replacing one sample in $\mathcal{S}$ by another sample from $\mathcal{D}$. Suppose the generated sample set $\mathcal{S}^{(i)} = \{\boldsymbol{z}_1', \boldsymbol{z}_2', \cdots, \boldsymbol{z}_n'\} = \{\boldsymbol{z}_1, \boldsymbol{z}_2, \cdots, \boldsymbol{z}_{i-1}, \boldsymbol{z}_i', \boldsymbol{z}_{i+1} \cdots, \boldsymbol{z}_n\}$ which only differs from the set $\mathcal{S}$ with the $i$-th sample. Then based on these two set, one can train the algorithm to obtain different

solution $\boldsymbol{\theta}$ of the function $F_{\mathcal{S}}(\boldsymbol{\theta})$. When using $\mathcal{S}^{(i)}$, we use $\widetilde{\boldsymbol{\theta}}_t$ and $\tilde{\boldsymbol{v}}_\tau^{(t)}$ to denote their corresponding versions $\boldsymbol{\theta}_t$ and $\boldsymbol{v}_\tau^{(t)}$ in Algorithm 1 trained on $\mathcal{S}$. Next, we can define

$$\boldsymbol{\delta}_\tau^{(t)} = \|\boldsymbol{u}_\tau^{(t)} - \widetilde{\boldsymbol{u}}_\tau^{(t)}\|_2.$$

Then for each iteration $(t, \tau)$ in Algorithm 1, with probability $1 - \frac{1}{n}$, the current selected samples in $\mathcal{S}$ and $\mathcal{S}^{(i)}$ are the same. Note, we update $\boldsymbol{u}_{\tau+1}^{(t)}$ as follows:

$$\boldsymbol{u}_{\tau+1}^{(t)} = \alpha \boldsymbol{v}_\tau^{(t)} + (1 - \alpha)\boldsymbol{v}_0^{(t)} = \boldsymbol{u}_\tau^{(t)} - \alpha \eta_\tau^{(t)} \boldsymbol{g}_\tau^{(t)},$$

where $\boldsymbol{g}_\tau^{(t)}$ denotes the stochastic gradient at the point $\boldsymbol{v}_\tau^{(t)}$. In this case, by using the results in the first part of Lemma 1, we know that $\|\boldsymbol{u}_{\tau+1}^{(t)} - \widetilde{\boldsymbol{u}}_{\tau+1}^{(t)}\|_2 \leq \left(1 + \alpha \eta_\tau^{(t)} L\right) \|\boldsymbol{u}_\tau^{(t)} - \widetilde{\boldsymbol{u}}_\tau^{(t)}\|_2$. Meanwhile with probability $\frac{1}{n}$, the selected samples are different in which we can use Lemma 1: $\|\boldsymbol{u}_{\tau+1}^{(t)} - \widetilde{\boldsymbol{u}}_{\tau+1}^{(t)}\|_2 \leq \|\boldsymbol{u}_\tau^{(t)} - \widetilde{\boldsymbol{u}}_\tau^{(t)}\|_2 + 2\alpha \eta_\tau^{(t)} G$. So combining these two cases yields

$$\mathbb{E}\left[\boldsymbol{\delta}_{\tau+1}^{(t)}\right] = \left(1 - \frac{1}{n}\right)\left(1 + \eta_\tau^{(t)}\alpha L\right)\mathbb{E}\left[\boldsymbol{\delta}_\tau^{(t)}\right] + \frac{1}{n}\mathbb{E}\left[\boldsymbol{\delta}_{\tau+1}^{(t)}\right] + \frac{2\alpha \eta_\tau^{(t)} G}{n}$$

$$= \left(1 + \left(1 - \frac{1}{n}\right)\alpha \eta_\tau^{(t)} L\right)\mathbb{E}\left[\boldsymbol{\delta}_\tau^{(t)}\right] + \frac{2\alpha \eta_\tau^{(t)} G}{n}$$

$$\overset{①}{\leq} \exp\left(\left(1 - \frac{1}{n}\right)\alpha \eta_\tau^{(t)} L\right)\mathbb{E}\left[\boldsymbol{\delta}_\tau^{(t)}\right] + \frac{2\alpha \eta_\tau^{(t)} G}{n}$$

where ① holds by using $1 + x \leq \exp(x)$. At the same time, because $\boldsymbol{u}_\tau^{(t)} = \alpha \boldsymbol{v}_\tau^{(t)} + (1 - \alpha)\boldsymbol{\theta}_{t-1} = \alpha \boldsymbol{v}_\tau^{(t)} + (1 - \alpha)\boldsymbol{v}_0^{(t)}$, we have $\boldsymbol{\theta}_t = \boldsymbol{u}_k^{(t)}$, $\boldsymbol{\theta}_{t-1} = \boldsymbol{u}_0^{(t)}$, and

$$\boldsymbol{u}_{\tau+1}^{(t)} = \alpha \boldsymbol{v}_\tau^{(t)} + (1 - \alpha)\boldsymbol{v}_0^{(t)} = \boldsymbol{u}_\tau^{(t)} - \alpha \eta_\tau^{(t)} \boldsymbol{g}_\tau^{(t)},$$

where $\boldsymbol{g}_\tau^{(t)}$ denotes the stochastic gradient at the point $\boldsymbol{v}_\tau^{(t)}$. In this way, we have

$$\mathbb{E}\left[\|\boldsymbol{\theta}_t - \widetilde{\boldsymbol{\theta}}_t\|\right] = \mathbb{E}\left[\boldsymbol{\delta}_k^{(t)}\right]$$

$$\leq \exp\left(\left(1 - \frac{1}{n}\right)\alpha \sum_{i=0}^{k-1} \eta_i^{(t)} L\right)\mathbb{E}\left[\boldsymbol{\delta}_0^{(t)}\right] + \frac{2\alpha G}{n}\sum_{i=0}^{k-1} \eta_i^{(t)} \exp\left(\left(1 - \frac{1}{n}\right)\alpha \sum_{j=i+1}^{k-1} \eta_j^{(t)} L\right)$$

$$= \exp\left(\left(1 - \frac{1}{n}\right)\alpha \sum_{i=0}^{k-1} \eta_i^{(t)} L\right)\mathbb{E}\left[\|\boldsymbol{\theta}_{t-1} - \widetilde{\boldsymbol{\theta}}_{t-1}\|\right] + \frac{2\alpha G}{n}\sum_{i=0}^{k-1} \eta_i^{(t)} \exp\left(\left(1 - \frac{1}{n}\right)\alpha \sum_{j=i+1}^{k-1} \eta_j^{(t)} L\right)$$

For brevity, define $\eta_{ik+j} = \eta_j^{(i)}$. Then we can reformulate the above equation as follows:

$$\mathbb{E}\left[\|\boldsymbol{\theta}_t - \widetilde{\boldsymbol{\theta}}_t\|\right] \leq \exp\left(\left(1 - \frac{1}{n}\right)\alpha \sum_{i=t_0 k+\tau_0}^{tk} \eta_i L\right)\mathbb{E}\left[\|\boldsymbol{\delta}_{\tau_0}^{(t_0)}\|\right] + \frac{2\alpha G}{n}\sum_{i=t_0 k+\tau_0+1}^{tk} \eta_i \exp\left(\left(1 - \frac{1}{n}\right)\alpha \sum_{j=i+1}^{tk} \eta_j L\right)$$

$$\overset{①}{=} \frac{2G}{\mu n}\sum_{i=t_0 k+\tau_0+1}^{tk} \frac{1}{i} \exp\left(\left(1 - \frac{1}{n}\right)\frac{L}{\mu}\sum_{j=i+1}^{tk} \frac{1}{j}\right)$$

$$\leq \frac{2G}{\mu n}\sum_{i=t_0 k+\tau_0+1}^{tk} \frac{1}{i} \exp\left(\left(1 - \frac{1}{n}\right)\frac{L}{\mu} \log \frac{tk}{i}\right)$$

$$\leq \frac{2G(tk)^{\left(1-\frac{1}{n}\right)\frac{L}{\mu}}}{\mu n}\sum_{i=t_0 k+\tau_0+1}^{tk} \frac{1}{(i)^{\left(1-\frac{1}{n}\right)\frac{L}{\mu}+1}}$$

$$\leq \frac{2G}{\mu(n-1)}\left(\frac{tk}{t_0 k + \tau_0}\right)^{\left(1-\frac{1}{n}\right)\frac{L}{\mu}}$$

where ① holds by setting $\eta_\tau^{(t)} = \frac{1}{\alpha\mu((t-1)k+\tau+c)}$ with $c = 1$.

By setting $t_0 k + \tau_0 = \left[\frac{2\alpha G^2}{\mu\ell_{\max}}(tk)^{\left(1-\frac{1}{n}\right)\frac{\alpha L}{\mu}}\right]^{\frac{1}{1+\left(1-\frac{1}{n}\right)\frac{\alpha L}{\mu}}}$, from Lemma 3 we can obtain

$$\mathbb{E}|\ell(f(\boldsymbol{x};\boldsymbol{u}_\tau^{(t)});\boldsymbol{y}) - \ell(f(\boldsymbol{x};\widetilde{\boldsymbol{u}}_\tau^{(t)});\boldsymbol{y})| \leq \frac{(t_0 k + \tau_0)\ell_{\max}}{n} + \frac{2\alpha G^2}{\mu n}\left(\frac{tk}{t_0 k + \tau_0}\right)^{\left(1-\frac{1}{n-1}\right)\frac{\alpha L}{\mu}}$$

$$\leq \frac{\ell_{\max}^{\frac{\beta}{1+\beta}}}{n-1}\left[\frac{2\alpha G^2}{\mu}\right]^{\frac{1}{1+\beta}}(tk)^{\frac{\beta}{\beta+1}}$$

where $\beta = \left(1-\frac{1}{n}\right)\frac{\alpha L}{\mu}$.

**Step 3. Excess risk error by combining optimization error and generalization error.** Now we combine all results together, including the above optimization error and generalization error, and use Lemma 1 in the manuscript to obtain

$$\varepsilon_{\text{opt}} + \varepsilon_{\text{gen}}$$

$$\leq \frac{4}{(tk+c)^{2\alpha}}\mathbb{E}[F_\mathcal{S}(\boldsymbol{\theta}_0) - F_\mathcal{S}(\boldsymbol{\theta}_\mathcal{S}^*)] + \frac{2\alpha L G^2(\alpha + 2(1-\alpha)(k-1))}{\mu^2(tk+c)^{2\alpha-1}} + \frac{\ell_{\max}^{\frac{\beta}{1+\beta}}}{n-1}\left[\frac{2\alpha G^2}{\mu}\right]^{\frac{1}{1+\beta}}(tk)^{\frac{\beta}{\beta+1}}$$

where $\beta = \left(1-\frac{1}{n}\right)\frac{\alpha L}{\mu}$. The proof is completed. $\qquad\square$

# E    Proof of The Results in Sec. 5

## E.1    Proof of Theorem 5

*Proof.* Here we prove our results in three steps. In the first step, we first prove the optimization error. Then in the second step, we consider to prove the generalization error bound. Finally, we combine these two error bounds by using the risk decomposition.

**Step 1. Optimization error.** Here we consider the following problem:

$$\boldsymbol{\theta}_t^* = \arg\min_{\boldsymbol{\theta}}\left\{F_t(\boldsymbol{\theta}) \triangleq F_\mathcal{S}(\boldsymbol{\theta}) + \frac{\beta_t}{2}\|\boldsymbol{\theta} - \boldsymbol{\theta}_t\|_2^2\right\}.$$

Since $F(\boldsymbol{\theta})$ is $\lambda$-strongly convex and $L$-smooth, then the new function $F_t(\boldsymbol{\theta})$ is $(\lambda + \beta_t)$-strongly convex and $(L + \beta_t)$-smooth. Then from Theorem 3, we have

$$\mathbb{E}[F_t(\boldsymbol{\theta}_t) - F_t(\boldsymbol{\theta})] \leq \frac{\mathbb{E}\left[\|\boldsymbol{\theta}_{t-1} - \boldsymbol{\theta}\|^2\right]}{2\alpha_t\eta_t k_t T_t} + \frac{\eta_t G^2}{2}, \tag{10}$$

where $\boldsymbol{\theta}$ is an arbitrary vector. Note, in the proof of Theorem 3, we let $\boldsymbol{\theta} = \boldsymbol{\theta}_\mathcal{S}^*$ where $\boldsymbol{\theta}_\mathcal{S}^*$ is the optimum. But we can directly follow the proof of Theorem 3, and prove Eqn. (10).

Then we aim to prove $\mathbb{E}[F_\mathcal{S}(\boldsymbol{\theta}_s) - F_\mathcal{S}(\boldsymbol{\theta}_\mathcal{S}^*)] \leq \varepsilon_s$ via selecting proper parameters, where $\varepsilon_s = \frac{\varepsilon_0}{2^s}$. By letting $\boldsymbol{\theta} = \boldsymbol{\theta}_\mathcal{S}^*$, we consider the $(s+1)$-th stage and can directly obtain

$$\mathbb{E}[F(\boldsymbol{\theta}_{s+1}) - F(\boldsymbol{\theta}_\mathcal{S}^*)] \leq -\frac{\beta_{s+1}}{2}\|\boldsymbol{\theta}_{s+1} - \boldsymbol{\theta}_s\|^2 + \frac{\beta_{s+1}}{2}\|\boldsymbol{\theta}_\mathcal{S}^* - \boldsymbol{\theta}_s\|^2 + \frac{\mathbb{E}\left[\|\boldsymbol{\theta}_s - \boldsymbol{\theta}_\mathcal{S}^*\|^2\right]}{2\alpha_s\eta_s k_s T_s} + \frac{\eta_s G^2}{2}$$

$$\leq \frac{\beta_{s+1}}{\lambda}\mathbb{E}[F(\boldsymbol{\theta}_s) - F(\boldsymbol{\theta}_\mathcal{S}^*)] + \frac{\mathbb{E}[F(\boldsymbol{\theta}_s) - F(\boldsymbol{\theta}_\mathcal{S}^*)]}{\mu\alpha_s\eta_s k_s T_s} + \frac{\eta_s G^2}{2}$$

$$\leq \frac{\beta_{s+1}}{\lambda}\varepsilon_s + \frac{\varepsilon_s}{\lambda\alpha_s\eta_s k_s T_s} + \frac{\eta_s G^2}{2}$$

$$\overset{①}{\leq} \varepsilon_{s+1}$$

where ① holds by setting $\beta_{s+1} \leq \frac{1}{6}\lambda, \eta_s \leq \frac{\varepsilon_s}{3G^2}, \eta_s k_s T_s \geq \frac{6}{\lambda\alpha_s}$. This means

$$\mathbb{E}[F(\boldsymbol{\theta}_s) - F(\boldsymbol{\theta}_\mathcal{S}^*)] \leq \frac{\mathbb{E}[F(\boldsymbol{\theta}_0) - F(\boldsymbol{\theta}_\mathcal{S}^*)]}{2^s}.$$

To achieve $\mathbb{E}[F_{\mathcal{S}}(\boldsymbol{\theta}_s) - F_{\mathcal{S}}(\boldsymbol{\theta}_{\mathcal{S}}^*)] \leq \epsilon$, the total stage number $S$ satisfies $S \geq \log \frac{\Delta}{\epsilon}$ where $\Delta = \mathbb{E}[F(\boldsymbol{\theta}_0) - F(\boldsymbol{\theta}_{\mathcal{S}}^*)] = \varepsilon_0$. The total stochastic gradient complexity is $\sum_{s=1}^{S} T_s k_s = \sum_{s=1}^{S} \frac{6}{\lambda \alpha \eta_s} = \sum_{s=1}^{S} \frac{6}{\lambda \alpha} \frac{3G^2 2^s}{\varepsilon_0} = \frac{36G^2}{\lambda \alpha \epsilon}$.

**Step 2. Generalization Error via Stability Analysis**. Now we first consider one stage, such as the $s$-th stage. For brevity, let

$$\boldsymbol{\delta}_\tau^{(t)} = \begin{cases} \|\boldsymbol{v}_0^{(t)} - \tilde{\boldsymbol{v}}_0^{(t)}\|_2 = \|\boldsymbol{\theta}_{t-1} - \widetilde{\boldsymbol{\theta}}_{t-1}\|_2, & \text{if } \tau = 0 \\ \|\boldsymbol{v}_\tau^{(t)} - \tilde{\boldsymbol{v}}_\tau^{(t)}\|_2, & \text{if } \tau \neq 0 \end{cases}$$

Here we first consider each stage and then analyze multi-stage algorithm. We first analyze the $s$-th stage. Now we consider the average case where $\bar{\boldsymbol{v}}_k^{(T)} = \frac{1}{kT} \sum_{i=1}^{T} \sum_{\tau=0}^{k-1} \boldsymbol{v}_\tau^{(t)}$ is the output which is consistent with our optimization analysis which also needs output the average of all $\boldsymbol{v}_\tau^{(t)}$. To begin with, for the regularized function $F_t(\boldsymbol{\theta}) = F(\boldsymbol{\theta}) + \frac{1}{2}\|\boldsymbol{\theta} - \boldsymbol{\theta}_0\|$ where $\boldsymbol{\theta}_0$ denotes the output of the previous stage. Then by using induction, we can easily obtain

$$\boldsymbol{v}_\tau^{(0)} = \boldsymbol{v}_0^{(0)} - \eta \sum_{i=0}^{\tau-1} (1 - \eta\beta)^{\tau-i-1} \boldsymbol{g}_i^{(0)}.$$

This means $\boldsymbol{v}_k^{(0)} = \boldsymbol{v}_0^{(0)} - \eta \sum_{i=0}^{k-1} (1 - \eta\beta)^{\tau-i-1} \boldsymbol{g}_i^{(0)}$. Then because $\boldsymbol{v}_0^{(0)} = \boldsymbol{\theta}_0$, for $\boldsymbol{\theta}_1$, we have

$$\boldsymbol{v}_0^{(1)} = \boldsymbol{\theta}_1 = (1 - \alpha)\boldsymbol{\theta}_0 + \alpha\boldsymbol{v}_k^{(0)} = \boldsymbol{v}_0^{(0)} - \alpha\eta \sum_{i=0}^{k-1} (1 - \eta\beta)^{k-i-1} \boldsymbol{g}_i^{(0)}$$

Similarly, we can obtain

$$\boldsymbol{v}_0^{(t+1)} = \boldsymbol{\theta}_t = \boldsymbol{v}_0^{(0)} - \alpha\eta \sum_{j=0}^{t-1} \sum_{i=0}^{k-1} (1 - \eta\beta)^{tk-jk-i-1} \boldsymbol{g}_i^{(j)}.$$

In this way, we can know the formulation of the average of all $\boldsymbol{v}_\tau^{(t)}$ as follows:

$$\begin{aligned} \bar{\boldsymbol{v}}_k^{(T)} &= \frac{1}{kT} \sum_{i=1}^{T} \sum_{\tau=0}^{k-1} \boldsymbol{v}_\tau^{(t)} = \boldsymbol{v}_0^{(0)} - \alpha\eta \sum_{j=0}^{T-1} \sum_{i=0}^{k-1} (1 - \eta\beta)^{Tk-jk-i-1} \boldsymbol{g}_i^{(j)} \\ &= (1 - \eta\beta)\bar{\boldsymbol{v}}_{k-1}^{(T)} + \eta\beta\boldsymbol{v}_0^{(0)} - \alpha\eta\boldsymbol{g}_{k-1}^{(T)} \end{aligned}$$

Assume $\bar{\boldsymbol{v}}_k^{(T)}$ is obtained by running the algorithm on the dataset $\mathcal{S}$ and $\bar{\boldsymbol{v}}_k^{('T)}$ denotes the solution obtained by running the algorithm on $\mathcal{S}'$. In this way, we can conclude that for any $t$, we have

$$\bar{\boldsymbol{v}}_k^{(t)} = \boldsymbol{v}_k^{(t)}, \qquad \bar{\boldsymbol{v}}_k^{('t)} = \tilde{\boldsymbol{v}}_k^{(t)}, \qquad \|\bar{\boldsymbol{v}}_k^{(t)} - \bar{\boldsymbol{v}}_k^{('t)}\| = \|\boldsymbol{v}_k^{(t)} - \tilde{\boldsymbol{v}}_k^{(t)}\|. \qquad (11)$$

In the following, we try to bound the difference between $\|\bar{\boldsymbol{v}}_k^{(T)} - \bar{\boldsymbol{v}}_k^{('T)}\|$. For each iteration $(t, \tau)$ in Algorithm 1, with probability $1 - \frac{1}{n}$, the current selected samples in $\mathcal{S}$ and $\mathcal{S}^{(i)}$ are the same. In this case, by using the first part of Lemma 1, we know that

$$\begin{aligned} &\|\bar{\boldsymbol{v}}_{\tau+1}^{(t)} - \bar{\boldsymbol{v}}_{\tau+1}^{('t)}\|_2 \\ =&\|(1 - \eta\beta)\bar{\boldsymbol{v}}_\tau^{(t)} + \eta\beta\boldsymbol{v}_0^{(0)} - \alpha\eta\boldsymbol{g}_\tau^{(t)} - (1 - \eta\beta)\bar{\boldsymbol{v}}_\tau^{('t)} - \eta\beta\boldsymbol{v}_0^{(0)} + \alpha\eta\boldsymbol{g}_\tau^{('t)}\|_2 \\ =&(1 - \eta\beta) \left\| \bar{\boldsymbol{v}}_\tau^{(t)} - \frac{\alpha\eta}{1 - \eta\beta}\boldsymbol{g}_\tau^{(t)} - \bar{\boldsymbol{v}}_\tau^{('t)} + \frac{\alpha\eta}{1 - \eta\beta}\boldsymbol{g}_\tau^{('t)} \right\|_2 \\ \overset{\text{①}}{\leq}&(1 - \eta\beta)\left(1 - \frac{\alpha\eta}{1 - \eta\beta}\frac{\lambda L}{\lambda + L}\right)\|\boldsymbol{v}_\tau^{(t)} - \tilde{\boldsymbol{v}}_\tau^{(t)}\|_2 \end{aligned}$$

where ① uses the first part of Lemma 1. Meanwhile with probability $\frac{1}{n}$, the selected samples are different in which we can use the first part of Lemma 1:

$$\|\boldsymbol{v}_{\tau+1}^{(t)} - \tilde{\boldsymbol{v}}_{\tau+1}^{(t)}\|_2$$

$$=(1-\eta\beta)\left\|\bar{\boldsymbol{v}}_\tau^{(t)} - \frac{\alpha\eta}{1-\eta\beta}\boldsymbol{g}_\tau^{(t)} - \bar{\boldsymbol{v}}_\tau^{('t)} + \frac{\alpha\eta}{1-\eta\beta}\boldsymbol{g}_\tau^{('t)}\right\|_2$$

$$\overset{①}{\leq}(1-\eta\beta)\left(1-\frac{\alpha\eta}{1-\eta\beta}\frac{\lambda L}{\lambda+L}\right)\|\boldsymbol{v}_\tau^{(t)} - \tilde{\boldsymbol{v}}_\tau^{(t)}\|_2 + (1-\eta\beta)\frac{2\alpha\eta G}{1-\eta\beta}$$

$$=(1-\eta\beta)\left(1-\frac{\alpha\eta}{1-\eta\beta}\frac{\lambda L}{\lambda+L}\right)\|\boldsymbol{v}_\tau^{(t)} - \tilde{\boldsymbol{v}}_\tau^{(t)}\|_2 + 2\alpha\eta G$$

where ① holds by using the first part of Lemma 1. So combining these two cases yields

$$\mathbb{E}\left[\boldsymbol{\delta}_{\tau+1}^{(t)}\right] = \|\boldsymbol{v}_{\tau+1}^{(t)} - \tilde{\boldsymbol{v}}_{\tau+1}^{(t)}\|_2$$

$$\leq \left(1-\frac{1}{n}\right)\left[(1-\eta\beta)\left(1-\frac{\alpha\eta}{1-\eta\beta}\frac{\lambda L}{\lambda+L}\right)\|\boldsymbol{v}_\tau^{(t)} - \tilde{\boldsymbol{v}}_\tau^{(t)}\|_2\right]$$

$$+\frac{1}{n}\left[(1-\eta\beta)\left(1-\frac{\alpha\eta}{1-\eta\beta}\frac{\lambda L}{\lambda+L}\right)\|\boldsymbol{v}_\tau^{(t)} - \tilde{\boldsymbol{v}}_\tau^{(t)}\|_2 + 2\alpha\eta G\right]$$

$$=\left(1-\eta\beta-\frac{\alpha\eta\lambda L}{\lambda+L}\right)\mathbb{E}\boldsymbol{\delta}_\tau^{(t)} + \frac{2\alpha\eta G}{n}$$

$$=\left(1-\eta\beta-\frac{\alpha\eta\lambda L}{\lambda+L}\right)^{(t-1)k+\tau+1}\mathbb{E}\left[\boldsymbol{\delta}_0^{(0)}\right] + \frac{2\alpha\eta G}{n}\sum_{j=0}^{t-1}\sum_{i=0}^{\tau}\left(1-\eta\beta-\frac{\alpha\eta\lambda L}{\lambda+L}\right)^{jk+i}$$

$$=\left(1-\eta\beta-\frac{\alpha\eta\lambda L}{\lambda+L}\right)^{(t-1)k+\tau+1}\mathbb{E}\left[\boldsymbol{\delta}_0^{(0)}\right] + \frac{2\alpha G}{n}\frac{1}{\beta+\frac{\alpha\lambda L}{\lambda+L}}\left[1-\left(1-\eta\beta-\frac{\alpha\eta\lambda L}{\lambda+L}\right)^{(t-1)k+\tau+1}\right].$$

Then by using (11), we have that for the output of the $s$-th stage, it holds

$$\mathbb{E}\left[\|\boldsymbol{\theta}_{s-\text{stage}} - \widetilde{\boldsymbol{\theta}}_{s-\text{stage}}\|_2\right] = \mathbb{E}\left[\|\boldsymbol{\theta}_{T_s} - \widetilde{\boldsymbol{\theta}}_{T_s}\|_2\right] = \mathbb{E}\left[\|\boldsymbol{v}_k^{(T_s)} - \tilde{\boldsymbol{v}}_k^{(T_s)}\|_2\right]$$

$$\leq \left(1-\eta_s\beta_s-\frac{\alpha\eta_s\lambda L}{\lambda+L}\right)^{T_s k_s}\mathbb{E}\left[\|\boldsymbol{\theta}_{(s-1)-\text{stage}} - \widetilde{\boldsymbol{\theta}}_{(s-1)-\text{stage}}\|\right] + \frac{2\alpha G}{n}\frac{1}{\beta+\frac{\alpha\lambda L}{\lambda+L}}\left[1-\left(1-\eta\beta-\frac{\alpha\eta\lambda L}{\lambda+L}\right)^{T_s k_s}\right].$$

Assume that at stage $s$, we have $\mathbb{E}\left[\|\boldsymbol{\theta}_{s-\text{stage}} - \widetilde{\boldsymbol{\theta}}_{s-\text{stage}}\|_2\right] = 0$. Then it holds that

$$\mathbb{E}\left[\|\boldsymbol{\theta}_{s-\text{stage}} - \widetilde{\boldsymbol{\theta}}_{s-\text{stage}}\|_2\right]$$

$$\leq \left(1-\eta_s\beta_s-\frac{\alpha\eta_s\lambda L}{\lambda+L}\right)^{T_s k_s}\mathbb{E}\left[\|\boldsymbol{\theta}_{(s-1)-\text{stage}} - \widetilde{\boldsymbol{\theta}}_{(s-1)-\text{stage}}\|\right] + \frac{2\alpha G}{n}\frac{1}{\beta+\frac{\alpha\lambda L}{\lambda+L}}\left[1-\left(1-\eta\beta-\frac{\alpha\eta\lambda L}{\lambda+L}\right)^{T_s k_s}\right]$$

$$\leq \sum_{i=t_0+1}^{s}\frac{2\alpha G}{n}\frac{1}{\beta_i+\frac{\alpha\lambda L}{\lambda+L}}\left[1-\left(1-\eta_i\beta_i-\frac{\alpha\eta_i\lambda L}{\lambda+L}\right)^{T_i k_i}\right]\prod_{j=i+1}^{s}\left(1-\eta_j\beta_j-\frac{\alpha\eta_j\lambda L}{\lambda+L}\right)^{T_j k_j}$$

$$\leq \sum_{i=t_0+1}^{s}\frac{2\alpha G}{n}\frac{1}{\beta_i+\frac{\alpha\lambda L}{\lambda+L}}\left[1-\left(1-\eta_i\beta_i-\frac{\alpha\eta_i\lambda L}{\lambda+L}\right)^{T_i k_i}\right]\prod_{j=i+1}^{s}\exp\left(-\eta_j\beta_j T_j k_j - \frac{\alpha\eta_j T_j k_j\lambda L}{\lambda+L}\right)$$

$$\overset{①}{\leq}\sum_{i=t_0+1}^{s}\frac{2\alpha G}{n}\frac{1}{\beta_i+\frac{\alpha\lambda L}{\lambda+L}}\left[1-\left(1-\eta_i\beta_i-\frac{\alpha\eta_i\lambda L}{\lambda+L}\right)^{T_i k_i}\right]\exp\left(-\frac{6L(s-j)}{\lambda+L}\right)$$

$$\leq \sum_{i=t_0+1}^{s}\frac{2\alpha G}{n}\frac{1}{\beta_i+\frac{\alpha\lambda L}{\lambda+L}}\exp\left(-\frac{6L(s-j)}{\lambda+L}\right)$$

$$\leq \frac{2\alpha G}{n}\frac{1}{\beta+\frac{\alpha\lambda L}{\lambda+L}}\frac{1-\exp\left(-\frac{6sL}{\lambda+L}\right)}{1-\exp\left(-\frac{6L}{\lambda+L}\right)}$$

where ① holds since for optimization, we set $\beta_{s+1} \leq \frac{1}{6}\lambda, \eta_s \leq \frac{\varepsilon_s}{3G^2}, \eta_s k_s T_s \geq \frac{6}{\lambda\alpha_s}$ and thus have $\eta_j\beta_j T_j k_j + \frac{\alpha\eta_j T_j k_j \lambda L}{\lambda+L} \geq \frac{6L}{\lambda+L}$.

On the other hand, we have that function $\ell(f(\cdot; \boldsymbol{\theta}); \cdot)$ is $G$-Lipschitz, and thus obtain

$$\mathbb{E}\left[|\ell(f(\boldsymbol{x}; \boldsymbol{\theta}_T); \boldsymbol{y}) - \ell(f(\boldsymbol{x}; \widetilde{\boldsymbol{\theta}}_T); \boldsymbol{y})|\right] \leq G\mathbb{E}\left[\|\boldsymbol{\theta}_{t+1} - \widetilde{\boldsymbol{\theta}}_{t+1}\|_2\right] \leq \frac{2\alpha G^2}{n\left(\beta + \frac{\alpha\lambda L}{\lambda+L}\right)} \frac{1 - \exp\left(-\frac{6sL}{\lambda+L}\right)}{1 - \exp\left(-\frac{6L}{\lambda+L}\right)}.$$

**Step 3. Excess risk error by combining optimization error and generalization error.** Now we combine all results together, including the above optimization error and generalization error, and use Lemma 1 in the manuscript to obtain

$$\varepsilon_{\text{opt}} + \varepsilon_{\text{gen}} \leq \frac{\Delta'}{2^s} + \frac{2\alpha G^2}{n\left(\beta + \frac{\alpha\lambda L}{\lambda+L}\right)} \frac{1 - \exp\left(-\frac{6sL}{\lambda+L}\right)}{1 - \exp\left(-\frac{6L}{\lambda+L}\right)},$$

where $\Delta' = \mathbb{E}[F(\boldsymbol{\theta}_0) - F(\boldsymbol{\theta}_{\mathcal{S}}^*)]$. The proof is completed. $\qquad\square$

## E.2 Proof of Theorem 6

*Proof.* Here we prove our results in three steps. In the first step, we first prove the optimization error. Then in the second step, we consider to prove the generalization error bound. Finally, we combine these two error bounds by using the risk decomposition.

**Step 1. Optimization error.** Here we consider the following problem:

$$\boldsymbol{\theta}_t^* = \underset{\boldsymbol{\theta}}{\arg\min}\left\{F_t(\boldsymbol{\theta}) \triangleq F_{\mathcal{S}}(\boldsymbol{\theta}) + \frac{\beta_t}{2}\|\boldsymbol{\theta} - \boldsymbol{\theta}_t\|_2^2\right\}.$$

Since $F(\boldsymbol{\theta})$ is $L$-smooth and its Hessian has minimum eigenvalue $-\sigma$ ($\sigma < 0$), then the new function $F_t(\boldsymbol{\theta})$ is $(\beta_t - \sigma)$-strongly convex and $(L + \beta_t)$-smooth, where we set $\beta_t \geq \sigma$.

We first consider the case $\sigma \leq \beta_s \leq \frac{1}{6}\mu$. From Theorem 3, we have

$$\mathbb{E}[F_t(\boldsymbol{\theta}_t) - F_t(\boldsymbol{\theta})] \leq \frac{\mathbb{E}\left[\|\boldsymbol{\theta}_{t-1} - \boldsymbol{\theta}\|^2\right]}{2\alpha_t\eta_t k_t T_t} + \frac{\eta_t G^2}{2}, \tag{12}$$

where $\boldsymbol{\theta}$ is an arbitrary vector. Note, in the proof of Theorem 3, we let $\boldsymbol{\theta} = \boldsymbol{\theta}_{\mathcal{S}}^*$ where $\boldsymbol{\theta}_{\mathcal{S}}^*$ is the optimum. But we can directly follow the proof of Theorem 3, and prove Eqn. (12).

Then we aim to prove $\mathbb{E}[F_{\mathcal{S}}(\boldsymbol{\theta}_s) - F_{\mathcal{S}}(\boldsymbol{\theta}_{\mathcal{S}}^*)] \leq \varepsilon_s$ via selecting proper parameters, where $\varepsilon_s = \frac{\varepsilon_0}{2^s}$. By letting $\boldsymbol{\theta} = \boldsymbol{\theta}_{\mathcal{S}}^*$ and using the PL condition, we consider the $(s + 1)$-th stage and can directly obtain

$$\begin{aligned}
\mathbb{E}[F(\boldsymbol{\theta}_{s+1}) - F(\boldsymbol{\theta}_{\mathcal{S}}^*)] &\leq -\frac{\beta_{s+1}}{2}\|\boldsymbol{\theta}_{s+1} - \boldsymbol{\theta}_s\|^2 + \frac{\beta_{s+1}}{2}\|\boldsymbol{\theta}_{\mathcal{S}}^* - \boldsymbol{\theta}_s\|^2 + \frac{\mathbb{E}\left[\|\boldsymbol{\theta}_s - \boldsymbol{\theta}_{\mathcal{S}}^*\|^2\right]}{2\alpha_s\eta_s k_s T_s} + \frac{\eta_s G^2}{2} \\
&\overset{①}{\leq} \frac{\beta_{s+1}}{\mu}\mathbb{E}[F(\boldsymbol{\theta}_s) - F(\boldsymbol{\theta}_{\mathcal{S}}^*)] + \frac{\mathbb{E}[F(\boldsymbol{\theta}_s) - F(\boldsymbol{\theta}_{\mathcal{S}}^*)]}{\mu\alpha_s\eta_s k_s T_s} + \frac{\eta_s G^2}{2} \\
&\leq \frac{\beta_{s+1}}{\mu}\varepsilon_s + \frac{\varepsilon_s}{\mu\alpha_s\eta_s k_s T_s} + \frac{\eta_s G^2}{2} \\
&\overset{①}{\leq} \varepsilon_{s+1}
\end{aligned}$$

where ① holds by using the result $\|\boldsymbol{\theta} - \boldsymbol{\theta}_{\mathcal{S}}^*\|^2 \leq \frac{1}{2\mu}(F_{\mathcal{S}}(\boldsymbol{\theta}) - F_{\mathcal{S}}(\boldsymbol{\theta}_{\mathcal{S}}^*))$ in Lemma 2, ② holds by setting $\beta_{s+1} \leq \frac{1}{6}\mu, \eta_s \leq \frac{\varepsilon_s}{3G^2}, \eta_s k_s T_s \geq \frac{6}{\mu\alpha_s}$. This means

$$\mathbb{E}[F(\boldsymbol{\theta}_s) - F(\boldsymbol{\theta}_{\mathcal{S}}^*)] \leq \frac{\mathbb{E}[F(\boldsymbol{\theta}_0) - F(\boldsymbol{\theta}_{\mathcal{S}}^*)]}{2^s}.$$

To achieve $\mathbb{E}[F_{\mathcal{S}}(\boldsymbol{\theta}_s) - F_{\mathcal{S}}(\boldsymbol{\theta}_{\mathcal{S}}^*)] \leq \epsilon$, the total stage number $S$ satisfies $S \geq \log\frac{\Delta}{\epsilon}$ where $\Delta = \mathbb{E}[F(\boldsymbol{\theta}_0) - F(\boldsymbol{\theta}_{\mathcal{S}}^*)] = \varepsilon_0$. The total stochastic gradient complexity is $\sum_{s=1}^{S} T_s k_s = \sum_{s=1}^{S} \frac{6}{\lambda\alpha\eta_s} = \sum_{s=1}^{S} \frac{6}{\lambda\alpha}\frac{3G^2 2^s}{\varepsilon_0} = \frac{36G^2}{\lambda\alpha\epsilon}$. For this case, we actually requires $\sigma \leq \beta_s \leq \frac{1}{6}\mu$.

Next, we do consider the second case where we do not require $\sigma \leq \beta_s \leq \frac{1}{6}\mu$ but assuming there is a constant $\rho$ such that $\langle \nabla F_t(\boldsymbol{\theta}), \boldsymbol{\theta} - \boldsymbol{\theta}_{\mathcal{S}}^* \rangle \geq \rho(F_t(\boldsymbol{\theta}) - F_t(\boldsymbol{\theta}_{\mathcal{S}}^*))$. For this case, we first bound

$$
\begin{aligned}
\mathbb{E}\left[\|\boldsymbol{v}_{\tau+1}^{(t)} - \boldsymbol{\theta}_{\mathcal{S}}^*\|_2^2\right] &= \mathbb{E}\left[\|\boldsymbol{v}_{\tau}^{(t)} - \eta_{\tau}^{(t)}\boldsymbol{g}_{\tau}^{(t)} - \boldsymbol{\theta}_{\mathcal{S}}^*\|_2^2\right] \\
&= \mathbb{E}\left[\|\boldsymbol{v}_{\tau}^{(t)} - \boldsymbol{\theta}_{\mathcal{S}}^*\|_2^2 - 2\eta_{\tau}^{(t)}\langle \boldsymbol{v}_{\tau}^{(t)} - \boldsymbol{\theta}_{\mathcal{S}}^*, \boldsymbol{g}_{\tau}^{(t)}\rangle + (\eta_{\tau}^{(t)})^2\|\boldsymbol{g}_{\tau}^{(t)}\|_2^2\right] \\
&= \mathbb{E}\left[\|\boldsymbol{v}_{\tau}^{(t)} - \boldsymbol{\theta}_{\mathcal{S}}^*\|_2^2 - 2\eta_{\tau}^{(t)}\langle \boldsymbol{v}_{\tau}^{(t)} - \boldsymbol{\theta}_{\mathcal{S}}^*, \nabla F_t(\boldsymbol{v}_{\tau}^{(t)})\rangle + (\eta_{\tau}^{(t)})^2\|\boldsymbol{g}_{\tau}^{(t)}\|_2^2\right] \\
&\leq \mathbb{E}\left[\|\boldsymbol{v}_{\tau}^{(t)} - \boldsymbol{\theta}_{\mathcal{S}}^*\|_2^2 - 2\eta_{\tau}^{(t)}\langle \boldsymbol{v}_{\tau}^{(t)} - \boldsymbol{\theta}_{\mathcal{S}}^*, \nabla F_t(\boldsymbol{v}_{\tau}^{(t)})\rangle + (\eta_{\tau}^{(t)})^2\|\boldsymbol{g}_{\tau}^{(t)}\|_2^2\right] \\
&\leq \mathbb{E}\left[\|\boldsymbol{v}_{\tau}^{(t)} - \boldsymbol{\theta}_{\mathcal{S}}^*\|_2^2 + 2\eta_{\tau}^{(t)}\rho(F_t(\boldsymbol{\theta}_{\mathcal{S}}^*) - F_t(\boldsymbol{v}_{\tau}^{(t)})) + (\eta_{\tau}^{(t)})^2\|\boldsymbol{g}_{\tau}^{(t)}\|_2^2\right] \\
&\leq \mathbb{E}\left[\|\boldsymbol{v}_{\tau}^{(t)} - \boldsymbol{\theta}_{\mathcal{S}}^*\|_2^2 + 2\eta_{\tau}^{(t)}\rho(F_t(\boldsymbol{\theta}_{\mathcal{S}}^*) - F_t\boldsymbol{v}_{\tau}^{(t)})) + (\eta_{\tau}^{(t)})^2 G^2\right]
\end{aligned}
$$

Then by rearranging the above inequality, we can obtain

$$
F_t(\boldsymbol{v}_{\tau}^{(t)}) - F_t(\boldsymbol{\theta}_{\mathcal{S}}^*) \leq \frac{1}{2\eta_{\tau}^{(t)}\rho}\mathbb{E}\left[\|\boldsymbol{v}_{\tau}^{(t)} - \boldsymbol{\theta}_{\mathcal{S}}^*\|_2^2 - \|\boldsymbol{v}_{\tau+1}^{(t)} - \boldsymbol{\theta}_{\mathcal{S}}^*\|_2^2\right] + \frac{G^2}{2\rho}\eta_{\tau}^{(t)}.
$$

Next, by setting a constant learning rate $\eta_{\tau}^{(t)} = \eta$, we sum up the above inequality from $t = 1$ to $t = T$ and obtain

$$
\begin{aligned}
\frac{1}{k}\sum_{\tau=0}^{k-1}\left(F_t(\boldsymbol{v}_{\tau}^{(t)}) - F_t(\boldsymbol{\theta}_{\mathcal{S}}^*)\right) &\leq \frac{1}{2\eta\rho k}\mathbb{E}\left[\|\boldsymbol{v}_0^{(t)} - \boldsymbol{\theta}_{\mathcal{S}}^*\|_2^2 - \|\boldsymbol{v}_k^{(t)} - \boldsymbol{\theta}_{\mathcal{S}}^*\|_2^2\right] + \frac{\eta G^2}{2\rho} \\
&= \frac{1}{2\eta\rho k}\mathbb{E}\left[\|\boldsymbol{\theta}_{t-1} - \boldsymbol{\theta}_{\mathcal{S}}^*\|_2^2 - \|\boldsymbol{v}_k^{(t)} - \boldsymbol{\theta}_{\mathcal{S}}^*\|_2^2\right] + \frac{\eta G^2}{2\rho}.
\end{aligned}
$$

Now we consider the term $\|\boldsymbol{\theta}_{t-1} - \boldsymbol{\theta}_{\mathcal{S}}^*\|_2^2 - \|\boldsymbol{v}_k^{(t)} - \boldsymbol{\theta}_{\mathcal{S}}^*\|_2^2$ as follows:

$$
\begin{aligned}
\|\boldsymbol{\theta}_{t-1} - \boldsymbol{\theta}_{\mathcal{S}}^*\|_2^2 - \|\boldsymbol{v}_k^{(t)} - \boldsymbol{\theta}_{\mathcal{S}}^*\|_2^2 &= \left\langle \boldsymbol{\theta}_{t-1} - \boldsymbol{v}_k^{(t)}, \boldsymbol{\theta}_{t-1} + \boldsymbol{v}_k^{(t)} - 2\boldsymbol{\theta}_{\mathcal{S}}^*\right\rangle \\
&\overset{①}{=} \left\langle \boldsymbol{\theta}_{t-1} - \frac{\boldsymbol{\theta}_t - (1-\alpha)\boldsymbol{\theta}_{t-1}}{\alpha}, \boldsymbol{\theta}_{t-1} + \frac{\boldsymbol{\theta}_t - (1-\alpha)\boldsymbol{\theta}_{t-1}}{\alpha} - 2\boldsymbol{\theta}_{\mathcal{S}}^*\right\rangle \\
&= -\frac{1}{\alpha^2}\left\langle \boldsymbol{\theta}_{t-1} - \boldsymbol{\theta}_{\mathcal{S}}^* - (\boldsymbol{\theta}_t - \boldsymbol{\theta}_{\mathcal{S}}^*), (1-2\alpha)(\boldsymbol{\theta}_{t-1} - \boldsymbol{\theta}_{\mathcal{S}}^*) - (\boldsymbol{\theta}_t - \boldsymbol{\theta}_{\mathcal{S}}^*)\right\rangle \\
&= -\frac{1}{\alpha^2}\left[(1-\alpha)\|\boldsymbol{\theta}_{t-1} - \boldsymbol{\theta}_t\|^2 + \alpha\|\boldsymbol{\theta}_t - \boldsymbol{\theta}_{\mathcal{S}}^*\|^2 - \alpha\|\boldsymbol{\theta}_{t-1} - \boldsymbol{\theta}_{\mathcal{S}}^*\|^2\right] \\
&\leq \frac{1}{\alpha}\left[\|\boldsymbol{\theta}_{t-1} - \boldsymbol{\theta}_{\mathcal{S}}^*\|^2 - \|\boldsymbol{\theta}_t - \boldsymbol{\theta}_{\mathcal{S}}^*\|^2\right]
\end{aligned}
$$

where ① holds since $\boldsymbol{\theta}_t = (1-\alpha)\boldsymbol{\theta}_{t-1} + \alpha\boldsymbol{v}_k^{(t)}$. In this way, we can upper bound

$$
\frac{1}{k}\sum_{\tau=0}^{k-1}\left(F_t(\boldsymbol{v}_{\tau}^{(t)}) - F_t(\boldsymbol{\theta}_{\mathcal{S}}^*)\right) \leq \frac{1}{2\alpha\rho\eta k}\mathbb{E}\left[\|\boldsymbol{\theta}_{t-1} - \boldsymbol{\theta}_{\mathcal{S}}^*\|^2 - \|\boldsymbol{\theta}_t - \boldsymbol{\theta}_{\mathcal{S}}^*\|^2\right] + \frac{\eta G^2}{2\rho}.
$$

Finally, we can sum up from $t = 1$ to $t = T$ and obtain

$$
\frac{1}{kT}\sum_{t=1}^{T}\sum_{\tau=0}^{k-1}\left(F_t(\boldsymbol{v}_{\tau}^{(t)}) - F_t(\boldsymbol{\theta}_{\mathcal{S}}^*)\right) \leq \frac{1}{2\alpha\rho\eta kT}\mathbb{E}\left[\|\boldsymbol{\theta}_0 - \boldsymbol{\theta}_{\mathcal{S}}^*\|^2\right] + \frac{\eta G^2}{2\rho}
$$

Then by letting $\boldsymbol{\theta}_{s+1} = \frac{1}{T_s k_s}\sum_{t=1}^{T_s}\sum_{\tau=0}^{k_s-1}\boldsymbol{v}_{\tau}^{(t)}$, we have

$$
\mathbb{E}\left(F_s(\boldsymbol{\theta}_{s+1}) - F_s(\boldsymbol{\theta}_{\mathcal{S}}^*)\right) \leq \frac{1}{2\alpha\rho\eta kT}\mathbb{E}\left[\|\boldsymbol{\theta}_0 - \boldsymbol{\theta}_{\mathcal{S}}^*\|^2\right] + \frac{\eta G^2}{2\rho}
$$

Since $F_s(\boldsymbol{\theta}) = F_{\mathcal{S}}(\boldsymbol{\theta}) + \frac{\beta_s}{2}\|\boldsymbol{\theta} - \boldsymbol{\theta}_s\|^2$, we consider the $(s+1)$-th stage and can directly obtain

$$
\begin{aligned}
\mathbb{E}[F_{\mathcal{S}}(\boldsymbol{\theta}_{s+1}) - F_{\mathcal{S}}(\boldsymbol{\theta}_{\mathcal{S}}^*)] &\leq -\frac{\beta_{s+1}}{2}\|\boldsymbol{\theta}_{s+1} - \boldsymbol{\theta}_s\|^2 + \frac{\beta_{s+1}}{2}\|\boldsymbol{\theta}_{\mathcal{S}}^* - \boldsymbol{\theta}_s\|^2 + \frac{\mathbb{E}\left[\|\boldsymbol{\theta}_s - \boldsymbol{\theta}_{\mathcal{S}}^*\|^2\right]}{2\rho\alpha_s\eta_s k_s T_s} + \frac{\eta_s G^2}{2\rho} \\
&\overset{①}{\leq} \frac{\beta_{s+1}}{\mu}\mathbb{E}[F(\boldsymbol{\theta}_s) - F(\boldsymbol{\theta}_{\mathcal{S}}^*)] + \frac{\mathbb{E}[F(\boldsymbol{\theta}_s) - F(\boldsymbol{\theta}_{\mathcal{S}}^*)]}{\mu\rho\alpha_s\eta_s k_s T_s} + \frac{\eta_s G^2}{2\rho} \\
&\leq \frac{\beta_{s+1}}{\mu}\varepsilon_{\mathrm{s}} + \frac{\varepsilon_{\mathrm{s}}}{\mu\rho\alpha_s\eta_s k_s T_s} + \frac{\eta_s G^2}{2\rho} \\
&\overset{①}{\leq} \varepsilon_{\mathrm{s+1}}
\end{aligned}
$$

where ① holds by using the result $\|\boldsymbol{\theta} - \boldsymbol{\theta}_{\mathcal{S}}^*\|^2 \leq \frac{1}{2\mu}(F_{\mathcal{S}}(\boldsymbol{\theta}) - F_{\mathcal{S}}(\boldsymbol{\theta}_{\mathcal{S}}^*))$ in Lemma 2, ② holds by setting $\beta_{s+1} \leq \frac{1}{6}\mu, \eta_s \leq \frac{\rho\varepsilon_s}{3G^2}, \eta_s k_s T_s \geq \frac{6}{\mu\rho\alpha_s}$. This means

$$
\mathbb{E}[F(\boldsymbol{\theta}_s) - F(\boldsymbol{\theta}_{\mathcal{S}}^*)] \leq \frac{\mathbb{E}[F(\boldsymbol{\theta}_0) - F(\boldsymbol{\theta}_{\mathcal{S}}^*)]}{2^s}.
$$

To achieve $\mathbb{E}[F_{\mathcal{S}}(\boldsymbol{\theta}_s) - F_{\mathcal{S}}(\boldsymbol{\theta}_{\mathcal{S}}^*)] \leq \epsilon$, the total stage number $S$ satisfies $S \geq \log\frac{\Delta}{\epsilon}$ where $\Delta = \mathbb{E}[F(\boldsymbol{\theta}_0) - F(\boldsymbol{\theta}_{\mathcal{S}}^*)] = \varepsilon_0$. The total stochastic gradient complexity is $\sum_{s=1}^{S} T_s k_s = \sum_{s=1}^{S}\frac{6}{\mu\rho\alpha\eta_s} = \sum_{s=1}^{S}\frac{6}{\mu\rho\alpha}\frac{3G^2 2^s}{\rho\varepsilon_0} = \frac{36G^2}{\mu\rho^2\alpha\epsilon}$.

**Step 2. Generalization Error via Stability Analysis**. We first consider the case $\sigma \leq \beta_s \leq \frac{1}{6}\mu$. For this case, we know that the new function $F_t(\boldsymbol{\theta})$ is $(\beta_t - \sigma)$-strongly convex and $(L + \beta_t)$-smooth, where we set $\beta_t \geq \sigma$. In this way, we can directly follow the proof of Theorem 5 which provides the generalization analysis on strongly convex problem. So we can obtain the same generalization error bound as follows:

$$
\mathbb{E}\left[|\ell(\boldsymbol{\theta}_T;\xi) - \ell(\widetilde{\boldsymbol{\theta}}_t;\xi)|\right] \leq G\mathbb{E}\left[\|\boldsymbol{\theta}_{t+1} - \widetilde{\boldsymbol{\theta}}_{t+1}\|_2\right] \leq \frac{2\alpha G^2}{n\left(\beta - \sigma + \frac{\alpha\mu L}{\mu+L}\right)}\frac{1 - \exp\left(-\frac{6sL}{\mu+L}\right)}{1 - \exp\left(-\frac{6L}{\mu+L}\right)}.
$$

Now we first consider one stage, such as the $s$-th stage. For brevity, let

$$
\boldsymbol{\delta}_\tau^{(t)} = \begin{cases} \|\boldsymbol{v}_0^{(t)} - \tilde{\boldsymbol{v}}_0^{(t)}\|_2 = \|\boldsymbol{\theta}_{t-1} - \widetilde{\boldsymbol{\theta}}_{t-1}\|_2, & \text{if } \tau = 0 \\ \|\boldsymbol{v}_\tau^{(t)} - \tilde{\boldsymbol{v}}_\tau^{(t)}\|_2, & \text{if } \tau \neq 0 \end{cases}
$$

Here we first consider each stage and then analyze multi-stage algorithm. We first analyze the $s$-th stage. Now we consider the average case where $\bar{\boldsymbol{v}}_k^{(T)} = \frac{1}{kT}\sum_{i=1}^{T}\sum_{\tau=0}^{k-1}\boldsymbol{v}_\tau^{(t)}$ is the output which is consistent with our optimization analysis which also needs output the average of all $\boldsymbol{v}_\tau^{(t)}$. To begin with, for the regularized function $F_t(\boldsymbol{\theta}) = F(\boldsymbol{\theta}) + \frac{1}{2}\|\boldsymbol{\theta} - \boldsymbol{\theta}_0\|$ where $\boldsymbol{\theta}_0$ denotes the output of the previous stage. Then by using induction, we can easily obtain

$$
\boldsymbol{v}_\tau^{(0)} = \boldsymbol{v}_0^{(0)} - \eta\sum_{i=0}^{\tau-1}(1 - \eta\beta)^{\tau-i-1}\boldsymbol{g}_i^{(0)}.
$$

This means $\boldsymbol{v}_k^{(0)} = \boldsymbol{v}_0^{(0)} - \eta\sum_{i=0}^{k-1}(1 - \eta\beta)^{\tau-i-1}\boldsymbol{g}_i^{(0)}$. Then because $\boldsymbol{v}_0^{(0)} = \boldsymbol{\theta}_0$, for $\boldsymbol{\theta}_1$, we have

$$
\boldsymbol{v}_0^{(1)} = \boldsymbol{\theta}_1 = (1 - \alpha)\boldsymbol{\theta}_0 + \alpha\boldsymbol{v}_k^{(0)} = \boldsymbol{v}_0^{(0)} - \alpha\eta\sum_{i=0}^{k-1}(1 - \eta\beta)^{k-i-1}\boldsymbol{g}_i^{(0)}
$$

Similarly, we can obtain

$$
\boldsymbol{v}_0^{(t+1)} = \boldsymbol{\theta}_t = \boldsymbol{v}_0^{(0)} - \alpha\eta\sum_{j=0}^{t-1}\sum_{i=0}^{k-1}(1 - \eta\beta)^{tk-jk-i-1}\boldsymbol{g}_i^{(j)}.
$$

In this way, we can know the formulation of the average of all $\boldsymbol{v}_\tau^{(t)}$ as follows:

$$
\begin{aligned}
\bar{\boldsymbol{v}}_k^{(T)} &= \frac{1}{kT} \sum_{i=1}^{T} \sum_{\tau=0}^{k-1} \boldsymbol{v}_\tau^{(t)} = \boldsymbol{v}_0^{(0)} - \alpha\eta \sum_{j=0}^{T-1} \sum_{i=0}^{k-1} (1-\eta\beta)^{Tk-jk-i-1} \boldsymbol{g}_i^{(j)} \\
&= (1-\eta\beta)\bar{\boldsymbol{v}}_{k-1}^{(T)} + \eta\beta\boldsymbol{v}_0^{(0)} - \alpha\eta\boldsymbol{g}_{k-1}^{(T)}
\end{aligned}
$$

Assume $\bar{\boldsymbol{v}}_k^{(T)}$ is obtained by running the algorithm on the dataset $\mathcal{S}$ and $\bar{\boldsymbol{v}}_k^{('T)}$ denotes the solution obtained by running the algorithm on $\mathcal{S}'$. In this way, we can conclude that for any $t$, we have

$$
\bar{\boldsymbol{v}}_k^{(t)} = \boldsymbol{v}_k^{(t)}, \qquad \bar{\boldsymbol{v}}_k^{('t)} = \tilde{\boldsymbol{v}}_k^{(t)}, \qquad \|\bar{\boldsymbol{v}}_k^{(t)} - \bar{\boldsymbol{v}}_k^{('t)}\| = \|\boldsymbol{v}_k^{(t)} - \tilde{\boldsymbol{v}}_k^{(t)}\|. \tag{13}
$$

In the following, we try to bound the difference between $\|\bar{\boldsymbol{v}}_k^{(T)} - \bar{\boldsymbol{v}}_k^{('T)}\|$. For each iteration $(t,\tau)$ in Algorithm 1, with probability $1 - \frac{1}{n}$, the current selected samples in $\mathcal{S}$ and $\mathcal{S}^{(i)}$ are the same. In this case, by using the first part of Lemma 1, we know that

$$
\begin{aligned}
\|\bar{\boldsymbol{v}}_{\tau+1}^{(t)} - \bar{\boldsymbol{v}}_{\tau+1}^{('t)}\|_2 &= \|(1-\eta\beta)\bar{\boldsymbol{v}}_\tau^{(t)} + \eta\beta\boldsymbol{v}_0^{(0)} - \alpha\eta\boldsymbol{g}_\tau^{(t)} - (1-\eta\beta)\bar{\boldsymbol{v}}_\tau^{('t)} - \eta\beta\boldsymbol{v}_0^{(0)} + \alpha\eta\boldsymbol{g}_\tau^{('t)}\|_2 \\
&= (1-\eta\beta)\left\|\bar{\boldsymbol{v}}_\tau^{(t)} - \frac{\alpha\eta}{1-\eta\beta}\boldsymbol{g}_\tau^{(t)} - \bar{\boldsymbol{v}}_\tau^{('t)} + \frac{\alpha\eta}{1-\eta\beta}\boldsymbol{g}_\tau^{('t)}\right\|_2 \\
&\overset{①}{\leq} (1-\eta\beta)\left(1 + \frac{\alpha\eta L}{1-\eta\beta}\right) \|\boldsymbol{v}_\tau^{(t)} - \tilde{\boldsymbol{v}}_\tau^{(t)}\|_2
\end{aligned}
$$

where ① uses the first part of Lemma 1.

Meanwhile with probability $\frac{1}{n}$, the selected samples are different in which we can use the first part of Lemma 1:

$$
\begin{aligned}
\|\boldsymbol{v}_{\tau+1}^{(t)} - \tilde{\boldsymbol{v}}_{\tau+1}^{(t)}\|_2 &= (1-\eta\beta)\left\|\bar{\boldsymbol{v}}_\tau^{(t)} - \frac{\alpha\eta}{1-\eta\beta}\boldsymbol{g}_\tau^{(t)} - \bar{\boldsymbol{v}}_\tau^{('t)} + \frac{\alpha\eta}{1-\eta\beta}\boldsymbol{g}_\tau^{('t)}\right\|_2 \\
&\overset{①}{\leq} (1-\eta\beta)\left(1 + \frac{\alpha\eta L}{1-\eta\beta}\right) \|\boldsymbol{v}_\tau^{(t)} - \tilde{\boldsymbol{v}}_\tau^{(t)}\|_2 + (1-\eta\beta)\frac{2\alpha\eta G}{1-\eta\beta} \\
&= (1-\eta\beta)\left(1 + \frac{\alpha\eta L}{1-\eta\beta}\right) \|\boldsymbol{v}_\tau^{(t)} - \tilde{\boldsymbol{v}}_\tau^{(t)}\|_2 + 2\alpha\eta G
\end{aligned}
$$

where ① holds by using the first part of Lemma 1. So combining these two cases yields

$$
\begin{aligned}
\mathbb{E}\left[\boldsymbol{\delta}_{\tau+1}^{(t)}\right] &= \|\boldsymbol{v}_{\tau+1}^{(t)} - \tilde{\boldsymbol{v}}_{\tau+1}^{(t)}\|_2 \\
&\leq \left(1 - \frac{1}{n}\right)\left[(1-\eta\beta)\left(1 + \frac{\alpha\eta L}{1-\eta\beta}\right)\|\boldsymbol{v}_\tau^{(t)} - \tilde{\boldsymbol{v}}_\tau^{(t)}\|_2\right] \\
&\quad + \frac{1}{n}\left[(1-\eta\beta)\left(1 + \frac{\alpha\eta L}{1-\eta\beta}\right)\|\boldsymbol{v}_\tau^{(t)} - \tilde{\boldsymbol{v}}_\tau^{(t)}\|_2 + 2\alpha\eta G\right] \\
&= (1 - \eta\beta + \alpha\eta L)\mathbb{E}\boldsymbol{\delta}_\tau^{(t)} + \frac{2\alpha\eta G}{n} \\
&= (1 - \eta\beta + \alpha\eta L)^{(t-1)k+\tau+1}\mathbb{E}\left[\boldsymbol{\delta}_0^{(0)}\right] + \frac{2\alpha\eta G}{n}\sum_{j=0}^{t-1}\sum_{i=0}^{\tau}(1-\eta\beta+\alpha\eta L)^{jk+i} \\
&= (1 - \eta\beta + \alpha\eta L)^{(t-1)k+\tau+1}\mathbb{E}\left[\boldsymbol{\delta}_0^{(0)}\right] + \frac{2\alpha G}{n}\frac{1}{\beta-\alpha L}\left[1 - (1-\eta\beta+\alpha\eta L)^{(t-1)k+\tau+1}\right].
\end{aligned}
$$

Then by using (13), we have that for the output of the $s$-th stage, it holds

$$
\begin{aligned}
&\mathbb{E}\left[\|\boldsymbol{\theta}_{s-\text{stage}} - \widetilde{\boldsymbol{\theta}}_{s-\text{stage}}\|_2\right] = \mathbb{E}\left[\|\boldsymbol{\theta}_{T_s} - \widetilde{\boldsymbol{\theta}}_{T_s}\|_2\right] = \mathbb{E}\left[\|\boldsymbol{v}_k^{(T_s)} - \tilde{\boldsymbol{v}}_k^{(T_s)}\|_2\right] \\
&\leq (1 - \eta_s\beta_s + \alpha\eta_s L)^{T_s k_s}\mathbb{E}\left[\|\boldsymbol{\theta}_{(s-1)-\text{stage}} - \widetilde{\boldsymbol{\theta}}_{(s-1)-\text{stage}}\|\right] + \frac{2\alpha G}{n}\frac{1}{\eta_s\beta_s - \alpha\eta_s L}\left[1 - (1-\eta_s\beta_s+\alpha\eta_s L)^{T_s k_s}\right].
\end{aligned}
$$

Assume that at stage $s$, we have $\mathbb{E}\left[\|\boldsymbol{\theta}_{s-\text{stage}} - \widetilde{\boldsymbol{\theta}}_{s-\text{stage}}\|_2\right] = 0$. Then it holds that

$$\mathbb{E}\left[\|\boldsymbol{\theta}_{s-\text{stage}} - \widetilde{\boldsymbol{\theta}}_{s-\text{stage}}\|_2\right]$$

$$\leq (1 - \eta_s\beta_s + \alpha\eta_s L)^{T_s k_s}\,\mathbb{E}\left[\|\boldsymbol{\theta}_{(s-1)-\text{stage}} - \widetilde{\boldsymbol{\theta}}_{(s-1)-\text{stage}}\|\right] + \frac{2\alpha G}{n}\frac{1}{\beta_s - \alpha L}\left[1 - (1 - \eta_s\beta_s + \alpha\eta_s L)^{T_s k_s}\right]$$

$$= \sum_{i=t_0+1}^{s} \frac{2\alpha G}{n}\frac{1}{\beta_i - \alpha L}\left[1 - (1 - \eta_i\beta_i + \alpha\eta_i L)^{T_i k_i}\right]\prod_{j=i+1}^{s}(1 - \eta_j\beta_j + \alpha\eta_j L)^{T_j k_j}$$

$$\leq \sum_{i=t_0+1}^{s} \frac{2\alpha G}{n}\frac{1}{\beta_i - \alpha L}\left[1 - (1 - \eta_i\beta_i + \alpha\eta_i L)^{T_i k_i}\right]\prod_{j=i+1}^{s}\exp\left(-\eta_j\beta_j T_j k_j + \alpha\eta_j T_j k_j L\right)$$

$$\overset{①}{\leq} \sum_{i=t_0+1}^{s} \frac{2\alpha G}{n}\frac{1}{\beta_i - \alpha L}\left[1 - (1 - \eta_i\beta_i - \alpha\eta_i L)^{T_i k_i}\right]\exp\left(-\frac{6(s-j)(\beta_s - \alpha L)}{\mu\rho\alpha_s}\right)$$

$$\leq \sum_{i=t_0+1}^{s} \frac{2\alpha G}{n}\frac{1}{\beta_i - \alpha L}\exp\left(-\frac{6(s-j)(\beta_s - \alpha L)}{\mu\rho\alpha_s}\right)$$

$$\leq \frac{2\alpha G}{n(\beta_s - \alpha L)}\frac{1 - \exp\left(-\frac{6s(\beta_s - \alpha L)}{\mu\rho\alpha}\right)}{1 - \exp\left(-\frac{6(\beta_s - \alpha L)}{\mu\rho\alpha}\right)}$$

where ① holds since for optimization, we set $\beta_{s+1} \leq \frac{1}{6}\mu, \eta_s \leq \frac{\rho\varepsilon_s}{3G^2}, \eta_s k_s T_s \geq \frac{6}{\mu\rho\alpha_s}$ and thus have $-\eta_j\beta_j T_j k_j + \alpha\eta_j T_j k_j L \geq -\frac{6(\beta_s - \alpha L)}{\mu\rho\alpha_s}$.

On the other hand, we have that function $\ell(f(\cdot;\boldsymbol{\theta});\cdot)$ is $G$-Lipschitz, and thus obtain

$$\mathbb{E}\left[|\ell(f(\boldsymbol{x};\boldsymbol{\theta}_T);\boldsymbol{y}) - \ell(f(\boldsymbol{x};\widetilde{\boldsymbol{\theta}}_T);\boldsymbol{y})|\right] \leq G\mathbb{E}\left[\|\boldsymbol{\theta}_{t+1} - \widetilde{\boldsymbol{\theta}}_{t+1}\|_2\right] \leq \frac{2\alpha G^2}{n(\beta_s - \alpha L)}\frac{1 - \exp\left(-\frac{6s(\beta_s - \alpha L)}{\mu\rho\alpha}\right)}{1 - \exp\left(-\frac{6(\beta_s - \alpha L)}{\mu\rho\alpha}\right)}.$$

**Step 3. Excess risk error by combining optimization error and generalization error.** Now we combine all results together, including the above optimization error and generalization error, and use Lemma 1 in the manuscript. Specifically, for the case $\sigma \leq \beta_s \leq \frac{1}{6}\mu$, we have

$$\varepsilon_{\text{opt}} + \varepsilon_{\text{gen}} \leq \frac{\Delta'}{2^s} + \frac{2\alpha G^2}{n\left(\beta - \sigma + \frac{\alpha\mu L}{\mu+L}\right)}\frac{1 - \exp\left(-\frac{6sL}{\mu+L}\right)}{1 - \exp\left(-\frac{6L}{\mu+L}\right)}.$$

where $\Delta' = \mathbb{E}[F(\boldsymbol{\theta}_0) - F(\boldsymbol{\theta}_{\mathcal{S}}^*)]$. Then for the case where there is a constant $\rho$ such that $\langle\nabla F_t(\boldsymbol{\theta}), \boldsymbol{\theta} - \boldsymbol{\theta}_{\mathcal{S}}^*\rangle \geq \rho(F_t(\boldsymbol{\theta}) - F_t(\boldsymbol{\theta}_{\mathcal{S}}^*))$, we have

$$\varepsilon_{\text{opt}} + \varepsilon_{\text{gen}} \leq \frac{\Delta'}{2^s} + \frac{2\alpha G}{n(\beta_s - \alpha L)}\frac{1 - \exp\left(-\frac{6s(\beta_s - \alpha L)}{\mu\rho\alpha}\right)}{1 - \exp\left(-\frac{6(\beta_s - \alpha L)}{\mu\rho\alpha}\right)}.$$

The proof is completed. $\qquad\square$