# OpenReview forum: "Towards Understanding Why Lookahead Generalizes Better Than SGD and Beyond"
_NeurIPS.cc/2021/Conference — NeurIPS 2021 Poster_

### Official Review · Reviewer_bSD7 · 2021-07-13

**Rating:** 6
**Confidence:** 3

**Summary:**

The paper provides the excessive risk rounds for the lookahead algorithms on (strongly) convex and non-convex problems with PL condition. The theoretical results are obtained via (1) convergence analysis (2) the uniform stability analysis. From the excessive risk bounds, the authors provide justification for the performance of the lookahead algorithms. Finally, the paper proposes a stagewise locally-regularized lookahead algorithm called SLRLA, which adds a L2 regularizer and decays the learning rate for each stage.

**Limitations And Societal Impact:**

Yes.

**Main Review:**

## Originality

The convergence analysis and the uniform stability analysis of the lookahead algorithms are new and interesting.

## Quality

The theoretical part of the paper is sound.

- The authors provide convergence analysis for lookahead algorithms under different settings, e.g., (strongly) convex and PL-condition. For the generalization part, the authors derive the uniform stability bounds for lookahead algorithms. By combing two parts, the authors show that there is an optimal choice for $\alpha$ which balances optimization error and generalization error.

The empirical result is strong.

- The authors test their proposed algorithm in three commonly used datasets. They show that SLRLA is better than other baseline algorithms.

Some concerns for the theory part:

- One caveat of the uniform stability analysis is that for general convex problem (not strongly convex), the generalization depends badly on $T$. Therefore if we calculate the optimal $\alpha$ we found that $\alpha$ scales with $1/T$, which means for small excessive risk we might need to restrict the effective learning rate.

- Line 163: In [41], actually the optimal rate for SGD is $O(1/\lambda T)$. Also in Line 291: The dependency for $\lambda$ is optimal for SLA and SLRLA, but I think the $O(1/\lambda^2)$ bound for lookahead algorithms is due to your analysis techniques. The comparison may be unfair.

## Clarity

The paper is well-written.

**Time Spent Reviewing:**

1.5

---

> ### Author Response · Authors · 2021-08-09
> **Response to Reviewer bSD7**
>
> Thank you for the insightful and encouraging comments! We hope the concerns about theory can be addressed by the following clarification.
>
> (1) Regarding the desirable choice of $\alpha$ for arbitrary convex loss, we would like clarify that we are mostly interested in excess risk bound which scales as $=O(\frac{1}{\alpha \eta k T}+ \eta G^2+\frac{\alpha \eta G^2 kT}{n})$ in this case (see line 224). With the conventional choice of $\eta=\frac{1}{G\sqrt{kT}}$, we can set $\alpha=1\wedge \sqrt{\frac{n}{kT}}$ to  achieve excess risk bound of scale $O(\frac{1}{\sqrt{n}}+ \frac{1}{\sqrt{kT}})$. We believe such a choice of $\alpha$ is reasonable for LA as $kT$ is usually at the same scale of $n$. Per your comment, we will add a more comprehensive discussion on this important issue at the end of Sec. 4.2.
>
> (2)  Under similar assumptions,  Rakhlin et al. proved that for SGD, its  optimization error  is $O(\frac{1}{\lambda^2 kT})$ for the last-iterate  $\theta^{T}$ (see their Theorems 1 \& 2), and the error becomes $O(\frac{1}{\lambda kT})$ if one uses their proposed $\gamma$-Suffix averaging ($\gamma$-SF, i.e., averaging all parameters $(\theta_{i})_{i=(1-\gamma)T}^T$ at the last $\gamma T$ iterations) as the algorithm output (see their Theorem 5).  In our work, we focus on $\theta^{T}$ output at the last iteration, as this is a more common option than the $\gamma$-SF output in practice, especially in deep learning regime where LA targets to serve.  In this case, we derive  the optimization error $O(\frac{1}{\lambda^2 kT})$ for LA and $O(\frac{1}{\lambda kT})$  for SLA and SLRLA. By using the $\gamma$-SF, one could expect to improve the factor $\frac{1}{\lambda^2}$ in LA to $\frac{1}{\lambda}$. However, as discussed in Sec. 5.1 that SLA and SLRLA also improve LA in terms of other factors (e.g. $\alpha$) even though they do not use the $\gamma$-SF trick. We will update these explanations into the revision to better highlight the advantages of SLA and SLRLA over the conventional LA method.

---

> > ### Comment · Reviewer_bSD7 · 2021-08-12
> > **Reply**
> >
> > I sincerely thank the authors for the detailed feedback. I would maintain my score and provide the authors with additional comments below.
> >
> > ### (1).
> >
> > For writing, it would be better to include the possible choice that $\eta=O(1/\sqrt{T})$ and $\alpha=O(1/\sqrt{T})$ and the resulting risk bound into the paper.
> >
> > I still want to make clear my point about the caveat of the uniform stability analysis. For this choice of $\eta$ and $\alpha$, the final parameter can be viewed as being restricted in a ball of $O(1)$-radius. This is because for each step the effective displacement of the model parameter is $O(1/T)$, and $T$ steps constitute a displacement of order $O(1)$. Therefore, to ensure that you get a non-vacuous bound, you are actually **restricting** the model parameter in a ball of $O(1)$-radius. Therefore the uniform stability analysis may not be a good way to study the generalization property of algorithms on general convex/non-convex models.
> >
> > ### (2).
> >
> > For theoretical analysis and comparison, I don't think it is good to avoid LA + iterate-averaging by arguing that in practical deep models people just use the last iterate. Also, please notice that for both iterate-averaging and the $O(1/\lambda T)$ v.s.  $O(1/\lambda^2 T)$ we are considering the **strongly convex** case (which, by the virtue of your argument, is already not common). Therefore, it is not wise to claim SLA/SLRLA beat the "not-optimal" LA in terms of the dependency of $\lambda$.

---

> > > ### Author Response · Authors · 2021-08-15
> > > **Response to Reviewer bSD7**
> > >
> > > Thanks for your  insightful comments reply and further clarifying your questions. Per your suggestions, we will include the excess risk error discussion when $\eta = O(\frac{1}{\sqrt{kT}})$ and $\alpha=O(1\wedge \sqrt{\frac{n}{kT}})$, and also comprehensively explain the advantages of SLA/SLRLA over LA in terms of factor $\lambda$.
> > >
> > >
> > > In the following, we want to further explain our points for your question (1). Let $\Delta=\mathbb{E}[\|\|\theta_0-\theta_S^{\star}\|\|^2]$ with  initialization $\theta_{0}$ and optimum $\theta_S^{\star}$. For $\eta = \frac{\sqrt{\Delta}}{G\sqrt{kT}}$, we follow its conventional choice in many optimization works, e.g. [35] and its reference [29], to obtain the optimal optimization error of vanilla SGD. Note, the optimal $\eta = \frac{\sqrt{\Delta}}{G\sqrt{kT}}$ is also demonstrated by our Theorem 3, since  when $\alpha=1$, LA degenerates to SGD and thus the optimization error of SGD is  $O(\frac{\Delta}{\eta k T} + \eta G^2 )$.  In this way, we should set $\alpha=1\wedge \sqrt{\frac{n}{kT}}$ to achieve smallest excess risk error. In this way, by running $kT$ iterations, the distance $\|\|\theta_{T} - \theta_{0}\|\|$ between the initialization $\theta_{0}$ and  the solution $\theta_{T}$ found by LA is at the order of $\|\|\theta_{T} - \theta_{0}\|\| = O(\eta \alpha \|\|\sum_{t=1}^{T}\sum_{\tau=0}^{k-1} g_{\tau}^{(t)} \|\|)$ (see line 166), where $g_{\tau}^{(t)}$ is the stochastic gradient at the $(t,\tau)$-th iteration.  So if viewing $\|\|g_{\tau}^{(t)}\|\|$ as a constant $G$ ($G$-Lipschitz assumption in Theorem 3), then the distance $\|\|\theta_{T} - \theta_{0}\|\|$ becomes $O(\eta \alpha \sqrt{kT} G) = O(\sqrt{\Delta})$. In this way,  the solution $\theta_{T}$  found by LA can approach/converge to the optimum solution $\theta_{S}^{\star}$, since $\theta_{T}$ locates at a point around the initialization $\theta_0$ with $O(\sqrt{\Delta})$-radius and $\Delta$ is defined as $\Delta=\mathbb{E}[\|\|\theta_0-\theta_S^{\star}\|\|^2]$.  Ignoring the constant factor $\Delta$ in optimal $\eta$ leads to the spurious $O(1)$-radius bound between $\theta_0$ and $\theta_T$. We will update   these explanations into the revision to help readers better  understand.  Thanks.

---

### Official Review · Reviewer_bbpy · 2021-07-14

**Rating:** 7
**Confidence:** 3

**Summary:**

Lookahead is a relatively recent algorithm that has shown great performances in both single-objective and two-objective training, but it is not well understood theoretically why it improves training particularly in the former case. This paper aims to elaborate why Lookahead improves test accuracy using excess risk error which is decomposed into optimization and generalization terms. Interestingly, the authors show that the extra hyperparameters in Lookahead when combined with SGD allow for a better trade-off between the two terms relative to SGD. Moreover, they propose a new "stagewise" variant algorithm that: (i) uses constant step sizes for Lookahead, but changes/decreases these at each stage, and (ii) adds regularization term that enforces doing more "conservative" steps (the term penalizes high distance between predicted iterate and last iterate of the previous stage, see Alg.2).

**Limitations And Societal Impact:**

Yes

**Main Review:**

In my opinion, this paper does a nice balance between theoretical discussions and experiments, and I find it very well written. Regarding the latter, the authors introduce the necessary definitions, list theoretical results, but it is nice that they also discuss the results.

While I enjoyed reading the first part discussing the original Lookahead (LA), I have several questions/concerns for the newly proposed algorithm called SLRLA.


--- Questions ---

1. Following your analysis on LA, could you elaborate on how SLRLA is motivated? While I understand that given its formulation you obtain a small excess risk error, my question is towards understanding how your LA analysis guided finding new algorithms,  and it might be useful adding such a comment to also connect the two parts.

2. Is introducing the stage-wise step-size strategy necessary for SLRLA, or alternatively, is the excess risk error worse in that case? In contrast, if I understand well, for LA you used decaying learning rate in the analysis. I am not sure I understand, could a stage-wise step-size strategy provide better excess risk error for LA than the one provided?

3. While SLRLA has better excess risk error, it also introduces two additional hyper-parameters making it harder to tune ($Q$ and $\beta_q$). Could you comment from the analysis perspective whether the obtained results require optimal hyperparameters? Also, is $\beta_q$ easy to select in your experiments, and do the performances vary across the values you used in the cross-validation?

4. Regarding the empirical setup, seems like only the $\beta_q$ hyperparameter of SLRLA is cross-validated, and the rest are fixed. This might give an advantage to SLRLA.

5. Figure 1. Surprisingly, LA in (b) performs poorly (for MNIST). How are the hyperparameters selected for it, and what are the performances of SGD? Also, as test accuracy is not used in both a.3 and b, it is harder to compare the performances of SLRLA and SLA with varying hyperparameters.


6. As a comment, in my opinion, the nested version of Lookahead proposed in [21] -- which applies  LA recursively -- is related to SLRLA as both push the current iterate toward a past "anchored" iterate, but they are yet different methods. It would be interesting to explore how the two differ (in terms of the error trade-off).


--- Minor/typos ---


ln.67: has->have

Fig.2: maybe ensure legend text does not overlap much,  and improve caption

--- Recommendation ---


In summary, I find the result/analysis on the original Lookahead interesting and intuitive, and in my opinion, the paper is well-written. In terms of novelty,  (i) the analysis used in this paper is not novel, (ii) while the paper shows how LA hyperparameters influence the trade-off between the two errors terms, it is not clear how these can be selected -- thus several questions still remain open for LA (how to select the hyperparameters of LA in practice and deriving tighter rates for it).
However, I would be glad to raise my score if the authors address some of the above questions.

--- Update after discussion ---

The authors addressed most of my concerns, and I increased my score.

**Time Spent Reviewing:**

7

---

> ### Author Response · Authors · 2021-08-09
> **Response to Reviewer bbpy**
>
> Thank you for the insightful and encouraging comments! We hope the main concerns can be addressed by the following clarification.
>
> (1)	Motivation of SLRLA. As shown in Eqn. (2) and the analysis of SLA (e.g. Theorems 3 \& 4), the excess risk of an algorithm can be decomposed into an optimization error term $\varepsilon_{\text{opt}}$ and a generalization error term $\varepsilon_{\text{gen}}$, which inspires us to reduce $\varepsilon_{\text{opt}}$  and $\varepsilon_{\text{gen}}$ as much as possible. To this end, one of the simplest and most natural methods is regularization. This point is also supported by our analysis for LA and SLA. Particularly for strongly-convex problems, as shown in Theorems 1 \& Theorem 2,    the optimization error and generalization error depend on  $O(1/\lambda)$, where $\lambda$ denotes the strongly-convex parameter. For nonconvex problems, Theorem 4 also shows that optimization error and generalization error scale with $O(1/\mu)$, where $\mu$ is the PL condition parameter. Therefore, for strongly convex problems, adding a regularization to the vanilla loss can enhance the convexity and reduce both optimization and generalization errors; on nonconvex problems, regularization can also increase the PL condition parameter $\mu$ and thus help generalization.
>
> Keeping the above arguments in mind, the remaining questions are (a) what kinds of regularization we should add? And (b) in which step in Algorithm 2 should we add the reguarlizer?  For question (a), as mentioned in lines 260-263, one often uses $\|\|\theta - 0\|\|^2$ or $\|\|\theta-\theta_0\|\|^2$ as the regularizer. In this work, we use $\|\|\theta-\theta_{q-1}\|\|^2$ instead, since compared with 0 and $\theta_0$, $\theta_{q-1}$  is closer to the optimum $\theta^*$ of the vanilla loss and thus allows us to use larger regularization modulus $\beta_{q}$, improving the convexity of a loss more and benefiting convergence and generalization more (see theoretical results and discussions in Sec. 5.1 \& 5.2). Concerning question (b), there are two common choices. The first one is to impose the regularization at the beginning of each stage, which corresponds to Algorithm 2. The second one is more general, i.e. adding a regularization for every $c$-iterations with a constant $c$. This latter strategy, however, is hard to analyze. So in this work, we choose to work on the former one which has provable strong theoretical guarantees and also works very well in practice.
>
> The above intuition has been briefly mentioned at the beginning of Sec. 5.1. We will explain this point in more detail in the revised paper.
>
>
> (2) Regarding the stagewise learning rate (LR) decaying  strategy, we would like to stress that it does benefit SLRLA. We first introduce why it is important in lookahead (LA) by using our results on strongly-convex problems, since SLA is a special case of SLRLA where $\beta_{q}=0$. Sec. 4 shows that to achieve $\epsilon$-optimization error, LA with linear LR decaying strategy for strongly-convex problems has stochastic computational complexity (SCC) $O((\frac{L}{\epsilon})^{\frac{1}{2\alpha}} + (\frac{LG^2}{(1-2\alpha)\lambda^2\epsilon})^{\frac{1}{2\alpha}})$ for $\alpha\in(0,\frac{1}{2})$,  	$O(\frac{LG^2 \log\frac{1}{\epsilon}}{(1-2\alpha)\lambda^2\epsilon})$ for $\alpha=\frac{1}{2}$,  and $O(\frac{LG^2}{(2\alpha-1)\lambda^2\epsilon})$ for $\alpha\in(\frac{1}{2},1]$. In contrast, Sec. 5 proves that  stagewise LA (SLA) which uses stagewise LR decaying strategy enjoys $O(\frac{G^2}{\lambda \alpha \epsilon})$ SCC complexity which is obtained by setting $\beta_{q}=0$ in Theorem 5. By comparison, for factor $\lambda$, SLA improves  $O(\frac{1}{\lambda^2})$ the dependence in LA to $O(\frac{1}{\lambda})$.  For factor $\alpha$, SLA improves the  exponential dependence to linear dependence for $\alpha\in(0, \frac{1}{2})$, and removes the factor $\log \frac{1}{\epsilon}$ in LA.  For generalization error, both LA and SLA have the same error order $O(\frac{G^2}{n\lambda})$. So SLA has smaller excess risk error than LA, which shows the advantages of the  stagewise LR decaying strategy.
>
>
> For SLRLA on strongly-convex problems,  if we use linearly-decaying LR instead of stagewise LR decaying strategy, we cannot derive the linear convergence rate in Theorem 5, which means larger optimization error. For generalization error, following similar proof of Theorem 2, linearly-decaying LR leads to an error bound  $O(\frac{G^2}{n\lambda})$ similar to the results in Theorem 2. Overall, these results show the importance of the stagewise LR decaying strategy in SLRLA. Per your comment, we will further clarify this point in revision.
>
>
> (3)	Compared with SLA, SLRLA only introduces one additional hyper-parameter $\beta_q$, since one often uses stagewise lookahead (SLA) in practice which also involves stage number $Q$.  For both $Q$ and $\beta_{q}$, our analysis, e.g. Theorem 5, only requires them to satisfy certain conditions, e.g. $Q \geq \log  \frac{\Delta'}{\epsilon}$ and $\beta_{q} \leq \frac{1}{6} \lambda$ in Theorem 5. So our analysis does not require specific optimal values of $Q$ and $\beta_{q}$, which gives value ranges for practicers to tune. For experiments,  we follow conventional  stagewise  setting in [1, 49, 50] to set $Q$: $Q=4$ and decaying LR by $0.2$ at the ${0.3S, 0.6S, 0.8S}$-th epoch with total epoch number $S$.  For $\beta_{q}$, we select it from the set {$0.02,0.2,2,20$} via cross-validation, which is simple and commonly used in practice. In our cross-validation, for simplicity,  we randomly select some training samples (e.g. 20000 on ImageNet) as the validation set to tune $\beta_{q}$ and use the remaining data to train the model.  The selected value of $\beta_{q}$ is always 0.2 in three trials. Moreover, by fixing the training set, Fig. 4 in Appendix reports the test performance of different $\beta_{q}$ on ImageNet, and shows the stable performance of SLRLA when $\beta_{q}$ varies in a large range. This is mentioned in line 379.  For other randomness, e.g. minibatch sampling and net initializations, we run five times by selecting five different initialization seed and sampling seed, and report the mean and variance in Table 1 which shows stable performance as well.
>
>
> (4)  For other parameters besides $\beta_q$, e.g. learning rate, we largely follow the conventional settings in [1, 49, 50]. Among others, we note that AdaBelief (see its Sec. 3) and SLA (see its Appendix C) directly do grid search to select hyper-parameters on the test dataset, e.g. selecting LR from {$0.01, 0.02, 0.05, 0.1, 0.2, 0.3$} in SLA. So their hyper-parameter selection strategy is actually much stricter than ours, and their selected  hyper-parameters may not be optimal for ours.  Even under this case, our SLRLA  achieves superior performance on three commonly used datasets.  So the advantage of SLRLA comes from the algorithm itself rather than hyper-parameter tuning.
>
>
> (5) LA on MNIST achieves an unsatisfactory accuracy (90.1\%), as it uses linearly decaying LR which decays  LR  fast and could not effectively update the model. In the experiments, we tune the initial LR for LA and report its highest accuracy 90.1\%.  In contrast, stagewise LA (SLA) only decays LR several times during training and avoids too small LR, achieving 92.8\% accuracy.   Indeed, Zhang et al. proposed LA but reported experimental results of SLA in their paper.  SGD with linearly decaying LR achieves 89.6\%, and stagewise SGD  achieves a higher accuracy 92.5\%. These empirical results well support Theorem 5 which shows the advantage of SLA over LA.  Meanwhile, this also explains why one often uses stagewise LR decaying strategy in practice.
>
> The three subfigures in Fig. 1 (a) aim to investigate the effects of $\alpha$ on the behaviors of optimization error and generalization error which are often defined on the function loss.   Moreover, we found that smaller test loss generally gives higher test accuracy which accords with most cases in practice. So we only report test loss in Fig. 1 (a.3).  For clarity in comparison, we only report the highest accuracy of SLA ($\alpha=0.5$) and lookahead ($\alpha=0.8$), and investigate the test accuracy of SLRLA under varying $\alpha$ for comparison. Per your comment, in revision, we will additionally provide the test accuracy comparison by using the results in Fig. 1 (a.3).
>
> (6) Chavdarova et al. [21] proposed a lookahead-minmax algorithm for minmax problems which achieves promising results. As shown in their Algorithm 1, in the inner-loop, they respectively updated the parameters $w_{\text{min}}$ of minimization problem and $w_{\text{max}}$  of  maximization problem with an inner optimizer to obtain $w_{\text{min}}^{k}$ and $w_{\text{max}}^{k}$; then in the outer-loop, they receptively updated  $w_{\text{min}}^{+} = w_{\text{min}}^{+} + \alpha_{\text{min}} (w_{\text{min}}^k - w_{\text{min}})$  and $w_{\text{max}}^{+} = w_{\text{max}}^{+} + \alpha_{\text{max}} (w_{\text{max}}^k - w_{\text{max}})$. One can observe that their updating steps are vanilla lookahead  but on minmax problems, and do not introduce any extra  "anchored point". In contrast, we divide the whole optimization process into $Q$ stages, and introduce an extra "anchored point" $\theta_{q-1}$ to construct a locally-regularized loss $F_q(\theta)= F(\theta) + \frac{\beta_q}{2} \|\| \theta- \theta_{q-1}\|\|^2$ where $\theta_{q-1}$ denotes the output of the previous stage. So these two algorithms differ  from each other very much. We will discuss this in revision.

---

> > ### Comment · Reviewer_bbpy · 2021-08-29
> > **Thanks for the detailed response**
> >
> > The authors have addressed most of my concerns, and as promised I am increasing my score from 6 to 7.
> >
> > Thanks

---

> > > ### Author Response · Authors · 2021-08-29
> > > **Response to Reviewer bbpy**
> > >
> > > We are very glad that our response can address your concerns. Thanks for your insightful comments and positive feedback again!！

---

### Official Review · Reviewer_3SmH · 2021-07-15

**Rating:** 7
**Confidence:** 3

**Summary:**

The authors analyze the Lookahead optimizer (Zhang et al.) algorithm, showing that it has nice bounds on the excess risk error compared to SGD on strongly convex problems and noncovex problems which satisfy the PL condition. They also propose a locally-regularized version of the Lookahead algorithm which has some better theoretical properties and performs well on empirical evaluations.

**Limitations And Societal Impact:**

Limitations are sufficiently discussed throughout the paper and in the appendix. The authors do not directly address societal impact ("This is not applicable to a theoretical work"), though I believe it is worth some further thought e.g. see the AdaBelief paper.

**Main Review:**

Strengths
- The theoretical results are novel and useful for understanding the lookahead algorithm and how it compares to SGD. Theorem 1 provides bounds on the optimization error of lookahead(SGD); Theorem 2 bounds the generalization error.
- The authors do a good job of offering intuition for the their theoretical results and discussing implications.
- Stagewise locally-regularized lookahead is well-motivated and the proof is technically sound. It is impressive to improve on an existing, popular algorithm.
- While most of the results are theoretical, the empirical evaluations are strong and performance on CIFAR and ImageNet exceeds the performance on the original Lookahead algorithm across 5 seeds.

Weaknesses
- Writing can be a bit choppy at points and would benefit from careful proofreading.

Questions
- Does stagewise lookahead refer to conventional lookahead (Zhang et al.) with a stagewise learning rate (geometric) decay? If so, I believe it would be better to make this point more explicit and mention that it is the learning rate schedule typically used.
- Could you say anything about lookahead with other inner optimizers?

Minor
- 83: the sharp minima story is a bit more nuanced (as mentioned in some of your references) or "Sharp Minima Can Generalize For Deep Nets" (Dinh et al.)
- 86: typo in lookahead
- In algorithm 1, it would improve clarity if the order of the description of the inputs matches that of the function arguments

Post-rebuttal: I have read the other reviewer comments and responses and am maintaining my score.

**Time Spent Reviewing:**

9

---

> ### Author Response · Authors · 2021-08-09
> **Response to Reviewer 3SmH**
>
> Thank you for the insightful review and positive feedback!
>
> (1) Yes, the stagewise lookahead (SLA) method refers to a variant of  the conventional lookahead with stagewise-geometrically decaying learning rate (LR). Per your suggestion, we will further clarify this point in the context and explicitly stress that such a stagewise LR strategy is typically used in many works (see, e.g., Zhang et al., 2019).
>
>
> (2) Regarding the lookahead method with inner optimizers other than SGD, we believe that one could still expect similar performance trade-off between optimization and  generation with respect to the choice of $\alpha$. Intuitively, for any inner optimizer, let  $g_\tau^{t}$ denote the "gradient" (or any descent direction) at the $(t,\tau)$-th iteration. After the $k$ inner-steps, lookahead updates parameter $\theta$ as $\theta_t=\theta_{t-1}+\alpha(v_{t}-\theta_{t-1})=\theta_{t-1}-\alpha \sum_{\tau=1}^{k} g_\tau^{t}$.  Obviously, $\alpha \approx 1$ is preferable for preserving the optimization speed of the inner optimizer.
>
> When it comes to the generalization error, obviously the best possible performance occurs at the initialization point as it is not dependent on the training data. Along with more training iterations, the network will gradually fit the training data and thus could give larger and larger prediction discrepancy between training data and test data. Therefore, it is desirable to have $\alpha\ll 1$ as opposed to $\alpha \approx 1$ for generalization.
>
> Overall, for generic inner-loop optimizers, lookahead is still expected to be able to balance the optimization and generalization performances with proper choices of $\alpha$.
>
>
> (3) We appreciate your minor comment on our statement about sharp minima theory and would like to provide a more careful discussion on this point in the revised paper. Thanks for pointing out the typos!
>
>
> (4) Per your suggestion, we will carefully proofread our work and try our best to improve its readability.

---

### Decision · Program_Chairs · 2021-09-27

**Decision:**

Accept (Poster)

**Comment:**

A well written paper with solid theoretical contributions; well received by the reviewers. The authors also did a very good job of giving long and detailed responses to any questions and concerns.